# CONVERGENCE OF ADAFACTOR UNDER NON-CONVEX SMOOTH STOCHASTIC OPTIMIZATION

## ABSTRACT

Adafactor, a memory-efficient variant of Adam, has emerged as one of the popular choices for training deep learning tasks, particularly large language models. However, despite its practical success, there is limited theoretical analysis of Adafactor's convergence. In this paper, we present a comprehensive analysis of Adafactor in a non-convex smooth setting. We show that full-batch Adafactor finds a stationary point at a rate of $\tilde{O}(1/\sqrt{T})$ with the default setup, which could be accelerated to $\tilde{O}(1/T)$ with a constant step-size parameter. For stochastic Adafactor without update clipping, we prove a convergence rate of $\tilde{O}(1/\sqrt{T})$ with the right parameters covering the default setup. We also prove that Adafactor with a time-varying clipping threshold could also find a stationary point with the rate of $\tilde{\mathcal{O}}(1/\sqrt{T})$. Our theoretical results are further complemented by some experimental results.

## 1 INTRODUCTION

The adaptive gradient-based methods, such as the well-known AdaGrad (Duchi et al., 2011; Streeter & McMahan, 2010), RMSProp (Tieleman & Hinton, 2012), Adadelta (Zeiler, 2012), Adam (Kingma & Ba, 2015), and AMSGrad (Reddi et al., 2018) are one of the preferred approaches in solving the following unconstrained stochastic optimization problem in deep learning fields:

$$\min_{\boldsymbol{X} \in \mathbb{R}^{n \times m}} f(\boldsymbol{X}) = \mathbb{E}_{\boldsymbol{Z} \in \mathcal{P}}[l(\boldsymbol{X}; \boldsymbol{Z})], \tag{1}$$

where $l$ is a smooth potentially non-convex function and $\mathcal{P}$ denotes a probability distribution. During the training process, these adaptive methods require storing the historical gradients' information to tune their step-sizes adaptively. For example, both Adam and AdamW maintain the exponential average of gradients and squared gradients, and AdaGrad stores the cumulative of squared gradients. Despite their effectiveness, these algorithms pose substantial memory challenges for GPUs to save these additional gradients' information, especially when training large language models (LLMs), such as GPT-3 (Brown et al., 2020), which contains over 175 billion parameters.

To address memory constraints, several memory-efficient optimization algorithms have been developed, e.g., (Shazeer & Stern, 2018; Anil et al., 2019; Luo et al., 2023; Li et al., 2024). One of the popular optimizers is Adafactor (Shazeer & Stern, 2018), a memory-saved variant of Adam that employs a rank-1 matrix factorization to approximate the second-moment matrix. For an $n \times m$ weight matrices, this technique reduces memory from $\mathcal{O}(mn)$ to $\mathcal{O}(m + n)$ by only tracking the moving averages of the row and column sums of the squared gradients matrix. Additionally, Adafactor eliminates the first-order momentum used in Adam and incorporates update clipping to enhance training stability. In real applications, several LLMs including PaLM (Chowdhery et al., 2023)[1] and T5 (Raffel et al., 2020) have included Adafactor into their main optimizers (Zhao et al., 2023).

The empirical results reveal that Adafactor achieves comparable performance to Adam in training Transformer models (Shazeer & Stern, 2018), even though it sacrifices gradient information to save the memory. Despite Adafactor's empirical success, there is limited understanding on its convergence in theory, especially the explanation for its hyper-parameter setting in experiments. Theoretical results, e.g., (Zou et al., 2019; Défossez et al., 2022), have proved that Adam could converge to the stationarity with $\tilde{\mathcal{O}}(1/\sqrt{T})$ rate under specific hyper-parameter for non-convex smooth setup. Then, a natural question arises:

---

[1]PaLM applies Adafactor without matrix factorization.

> *Could Adafactor still achieve the same order of convergence rate as Adam while sacrificing gradient information for improved memory efficiency? If so, what's the requirement for its hyper-parameters setup?*

In this paper, we analyze Adafactor's convergence for non-convex smooth optimization problems, considering the typical bounded gradient setting as those for AdaGrad (Li & Orabona, 2019; Ward et al., 2020) and Adam (Zaheer et al., 2018). We aim to provide a similar convergence rate for Adafactor which complements the empirical observation that Adafactor could attain comparable performance to Adam while reducing memory usage. The analysis is non-trivial compared to other adaptive methods such as AdaGrad and Adam due to the unique matrix factorization and update clipping mechanisms in Adafactor. In the full-batch case, we rely on the special exponential moving averages of the row sums and column sums of the squared gradients to lower bound the first-order term in the Descent Lemma. In the stochastic case, we design a new proxy step-size to compute the conditional expectation of the first-order term that involved the stochastic gradient and the adaptive step-sizes. Further, we successfully control the additional error brought by this proxy step-size. We also extend a standard way in the analysis of SGD with clipping to handle the update clipping. Our main contributions are summarized as follows.

- We provide a convergence analysis for full-batch Adafactor under bounded gradients and a broader range of hyper-parameter settings which covers the default one in (Shazeer & Stern, 2018). The result shows that Adafactor can find a stationary point $\tilde{\mathcal{O}}(1/\sqrt{T})$ rate with default step-sizes. This rate can be accelerated to $\tilde{\mathcal{O}}(1/T)$ with a constant step-size parameter.

- We further investigate the more realistic stochastic Adafactor. It's found that a simple variant of Adafactor, which drops the update clipping, could attain the convergence rate of $\tilde{\mathcal{O}}(1/\sqrt{T})$ when the decay rate of the second moment is $1 - 1/k$. This rate is optimal, matching the lower bound (Arjevani et al., 2023) up to logarithm factors.

- We extend our study to include a time-varying clipping threshold. Our analysis implies that with proper selections of clipping threshold and hyper-parameters, Adafactor could also achieve the best convergence rate of $\tilde{\mathcal{O}}(1/\sqrt{T})$.

- We further provide some basic experiments on computer vision and natural language processing tasks to complement our theoretical results.

The rest of the paper are organized as follows. The next section provides some most relevant works. Section 3 presents some necessary notations definitions and problem setup. Section 4 reviews Adafactor and its major differences to Adam. Sections 5 and 6, separately provide convergence bounds for full-batch Adafactor and stochastic Adafactor (no update clipping) and further discuss the hyper-parameters' dependency. Section 7 investigates Adafactor using a time-increasing update clipping threshold. Section 8 summarizes the main proof challenges brought by Adafactor and our proof novelty. Section 9 provides experimental results to complement our theory. All the detailed proofs and some experiments can be found in the appendix.

## 2 RELATED WORK

Although there are limited works on Adafactor in theory, the convergence for other memory-unconstraint adaptive methods are widely studied. Here, we briefly list some typical works due to the page limitation.

**Convergence of adaptive methods.** Several studies (Li & Orabona, 2019; Ward et al., 2020; Zou et al., 2019) prove the convergence of AdaGrad in non-convex smooth settings assuming bounded stochastic gradients. Shi et al. (2020) shows that RMSProp could converge to the stationarity when the decay rate of the second moment is close to one. Several works (Chen et al., 2019; Zhou et al., 2020; Alacaoglu et al., 2020) provide convergence bounds for AMSGrad in non-convex smooth settings. A line of research, e.g., (Zaheer et al., 2018; De et al., 2018; Zou et al., 2019; Défossez et al., 2022) have investigated the convergence of Adam assuming bounded gradients and noise. Yao et al. (2021) designed AdaHessian, using Hutchinson's approximation to estimate the diagonal Hessian.

Sadiev et al. (2023) provided Scaled SARAH and Scaled L-SVRG to approximate the diagonal Hessian when the problem is ill-conditioned. Recently, several works focused on the heavy-tail noise, showing that clipping is also necessary for AdaGrad (Li & Liu, 2023) and Adam (Chezhegov et al., 2024) as the one for SGD (Gorbunov et al., 2020).

**Memory efficient algorithms.** As large models are increasingly used in deep learning, memory constraints have become a central issue during training. Consequently, several memory-efficient optimizers have been developed to address this challenge.

One approach to saving memory involves applying matrix factorization to optimization algorithms. For instance, Shazeer et al. (2017) used matrix factorization in the second moment estimator of gradients in Adam, similar to the concept behind Adafactor. Luo et al. (2023) introduced CAME, a variant of Adafactor, which incorporates a confidence-guided strategy to mitigate instability caused by erroneous updates. Zhao et al. (2024) proposed Adapprox, leveraging randomized low-rank matrix approximation for Adam's second moment estimator, demonstrating superior performance and reduced memory usage compared to AdamW.

There are some other techniques to save the memory. For example, Gupta et al. (2018) relied on a "Shampoo" technique to reduce the storage requirement of full-matrix preconditioning methods. Notably, their method could be further extended to the more realistic tensor case. Anil et al. (2019) presented a memory-saved version of AdaGrad, called SM3, by maintaining $k$ sets gradient accumulator. They proved the convergence guarantee of SM3 on online convex optimization and the effectiveness in experiments. Recently, Li et al. (2024) built a 4-bit Adam using quantization techniques to compress the first and second moment estimators in Adam, also reducing memory usage.

In summary, many existing optimizers, particularly adaptive methods like AdaGrad and Adam, face memory overhead. In response, the discussed works have designed memory-efficient optimizers that aim to achieve comparable performance to these existing methods while achieving memory benefits.

## 3 PROBLEM SETUP

To start with, we introduce some necessary notations.

**Notations.** The index set $[n]$ denotes $\{1, 2, \cdots, n\}$. $\|\cdot\|_F$ and $\|\cdot\|_\infty$ denote the Frobenius norm and $l_\infty$-norm respectively. $a \lesssim \mathcal{O}(b)$ denotes $a \leq C_0 b$ for some positive constant $C_0$. For any two matrices $\boldsymbol{X} = (x_{ij})_{ij}, \boldsymbol{Y} = (y_{ij})_{ij} \in \mathbb{R}^{n \times m}$, we define $\langle \boldsymbol{X}, \boldsymbol{Y} \rangle = \sum_{i=1}^n \sum_{j=1}^m x_{ij}y_{ij}$. $\boldsymbol{X} \odot \boldsymbol{Y}$, $\frac{\boldsymbol{X}}{\boldsymbol{Y}}$ and $\sqrt{\boldsymbol{X}}$ denote the coordinate-wise product, quotient, and squared root respectively. $\boldsymbol{0}_n$ and $\boldsymbol{1}_n$ denote the zero and one $n$-dimensional vector respectively, and $\boldsymbol{1}_{n \times m}$ denotes the one $n \times m$-dimensional matrix. For a positive sequence $\{\alpha_i\}_{i \geq 1}$, we define $\sum_{i=a}^b \alpha_i = 0$ and $\prod_{i=a}^b \alpha_i = 1$ if $a > b$. The operator $\mathrm{RMS}(\cdot)$ denotes

$$\mathrm{RMS}(\boldsymbol{X}) = \sqrt{\frac{1}{mn} \sum_{i=1}^n \sum_{j=1}^m x_{ij}^2}.$$

We consider unconstrained stochastic optimization (1) over $\mathbb{R}^{n \times m}$ with the Frobenius norm. The objective function $f : \mathbb{R}^{n \times m} \to \mathbb{R}$ is differentiable. Given an $n \times m$ matrix $\boldsymbol{X}$, we assume a gradient oracle that returns a random matrix $g(\boldsymbol{X}, \boldsymbol{Z}) \in \mathbb{R}^{n \times m}$ dependent on the random sample $\boldsymbol{Z}$. The gradient of $f$ at $\boldsymbol{X}$ is denoted by $\nabla f(\boldsymbol{X}) \in \mathbb{R}^{n \times m}$.

**Assumptions.** We make the following standard assumptions throughout the paper.

- **(A1)** $L$-smoothness: For any $\boldsymbol{X}, \boldsymbol{Y} \in \mathbb{R}^{n \times m}$, $\|\nabla f(\boldsymbol{Y}) - \nabla f(\boldsymbol{X})\|_F \leq L\|\boldsymbol{Y} - \boldsymbol{X}\|_F$;
- **(A2)** Bounded below: There exists $f^* > -\infty$ such that $f(\boldsymbol{X}) \geq f^*, \forall \boldsymbol{X} \in \mathbb{R}^{n \times m}$;
- **(A3)** Unbiased estimator: The gradient oracle provides an unbiased estimator of $\nabla f(\boldsymbol{X})$, i.e., $\mathbb{E}[g(\boldsymbol{X}, \boldsymbol{Z}) \mid \boldsymbol{X}] = \nabla f(\boldsymbol{X}), \forall \boldsymbol{X} \in \mathbb{R}^{n \times m}$;
- **(A4)** Almost surely bounded stochastic gradient: for any $\boldsymbol{X} \in \mathbb{R}^{n \times m}$, $\|g(\boldsymbol{X}, \boldsymbol{Z})\|_F \leq G$, a.s..

Combining with (A3) and (A4), it's easy to verify that $\|\nabla f(\boldsymbol{X})\| \leq G, \forall \boldsymbol{X} \in \mathbb{R}^{n \times m}$. Assumptions (A1)-(A3) are standard in the non-convex smooth convergence analysis. Although Assumption (A4) is a bit strong, it's still commonly used to derive the high probability convergence bound, see e.g., (Ward et al., 2020; Kavis et al., 2022), which is a stronger result than an expected convergence.It's also commonly appeared in several early convergence results for adaptive methods, e.g., (Kingma & Ba, 2015; Reddi et al., 2018; Zaheer et al., 2018; Défossez et al., 2022). We note that our analysis can be extended to the sub-Gaussian noise case, which is commonly used for analyzing adaptive methods, e.g., (Li & Orabona, 2020; Liu et al., 2023). We will discuss this in detail in Appendix B.4.

## 4 A REVIEW OF ADAFACTOR

In this section, we briefly introduce Adafactor and highlight its major differences from Adam. The pseudocode for Adafactor is presented in Algorithm 1.

---

**Algorithm 1** Adafactor

**Input:** Initialization point $\boldsymbol{X}_1 \in \mathbb{R}^{n \times m}$, $\boldsymbol{R}_0 = \boldsymbol{0}_m$, $\boldsymbol{C}_0 = \boldsymbol{0}_n^\top$, relative step-sizes $\{\rho_k\}_{k \geq 1}$, decay rate $\{\beta_{2,k}\}_{k \geq 1} \in [0, 1)$, regularization constants $\epsilon_1, \epsilon_2 > 0$, clipping threshold $d$.
**for** $k = 1, \cdots, T$ **do**
  $\boldsymbol{G}_k = g(\boldsymbol{X}_k, \boldsymbol{Z}_k)$;
  $\boldsymbol{R}_k = \beta_{2,k} \boldsymbol{R}_{k-1} + (1 - \beta_{2,k})(\boldsymbol{G}_k \odot \boldsymbol{G}_k + \epsilon_1 \boldsymbol{1}_n \boldsymbol{1}_m^\top) \boldsymbol{1}_m$;
  $\boldsymbol{C}_k = \beta_{2,k} \boldsymbol{C}_{k-1} + (1 - \beta_{2,k}) \boldsymbol{1}_n^\top (\boldsymbol{G}_k \odot \boldsymbol{G}_k + \epsilon_1 \boldsymbol{1}_n \boldsymbol{1}_m^\top)$;
  $\boldsymbol{W}_k = (\boldsymbol{R}_k \boldsymbol{C}_k)/\boldsymbol{1}_n^\top \boldsymbol{R}_k$;
  $\boldsymbol{U}_k = \boldsymbol{G}_k/\sqrt{\boldsymbol{W}_k}$;
  $\eta_k = \max\{\epsilon_2, \mathrm{RMS}(\boldsymbol{X}_k)\}\rho_k / \max\{1, \mathrm{RMS}(\boldsymbol{U}_k)/d\}$;
  $\boldsymbol{X}_{k+1} = \boldsymbol{X}_k - \eta_k \cdot \boldsymbol{G}_k/\sqrt{\boldsymbol{W}_k}$;
**end for**

---

**Matrix factorization.** Adafactor could be served as a saved-memory version of Adam. Throughout the training process, Adam maintains two $n \times m$ matrices $\boldsymbol{M}_k$ and $\boldsymbol{V}_k$ using exponential moving average update,

$$\boldsymbol{M}_k = \beta_{1,k} \boldsymbol{M}_{k-1} + (1 - \beta_{1,k})\boldsymbol{G}_k, \quad \boldsymbol{V}_k = \beta_{2,k} \boldsymbol{V}_{k-1} + (1 - \beta_{2,k})\boldsymbol{G}_k \odot \boldsymbol{G}_k, \quad (2)$$

where $\beta_{1,k}, \beta_{2,k} \in (0, 1)$, thereby tripling the memory usage. The innovation in Adafactor lies in its method of approximating $\boldsymbol{V}_k$ by factoring it into two rank-1 matrices, specifically the row sums and column sums of $\boldsymbol{V}_k$, thus sufficiently reducing the memory from $2mn$ to $m + n$. Although this factorization sacrifices some information about the squared gradients, Adafactor still delivers performance comparable to Adam in many real application tasks.

**Increasing decay rate.** In Adam, corrective terms are introduced into $\boldsymbol{M}_k$ and $\boldsymbol{V}_k$, resulting in two increasing-to-one decay rates. Theoretically, it has been demonstrated that a value close to one for $\beta_{2,k}$ would ensure the convergence, e.g., (Défossez et al., 2022; Zou et al., 2019; Zhang et al., 2022). Inspired by this observation, Adafactor used an increasing second-moment decay rate $\beta_{2,k} = 1 - 1/k^c, c > 0$, and the empirical default setting is $c = 0.8$. As pointed out by Shazeer & Stern (2018), this setting allows for enjoying the stability of a low $\beta_{2,k}$ at the early stage of training and the insurance of convergence from a high $\beta_{2,k}$ as the run progresses. Moreover, it also leverages the bias correction.

**Update clipping.** Adafactor modifies the update process by discarding the first-order moment $\boldsymbol{M}_k$ and instead applies an update clipping technique inside the step-size $\eta_k$. This involves dividing the root-mean-square of the update $\boldsymbol{U}_k$, denoted as $\mathrm{RMS}(\boldsymbol{U}_k)$, when it exceeds a threshold $d$. This mechanism helps to calibrate the second moment estimator $\boldsymbol{W}_k$ when it's larger-than-desired $\boldsymbol{G}_k \odot \boldsymbol{G}_k$. Empirical findings in (Shazeer & Stern, 2018) indicated that implementing update clipping leads to significant performance improvements when the warm-up technique is not used.

**Relative step-sizes.** Adafactor incorporates a step-size proportional to scale of $\boldsymbol{X}_k$, denoted by $\mathrm{RMS}(\boldsymbol{X}_k)$, which is shown in experiments more resilient to the more naive parameter initialization and scaling schemes (Shazeer & Stern, 2018).

## 5 CONVERGENCE RESULT FOR FULL-BATCH ADAFACTOR

We first provide the convergence bound for the full-batch Adafactor. At each iteration, full-batch Adafactor obtains the gradient $\nabla f(\boldsymbol{X}_k)$ and then updates $\boldsymbol{R}_k, \boldsymbol{C}_k$ using $\nabla f(\boldsymbol{X}_k)$ instead of $\boldsymbol{G}_k$ in Algorithm 1.

**Theorem 5.1.** *Let $\{\boldsymbol{X}_k\}_{k\geq 1}$ be generated by Algorithm 1 with $g(\boldsymbol{X}_k, \boldsymbol{Z}_k) = \nabla f(\boldsymbol{X}_k), \forall k \geq 1$. If Assumptions (A1) and (A2) hold, $\|\nabla f(\boldsymbol{X}_k)\|_F \leq G, \forall k \geq 1, \beta_{2,1} = 1/2, \rho_1 = \rho_0$ and*

$$\rho_k = \rho_0, \quad 0 < \beta_{2,k} < 1, \quad \forall k \geq 2, \tag{3}$$

*for some positive constant $\rho_0$, then for any $T \geq 1$,*

$$\min_{k\in[T]} \|\nabla f(\boldsymbol{X}_k)\|_F^2 \lesssim \mathcal{O}\left(\frac{\log T}{T}\right).$$

*When setting $\rho_k = \rho_0/\sqrt{k}, k \geq 1$, for any $T \geq 1$,*

$$\min_{k\in[T]} \|\nabla f(\boldsymbol{X}_k)\|_F^2 \lesssim \mathcal{O}\left(\frac{\log T}{\sqrt{T}}\right).$$

The result indicates that full-batch Adafactor could find a stationary point at a rate of $\mathcal{O}(\log T/T)$ under the non-convex smooth case, corresponding to the rate for gradient descent (Bottou et al., 2018) and full-batch Adam (Shi et al., 2020). We note that the time-decreasing step-size only leads to a sub-optimal rate in our framework. The hyper-parameter setting in (3) only requires $\beta_{2,k} \in (0,1)$, denoting a much wider range including the default one which requires $\beta_{2,k} = 1 - 1/k^{0.8}$. The detailed version for the above result can be found in Theorem A.1 from the appendix.

## 6 STOCHASTIC ADAFACTOR WITHOUT UPDATE CLIPPING

In the stochastic case, we start from the simple scenario where

$$\eta_k = \max\{\epsilon_2, \text{RMS}(\boldsymbol{X}_k)\}\rho_k \tag{4}$$

dropping the update clipping $1/\max\{1, \text{RMS}(\boldsymbol{U}_k)/d\}$. The main reasons are as follows.

- As pointed out in the experiments from (Shazeer & Stern, 2018), Adafactor's performance shows little difference with and without update clipping when implementing learning rate warm-up. Since the warm-up technique is a popular method in deep learning (Zhao et al., 2023), it's reasonable to drop the update clipping.

- In stochastic Adafactor, the correlation between $\boldsymbol{G}_k$ and $\eta_k$ would be more complex if the update clipping is involved. The proof would be simpler when dropping the update clipping, which could help to better understand the analysis for Adafactor.

Based on these reasons, we assume that the warm-up technique is implemented and drop the update clipping. In addition, we focus on the stage when the warm-up is finished, which allows us to focus on the stage that leads to the final output. Despite these reasons, we also believe that investigating the warm-up stage could be quite an interesting topic for future work. We now present the probabilistic convergence bound for Adafactor without update clipping as follows.

**Theorem 6.1.** *Let $\{\boldsymbol{X}_k\}_{k\geq 1}$ be generated by Algorithm 1 without update clipping where $\eta_k$ is given by (4) for each $k \geq 1$. If Assumptions (A1)-(A4) hold, and*

$$\beta_{2,1} = 1/2, \quad \rho_1 = \rho_0,$$
$$\beta_{2,k} = 1 - 1/k^c, \quad \rho_k = \rho_0/\sqrt{k}, \quad \forall k \geq 2, \tag{5}$$

*for some constants $1/2 \leq c \leq 1, \rho_0 > 0$, then for any $T \geq 1, \delta \in (0,1)$, with probability at least $1 - \delta$,*

$$\min_{k\in[T]} \|\nabla f(\boldsymbol{X}_k)\|_F^2 \lesssim \mathcal{O}\left(\frac{1}{T^{c-1/2}} \log\left(\frac{T}{\delta}\right)\right).$$

The detailed version of the above results can be found in Theorem B.1 from the appendix. We will make a detailed discussion on the convergence bound as well as some hyper-parameter dependencies in the next section.

## 6.1 DISCUSSION OF THE HYPER-PARAMETER DEPENDENCY.

In this section, we discuss the dependency of several important hyper-parameters in Theorem 6.1 and the detailed version in Theorem B.1 in the appendix. It's worthy to mention that the dominated order in our convergence bound is determined by the total iteration number $T$, whereas other hyper-parameters could be regarded as constants. However, we hope to improve the dependency of these hyper-parameters as much as possible to make the convergence bound tight.

**Discussion of $c$ and the optimal rate.** Theorem 6.1 reveals that when $c = 1, \beta_{2,k} = 1 - 1/k$ and $\rho_k = \rho_0/\sqrt{k}$, the convergence rate attains the optimal rate matching the lower bound. The result then complements the empirical results that the information lost in Adafactor does not essentially harm the convergence speed and Adafactor could still achieve comparable performance to Adam.

In addition, when $c$ increases from $1/2$ to $1$, the convergence rate improves, which could also be seen roughly in the experiment (see Figure 1). This phenomenon somehow explains that a small decay rate $\beta_{2,k}$ ($c$ is low) may harm the convergence speed, as $\beta_{2,k}$ should be closed enough to 1 to ensure convergence. This phenomenon is both verified by convergence bounds for Adam in e.g., (Zou et al., 2019; Défossez et al., 2022; Zhang et al., 2022; Wang et al., 2023) and negative results where a constant $\beta_2$ is not guaranteed to converge in e.g., (Reddi et al., 2018; Zhang et al., 2022).

**Dependency to $mn$.** It's clear to see that the convergence bounds in Theorem A.1 and Theorem B.1 are free of the curse of the dimension factor $mn$ as $mn$ only appears on the denominator in each coefficient. We think that solving the curse of dimension is vital since the applied range for Adafactor includes many deep learning tasks where $mn$ are comparable large to $T$.

**Dependency to $\epsilon_1, \epsilon_2$.** The convergence bounds in Theorem 6.1 is of order $\mathcal{O}(\epsilon_1^{-1} \log(1/\epsilon_1))$ on $\epsilon_1$.[2] Although the polynomial dependency to $\epsilon_1$ is a bit worse since $\epsilon_1$ usually takes a small value in experiments, e.g., $10^{-30}$ in the default setup, it's still common in some theoretical convergence results, e.g., (Zaheer et al., 2018; Li et al., 2023). We also perform some experiments to show that a relatively large $\epsilon_1$, roughly $10^{-5}$, makes no observable effect on the convergence speed (see Figure 4 in Appendix E). Thereby, $\epsilon_1$ could be regarded as a constant in comparison to $T$ and the influence brought by $1/\epsilon_1$ could be somehow acceptable.

Since the default value of $\epsilon_2$ is $10^{-3}$ in experiments, the dependency $\mathcal{O}(1/\epsilon_2)$ on $\epsilon_2$ shows little effect on convergence bounds given the sufficiently large $T$.

**Dependency on the scale of parameters.** The convergence bounds in Theorem B.1 contain a $\mathcal{O}(\Theta_{\max})$ factor where $\Theta_{\max}$ denotes the maximum values of $\|X_k\|_\infty, \forall k \geq 1$. However, the dependence on $\Theta_{\max}$ is not fundamental, as it arises from the relative step-size $\max\{\epsilon_2, \text{RMS}(X_k)\}$, which could be dropped by removing the relative step-size as done in Adam.

## 7 CONVERGENCE OF ADAFACTOR WITH UPDATE CLIPPING

In this section, we slightly change the update clipping threshold $d$ in Algorithm 1 to a time-varying threshold $d_k$. The step-size $\eta_k$ then becomes

$$\eta_k = \frac{\max\{\epsilon_2, \text{RMS}(X_k)\}\rho_k}{\max\{1, \text{RMS}(U_k)/d_k\}}. \tag{6}$$

The update-clipping in Adafactor differs from the standard clipping mechanism with the form $1/\max\{1, \lambda/\|G_k\|_F\}$, bringing some more essential challenges for analyzing. In what follows, we demonstrate that incorporating such clipping can still ensure convergence for Adafactor under bounded stochastic gradient assumption.

**Theorem 7.1.** *Let $\{X_k\}_{k\geq 1}$ be generated by Algorithm 1 with $\eta_k$ given by (6) for each $k \geq 1$. If Assumptions (A1)-(A4) hold, and*

$$d_1 = 1, \quad \beta_{2,1} = 1/2, \quad \rho_1 = \rho_0,$$
$$d_k = k^{\frac{c}{2(\alpha-1)}}, \quad \beta_{2,k} = 1 - 1/k^c, \quad \rho_k = \rho_0/\sqrt{k}, \quad \forall k \geq 2, \tag{7}$$

---

[2]The detailed discussion could be found in (41) and (42) in Appendix B.

*for some constants $\alpha > 1, 1/2 \leq c \leq 1, \rho_0 > 0$, then for any $T \geq 1, \delta \in (0,1)$, with probability at least $1 - \delta$,*

$$\min_{k \in [T]} \|\nabla f(\boldsymbol{X}_k)\|_F^2 \lesssim \mathcal{O}\left(\frac{1}{T^{c-1/2}} \log\left(\frac{T}{\delta}\right)\right).$$

**Discussion of Theorem 7.1.** The convergence result indicates that with a proper selection of the clipping threshold, along with the commonly used $\rho_k$ and $\beta_{2,k}$, Adafactor can find a stationary point with a rate of $\tilde{\mathcal{O}}(1/T^{c-1/2})$. The dependency on $c$ remains consistent with Theorem 6.1, achieving the optimal rate when $c = 1$. We thus conclude that Adafactor, equipped with matrix factorization to reduce the memory of Adam and update clipping, could still obtain a convergence rate as fast as Adam in theory. In addition, the convergence bound can still avoid the curse of dimension, which is shown in the detailed version Theorem D.1 from the appendix.

The additional hyper-parameter $\alpha$ primarily influences the dependency on $\epsilon_1$, specifically as $\mathcal{O}\left(\epsilon_1^{-\alpha} \log(1/\epsilon_1)\right)$. Thus, our convergence bound may deteriorate as $\alpha$ increases. This dependency could be potentially improved to $\mathcal{O}\left(\epsilon_1^{-1} \log(1/\epsilon_1)\right)$ when $mn$ is comparable to $1/\epsilon_1$, which is practical in large-size models.[3] In our experiments, we found that suitably small values, such as $\alpha = 4, 6, 7, 8$ can lead to convergence speed and training stability comparable to the default one (see Figure 5 and 6). This finding suggests that our new threshold setting plays a similar role in enhancing training stability as the default one, which is also the main motivation for update clipping. Since $\epsilon_1$ can be set to a relatively large value, e.g., $10^{-3}$, a dependency like $\mathcal{O}(\epsilon_1^{-4} \log(1/\epsilon_1))$ is somewhat acceptable for sufficiently large $T$.

The time-increasing $d_k$ provides the following intuition: As shown in (Shazeer & Stern, 2018, Figure 1), during the early stages of training, a high decay rate $\beta_{2,k}$ can cause larger-than-desired updates and training instability. Therefore, we set a low threshold $d_k$ to ensure that the update clipping mechanism effectively calibrates these larger-than-desired updates. As training progresses, the sequences and updates become more stable. Consequently, there is less need for update clipping, corresponding to a relatively large $d_k$.

## 8 SUMMARY OF PROOF CHALLENGES AND TECHNIQUES

In this section, we will summarize the main proof challenges brought by Adafactor, which are essentially different from other adaptive methods particularly Adam due to the unique matrix factorization and update clipping. We also present our proof techniques including a proof sketch for Theorem 6.1 in the solution part. The proof for other main results shares many similarities with this proof sketch.

We begin by the descent lemma of the smoothness and using the updated rule in Algorithm 1,

$$f(\boldsymbol{X}_{k+1}) \leq f(\boldsymbol{X}_k) \underbrace{-\eta_k \left\langle \nabla f(\boldsymbol{X}_k), \frac{\boldsymbol{G}_k}{\sqrt{\boldsymbol{W}_k}} \right\rangle}_{(\textbf{I})} + \underbrace{\frac{L\eta_k^2}{2} \left\| \frac{\boldsymbol{G}_k}{\sqrt{\boldsymbol{W}_k}} \right\|_F^2}_{(\textbf{II})}, \quad \forall k \geq 1, \tag{8}$$

**Challenge I. A new type of adaptive step-size (no update clipping).** We first consider the step-size excluding the update clipping. The analysis of Adafactor presents two unique challenges, both arising from its adaptive step-size involving a distinctive matrix factorization:

- Addressing the entanglement of the stochastic gradient $\boldsymbol{G}_k$, and the adaptive step-size matrix $\boldsymbol{W}_k$ that appears in component **(I)** in (8).

- Controlling the summation of the second-order term **(II)**.

A key difficulty in analyzing adaptive methods lies in computing the conditional expectation of **(I)** due to the correlation of $\boldsymbol{G}_k$ and $\boldsymbol{W}_k$. To overcome this, existing analyses typically introduce a proxy step-size matrix $\boldsymbol{A}_k$ that is conditional independent of $\boldsymbol{G}_k$. This approach is applied in works such as

---

[3]The detailed calculation could be found in (96) from the appendix.

(Ward et al., 2020; Défossez et al., 2022) for AdaGrad and (Wang et al., 2023; Hong & Lin, 2024) for Adam. Introducing $\boldsymbol{A}_k$ into (8) and summing up both sides over $k \in [t]$,

$$f(\boldsymbol{X}_{t+1}) \leq f(\boldsymbol{X}_1) - \sum_{k=1}^{t} \eta_k \left\| \frac{\bar{\boldsymbol{G}}_k}{\sqrt[4]{\boldsymbol{A}_k}} \right\|_F^2 - \underbrace{\sum_{k=1}^{t} \eta_k \left\langle \bar{\boldsymbol{G}}_k, \frac{\boldsymbol{G}_k - \bar{\boldsymbol{G}}_k}{\sqrt{\boldsymbol{A}_k}} \right\rangle}_{(\mathbf{A})}$$

$$+ \underbrace{\sum_{k=1}^{t} \eta_k \left\langle \bar{\boldsymbol{G}}_k, \boldsymbol{G}_k \odot \left( \frac{1}{\sqrt{\boldsymbol{A}_k}} - \frac{1}{\sqrt{\boldsymbol{W}_k}} \right) \right\rangle}_{(\mathbf{B})} + \underbrace{\sum_{k=1}^{t} \frac{L\eta_k^2}{2} \left\| \frac{\boldsymbol{G}_k}{\sqrt{\boldsymbol{W}_k}} \right\|_F^2}_{(\mathbf{C})}.$$

Note that (**A**) is a summation of a martingale difference sequence, which could be estimated through a concentration inequality. The primary challenge, however, comes from estimating the additional error (**B**). For Adam, the updated rule in (2) and AdaGrad, the updates $\boldsymbol{V}_k = \boldsymbol{V}_{k-1} + \boldsymbol{G}_k \odot \boldsymbol{G}_k$ ensures that $\boldsymbol{V}_k$ and $\boldsymbol{V}_{k-1}$ share a linear relation. Most existing works rely on this linear relation to design suitable proxy step-sizes, thereby tightly controlling (**B**) (see e.g., (Défossez et al., 2022, Lemma 5.1) and (Wang et al., 2023, Lemma 7)). However, the step-size matrix $\boldsymbol{W}_k$ in Adafactor does not exhibit a linear relationship with $\boldsymbol{W}_{k-1}$. Specifically, we let $\boldsymbol{G}_{k,\epsilon_1} = \boldsymbol{G}_k \odot \boldsymbol{G}_k + \epsilon_1$ and derive

$$\boldsymbol{W}_k = \frac{\left( \beta_{2,k} \boldsymbol{R}_{k-1} + (1 - \beta_{2,k}) \boldsymbol{R}_{\boldsymbol{G}_{k,\epsilon_1}} \right) \odot \left( \beta_{2,k} \boldsymbol{C}_{k-1} + (1 - \beta_{2,k}) \boldsymbol{C}_{\boldsymbol{G}_{k,\epsilon_1}} \right)}{\beta_{2,k} S_{k-1} + (1 - \beta_{2,k}) S_{\boldsymbol{G}_{k,\epsilon_1}}},$$

where $\boldsymbol{R}_{\boldsymbol{G}_{k,\epsilon_1}} = \boldsymbol{G}_{k,\epsilon_1} \mathbf{1}_m, \boldsymbol{C}_{\boldsymbol{G}_{k,\epsilon_1}} = \mathbf{1}_n^\top \boldsymbol{G}_{k,\epsilon_1}$ and $S_k, S_{\boldsymbol{G}_{k,\epsilon_1}}$ are the coordinate sum of $\boldsymbol{V}_k, \boldsymbol{G}_{k,\epsilon_1}$. The absence of a linear relation between $\boldsymbol{W}_k$ and $\boldsymbol{W}_{k-1}$ suggests that **B** may be unbounded using existing proxy step-sizes.

Existing results, such as (Ward et al., 2020, Lemma 3.2) for AdaGrad or (Défossez et al., 2022, Lemma 5.2) for Adam, show that the summation of the second-order term is restricted by logarithm order of $T$. However, these results could not be directly applied to Adafactor due to the rather different adaptive step-size and the time-varying $\beta_{2,k}$.

**Solution.** The solution part also serves as a proof sketch of Theorem 6.1. Motivated by the existing construction, we design a new proxy step-size matrix $\boldsymbol{A}_k$ as follows:

$$\boldsymbol{A}_k = \frac{(\beta_{2,k} \boldsymbol{R}_{k-1} + (1 - \beta_{2,k}) \mathcal{G}_1) \odot (\beta_{2,k} \boldsymbol{C}_{k-1} + (1 - \beta_{2,k}) \mathcal{G}_2)}{\beta_{2,k} S_{k-1} + (1 - \beta_{2,k}) \mathcal{G}},$$

where $\mathcal{G}_1, \mathcal{G}_2, \mathcal{G}$ are constants related to $G^4$. We note that $\boldsymbol{A}_k$ is conditional independent with the noise $\boldsymbol{G}_k - \bar{\boldsymbol{G}}_k$. Note that we omit update clipping in Theorem 6.1 and **A** is now a summation of the martingale difference sequence. Hence, we could use the concentration inequality to derive that $\mathbf{A} \lesssim \mathcal{O}\left( G^2 \log(T/\delta)/\epsilon_1 \right)$ with the detail in Lemma B.6. More importantly, the construction of $\boldsymbol{A}_k$ is delicate since we are able to control the relative distance (detailed in Lemma B.7) as

$$\frac{\left| w_{ij}^{(k)} - a_{ij}^{(k)} \right|}{\sqrt{a_{ij}^{(k)}}} \lesssim \mathcal{O}\left( G\sqrt{1 - \beta_{2,k}} \right), \forall k \geq 1, i \in [n], j \in [m].$$

Relying on this bound, we could control the error term (**B**) as

$$(\mathbf{B}) \lesssim \frac{1}{4} \sum_{k=1}^{t} \eta_k \left\| \frac{\bar{\boldsymbol{G}}_k}{\sqrt[4]{\boldsymbol{A}_k}} \right\|_F^2 + \mathcal{O}\left( G \sum_{k=1}^{t} (1 - \beta_{2,k}) \left\| \frac{\boldsymbol{G}_k}{\sqrt{\boldsymbol{W}_k}} \right\|_F^2 \right). \tag{9}$$

The remained thing is to control the second-order summation that emerged both in (**C**) and (9). We begin by analyzing the ratio of the second-order term for Adafactor and Adam. Then, we extend an inequality for Adam with a constant decay rate (Défossez et al., 2022, Lemma 5.2) to a time-varying setup. These results are summarized as (see the details in Lemma B.4 and B.5),

$$\left\| \frac{\boldsymbol{G}_k}{\sqrt{\boldsymbol{W}_k}} \right\|_F^2 \lesssim \mathcal{O}\left( \frac{G^2}{\epsilon_1} \left\| \frac{\boldsymbol{G}_k}{\sqrt{\boldsymbol{V}_k}} \right\|_F^2 \right), \sum_{k=1}^{t} (1 - \beta_{2,k}) \left\| \frac{\boldsymbol{G}_k}{\sqrt{\boldsymbol{V}_k}} \right\|_F^2 \lesssim \mathcal{O}\left( \log\left( \frac{G^2}{\epsilon_1} + \sum_{k=1}^{t} (1 - \beta_{2,k}) \right) \right).$$

---

[4]The detailed expression is given in (14).

These results help to derive that

$$(\mathbf{B}) + (\mathbf{C}) \lesssim \frac{1}{4} \sum_{k=1}^{t} \eta_k \left\| \frac{\bar{\boldsymbol{G}}_k}{\sqrt[4]{\boldsymbol{A}_k}} \right\|_F^2 + \mathcal{O}\left( \frac{G^3}{\epsilon_1} \left( \log\left( \frac{G^2}{\epsilon_1} \right) + \sum_{k=1}^{t} (1 - \beta_{2,k}) \right) \right).$$

Combining with the bounds for **(A)**,**(B)**,**(C)** and using $\beta_{2,k} = 1 - 1/k^c$, it holds that with probability at least $1 - \delta$,

$$\frac{1}{2} \sum_{k=1}^{t} \eta_k \left\| \frac{\bar{\boldsymbol{G}}_k}{\sqrt[4]{\boldsymbol{A}_k}} \right\|_F^2 \lesssim \mathcal{O}\left( \frac{G^3}{\epsilon_1} \log\left( \frac{GT}{\delta\epsilon_1} \right) + \sum_{k=1}^{t} \frac{1}{k^c} \right).$$

Finally, by upper bounding $\left\| \sqrt[4]{\boldsymbol{A}_k} \right\|_F$ with $G$ (see Lemma B.3), we can derive the desired result.

**Challenge II. Additional update clipping in the adaptive step-size.** We note that the solution to the first challenge only considers the matrix factorization but omits the update clipping. However, incorporating update clipping introduces an even more complex adaptive step-size $\eta_k$ as in Algorithm 1, and the conditional expectation of **(I)** is even harder to compute. To our knowledge, this structure causes all existing constructions of proxy step-size ineffective. We will face this challenge in the proof of Theorem 7.1.

**Solution.** We first rewrite the updated rule as

$$\boldsymbol{X}_{k+1} = \boldsymbol{X}_k - \hat{\rho}_k \frac{\tilde{\boldsymbol{G}}_k}{\sqrt{\boldsymbol{W}_k}}, \tilde{\boldsymbol{G}}_k = \frac{\boldsymbol{G}_k}{\max\{1, \text{RMS}(\boldsymbol{U}_k)/d_k\}}, \hat{\rho}_k = \max\{\epsilon_2, \text{RMS}(\boldsymbol{X}_k)\}\rho_k.$$

The first-order term in the descent lemma then become $(\tilde{\mathbf{I}}) = \sum_{k=1}^{t} -\hat{\rho}_k \left\langle \bar{\boldsymbol{G}}_k, \tilde{\boldsymbol{G}}_k/\sqrt{\boldsymbol{W}_k} \right\rangle$. Inspired by a standard way in the analysis of SGD with clipping, we provide a decomposition for $(\tilde{\mathbf{I}})$,

$$(\tilde{\mathbf{I}}) = -\sum_{k=1}^{t} \hat{\rho}_k \left\| \frac{\bar{\boldsymbol{G}}_k}{\sqrt[4]{\boldsymbol{A}_k}} \right\|_F^2 + \underbrace{\sum_{k=1}^{t} \hat{\rho}_k \left\langle \bar{\boldsymbol{G}}_k, \left( \frac{1}{\sqrt{\boldsymbol{A}_k}} - \frac{1}{\sqrt{\boldsymbol{W}_k}} \right) \odot \tilde{\boldsymbol{G}}_k \right\rangle}_{(\mathbf{1})}$$

$$\underbrace{-\sum_{k=1}^{t} \hat{\rho}_k \left\langle \bar{\boldsymbol{G}}_k, \frac{\tilde{\boldsymbol{G}}_k}{\sqrt{\boldsymbol{A}_k}} - \mathbb{E}_{\boldsymbol{Z}_k}\left[ \frac{\tilde{\boldsymbol{G}}_k}{\sqrt{\boldsymbol{A}_k}} \right] \right\rangle}_{(\mathbf{2})} + \underbrace{\sum_{k=1}^{t} \hat{\rho}_k \left\langle \bar{\boldsymbol{G}}_k, \frac{\bar{\boldsymbol{G}}_k}{\sqrt{\boldsymbol{A}_k}} - \mathbb{E}_{\boldsymbol{Z}_k}\left[ \frac{\tilde{\boldsymbol{G}}_k}{\sqrt{\boldsymbol{A}_k}} \right] \right\rangle}_{(\mathbf{3})}.$$

Here, **(2)** is a summation of a martingale difference sequence and **(1)** is an error term that can be estimated similarly to **(B)** in (9). The critical step is to handle the additional error term **(3)** using the maximum operator inside the update clipping (detailed in (109) and (110)),

$$(\mathbf{3}) \lesssim \mathcal{O}\left( \frac{G^{1+\alpha} \left( G^2 + \sqrt{\epsilon_1} \right)^{\alpha}}{\epsilon_1^{\alpha}} \sum_{k=1}^{t} \frac{1}{d_k^{\alpha-1}\sqrt{k}} \right).$$

To ensure that this error term remains controlled by a logarithm order of $t$, we should further require $d_k = k^{\frac{c}{2(\alpha-1)}}$.

**Challenge III. Lower bound first-order term (full-batch case).** A central problem in full-batch case is to lower bound **(I)** in (15). Existing results on Adam, e.g., (De et al., 2018) obtain that $\|\boldsymbol{V}_k\|_{\infty} \leq G^2$ based on exponential moving average property, thus lower bounding **(I)**. However, Adafactor does not enjoy such a property.

**Solution.** We first separate $[t]$ into two index set

$$E_1 = \left\{ k \in [t] \mid \|\boldsymbol{U}_k\|_F \geq d\sqrt{mn} \right\}, \quad E_2 = \left\{ k \in [t] \mid \|\boldsymbol{U}_k\|_F \leq d\sqrt{mn} \right\}.$$

Through Lemma A.3, we show that $\|\boldsymbol{W}_k\|_{\infty} \lesssim \mathcal{O}(G^2 + \epsilon_1), \|\boldsymbol{U}_k\|_F \lesssim \mathcal{O}(G^2/\epsilon_1)$. Then, for some constant $c_0 > 0$, **(I)** is lower bounded by

$$(\mathbf{I}) \gtrsim \mathcal{O}\left( \sum_{k \in E_1} \frac{\rho_k \|\bar{\boldsymbol{G}}_k\|_F^2}{\|\boldsymbol{U}_k\|_F \sqrt{\|\boldsymbol{W}_k\|_{\infty}}} + \sum_{k \in E_2} \frac{\rho_k \|\bar{\boldsymbol{G}}_k\|_F^2}{\sqrt{\|\boldsymbol{W}_k\|_{\infty}}} \right) \gtrsim \mathcal{O}\left( \frac{\min\{c_0, \epsilon_1/G^2\}}{G + \sqrt{\epsilon_1}} \sum_{k=1}^{t} \rho_k \|\bar{\boldsymbol{G}}_k\|_F^2 \right).$$

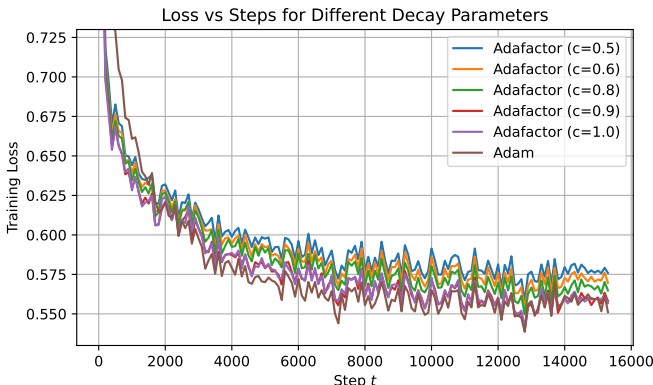

Figure 1: Training loss curve using BERT-Base model on GLUE/MNLI dataset. Adafactor: $\rho_0 = 7 \times 10^{-3}$, batch-size $= 128$. Adam: $\rho_0 = 3.5 \times 10^{-5}, \beta_1 = 0.9, \beta_2 = 0.999$, batch-size $= 128$.

## 9    EXPERIMENTS

In this section, we will report our experimental results based on our convergence results. We will mainly provide the following three experiments (the last two are included in Appendix E due to the page limitation):

- We will show the training/testing performance of Adafactor (no update clipping) under different decay rate parameters $c$ on CV and NLP tasks.
- We evaluate the sensitivity of Adafactor to different values of $\epsilon_1$, particularly showing that a relatively large $\epsilon_1$ does not significantly impact the convergence speed.
- We assess the training performance of Adafactor with a time-increasing $d_k$ setting, as described in Theorem 7.1, and compare it to the default constant setting.

We train BERT-Base model using Adafactor (no update clipping) with decay rate $c$ ranging from $0.5$ to $1.0$, while keeping other hyper-parameters the same. Each experiment is run with 4 epochs, and we plot the training loss curve in Figure 1. We also train the model with Adam as the baseline. The result indicates that convergence rates for Adafactor and Adam are comparable. In addition, the convergence rate for Adafactor grows fast as $c$ increases from $0.5$ to $1.0$, roughly aligning with Theorem 6.1.

The second experiment (Figure 4) shows that Adafactor is not sensitive to the choice of $\epsilon_1$, and a relatively large $\epsilon_1$, such as $10^{-3}$ can still lead to convergence, making the polynomial dependency $\mathcal{O}(1/\epsilon_1)$ in convergence bounds acceptable. The third experiment (Figure 5 and 6) indicates that, for $\alpha = 4, 6, 7, 8$, Adafactor achieves comparable convergence speed compared to the default threshold. All the detailed results could be found in Appendix E.

## 10    CONCLUSIONS

In this paper, we investigate the convergence behavior of Adafactor on non-convex smooth landscapes with bounded stochastic gradients. Our theoretical results complement an empirical observation that Adafactor could achieve comparable performance to Adam, despite sacrificing some gradient information to reduce memory usage. We introduce a new proxy step-size to decouple the stochastic gradients from the unique adaptive step-size and update clipping. We also rely on the unique structure of proxy step-sizes and an appropriate choice of $\beta_2$ to control the additional errors.

**Limitations.**    Several limitations warrant further investigation. First, the polynomial dependency on $\epsilon_1$ in convergence bounds may be improved to a better one, such as $\log(1/\epsilon_1)$. Second, the convergence bound for stochastic vanilla Adafactor remains unknown. Third, the bounded stochastic gradient can be relaxed as it may be unpractical in LLMs (Zhang et al., 2020). Finally, it's beneficial to further support our theoretical results through experiments on large language models.

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

# A    PROOF DETAIL FOR FULL-BATCH CASE

We first provide the full-batch Adafactor as follows. The only difference to Algorithm (1) is the replacement of stochastic gradient by deterministic gradient $\nabla f(\boldsymbol{X}_k)$ at each iteration.

---

**Algorithm 2** Full-batch Adafactor

---

**Input:** Initialization point $\boldsymbol{X}_1 \in \mathbb{R}^{n \times m}$, $\boldsymbol{R}_0 = \boldsymbol{0}_n$, $\boldsymbol{C}_0 = \boldsymbol{0}_m^\top$, relative step-sizes $\{\rho_k\}_{k \geq 1}$, decay
rate $\{\beta_{2,k}\}_{k \geq 1} \in [0, 1)$, regularization constants $\epsilon_1, \epsilon_2 > 0$, clipping threshold $d$.
 **for** $k = 1, \cdots, T$ **do**
  $\bar{\boldsymbol{G}}_k = \nabla f(\boldsymbol{X}_k)$;
  $\bar{\boldsymbol{R}}_k = \beta_{2,k} \bar{\boldsymbol{R}}_{k-1} + (1 - \beta_{2,k})(\bar{\boldsymbol{G}}_k \odot \bar{\boldsymbol{G}}_k + \epsilon_1 \boldsymbol{1}_n \boldsymbol{1}_m^\top) \boldsymbol{1}_m$;
  $\bar{\boldsymbol{C}}_k = \beta_{2,k} \bar{\boldsymbol{C}}_{k-1} + (1 - \beta_{2,k}) \boldsymbol{1}_n^\top (\bar{\boldsymbol{G}}_k \odot \bar{\boldsymbol{G}}_k + \epsilon_1 \boldsymbol{1}_n \boldsymbol{1}_m^\top)$;
  $\bar{\boldsymbol{W}}_k = (\bar{\boldsymbol{R}}_k \bar{\boldsymbol{C}}_k) / \boldsymbol{1}_n^\top \bar{\boldsymbol{R}}_k$;
  $\bar{\boldsymbol{U}}_k = \bar{\boldsymbol{G}}_k / \sqrt{\bar{\boldsymbol{W}}_k}$;
  $\hat{\eta}_k = \max\{\epsilon_2, \mathrm{RMS}(\boldsymbol{X}_k)\} \rho_k / \max\{1, \mathrm{RMS}(\bar{\boldsymbol{U}}_k)/d\}$;
  $\boldsymbol{X}_{k+1} = \boldsymbol{X}_k - \hat{\eta}_k \cdot \bar{\boldsymbol{G}}_k / \sqrt{\bar{\boldsymbol{W}}_k}$;
 **end for**

---

Then, we provide the detailed version of Theorem 5.1 as follows.

**Theorem A.1.** *Let $\{\boldsymbol{X}_k\}_{k \geq 1}$ be generated by Algorithm 2. If Assumptions (A1), (A2) hold,* $\|\nabla f(\boldsymbol{X}_k)\|_F \leq G, \forall k \geq 1$ *and* $\rho_1 = \rho_0, \beta_{2,1} = 1/2$,

$$\rho_k = \rho_0, \quad 0 < \beta_{2,k} < 1, \quad \forall k \geq 2,$$

*for some positive constant $\rho_0$, then for any $T \geq 1$,*

$$\min_{k \in [T]} \|\nabla f(\boldsymbol{X}_k)\|_F^2 \leq \frac{A_0 A_1 (f(\boldsymbol{X}_1) - f^* + \Delta_0^2 \log T + \Delta_0^2)}{T}.$$

*If let $\rho_k = \rho_0/\sqrt{k}$, then for any $T \geq 1$,*

$$\min_{k \in [T]} \|\nabla f(\boldsymbol{X}_k)\|_F^2 \leq \frac{A_0 A_1 (f(\boldsymbol{X}_1) - f^* + \Delta_0^2 \log T + \Delta_0^2)}{\sqrt{T}},$$

$$\min_{k \in [T]} \|\nabla f(\boldsymbol{X}_k)\|_F^2 \leq \frac{A_0 A_1' (f(\boldsymbol{X}_1) - f^* + \tilde{\Delta}_0^2 \log T + \tilde{\Delta}_0^2)}{\sqrt{T}}, \tag{10}$$

*where we define*

$$\Theta_{\max} = \max_{k \in [T]} \|\boldsymbol{X}_k\|_\infty, \quad \mathcal{G} = G^2 + mn\epsilon_1, \tag{11}$$

*and the other constant parameters are given by*

$$\Delta_0^2 = \frac{Ld^2 mn(\epsilon_2 + \Theta_{\max})^2 \rho_0^2}{2}, \quad \tilde{\Delta}_0^2 = \frac{LG^2 \mathcal{G}(\epsilon_2 + \Theta_{\max})^2 \rho_0^2}{2mn\epsilon_1^2 (1 - \beta_{2,1})^2},$$

$$A_0 = \frac{\max\left\{1, \frac{G\sqrt{\mathcal{G}}}{d\epsilon_1 mn(1 - \beta_{2,1})}\right\}}{\rho_0 \epsilon_2}, A_1 = \sqrt[4]{G^4 + G^2(m+n)\epsilon_1 + mn\epsilon_1^2}, \tag{12}$$

$$A_1' = \sqrt{2\left(\frac{G^4}{mn\epsilon_1} + G^2 + \epsilon_1\right)}.$$

## A.1    PRELIMINARY

We first denote the auxiliary matrix $\bar{\boldsymbol{G}}_{k,\epsilon_1}^2 = \bar{\boldsymbol{G}}_k \odot \bar{\boldsymbol{G}}_k + \epsilon_1 \boldsymbol{1}_n \boldsymbol{1}_m^\top$. In addition, we define $\bar{\boldsymbol{V}}_k = \left(\bar{v}_{ij}^{(k)}\right)_{ij}$ as follows,

$$\bar{\boldsymbol{V}}_0 = \boldsymbol{0}_{n \times m}, \quad \bar{\boldsymbol{V}}_k = \beta_{2,k} \bar{\boldsymbol{V}}_{k-1} + (1 - \beta_{2,k}) \bar{\boldsymbol{G}}_{k,\epsilon_1}^2, \quad k \geq 1. \tag{13}$$

To simplify the notation, we let $\bar{\boldsymbol{G}}_k = \left( \bar{g}_{ij}^{(k)} \right)_{ij}$, $R_{\bar{\boldsymbol{V}}_k}^{(i)}, C_{\bar{\boldsymbol{V}}_k}^{(j)}$ and $S_{\bar{\boldsymbol{V}}_k}$ be the $i$-th row sum, $j$-th column sum and the coordinate sum of $\bar{\boldsymbol{V}}_k$ respectively. The same definition principal is applied to the notation $R_{\bar{\boldsymbol{G}}_{k,\epsilon_1}^2}^{(i)}$ and $C_{\bar{\boldsymbol{G}}_{k,\epsilon_1}^2}^{(j)}$. We also use $\bar{w}_{ij}^{(k)}, \bar{v}_{ij}^{(k)}, \bar{u}_{ij}^{(k)}$ to denote the coordinates of $\bar{\boldsymbol{W}}_k, \bar{\boldsymbol{V}}_k, \bar{\boldsymbol{U}}_k$ in Algorithm 2 respectively. We also define values $\mathcal{G}_1, \mathcal{G}_2, \mathcal{G}$ as follows:

$$\mathcal{G}_1 = G^2 + m\epsilon_1, \quad \mathcal{G}_2 = G^2 + n\epsilon_1, \quad \mathcal{G} = G^2 + mn\epsilon_1. \tag{14}$$

A.2    TECHNICAL LEMMAS

Following the descent lemma for a $L$-smooth objective function $f$, we derive that

$$f(\boldsymbol{Y}) \leq f(\boldsymbol{X}) + \langle \nabla f(\boldsymbol{X}), \boldsymbol{Y} - \boldsymbol{X} \rangle + \frac{L}{2}\|\boldsymbol{Y} - \boldsymbol{X}\|_F^2, \quad \forall \boldsymbol{X}, \boldsymbol{Y} \in \mathbb{R}^{n \times m}. \tag{15}$$

In the following, we will provide some necessary technical lemmas.

**Lemma A.1.** *Let $\beta_{2,k} \in (0,1)$ and $\Gamma_k$ be defined by*

$$\Gamma_0 = 0, \quad \Gamma_k = \beta_{2,k}\Gamma_{k-1} + (1 - \beta_{2,k}), \quad \forall k \geq 1.$$

*Then, $(1 - \beta_{2,1}) \leq \Gamma_k \leq 1, \forall k \geq 1$.*

*Proof.* We could prove the result by induction. Since $\Gamma_0 = 0$, it's easy to derive that $(1 - \beta_{2,1}) = \Gamma_1 \leq 1$. Suppose that for any $j \in [k-1]$, $(1 - \beta_{2,1}) \leq \Gamma_j \leq 1$. Then

$$\Gamma_k \geq \beta_{2,k}(1 - \beta_{2,1}) + (1 - \beta_{2,k}) \geq 1 - \beta_{2,1}, \quad \Gamma_k \leq \beta_{2,k} + (1 - \beta_{2,k}) \leq 1.$$

The induction is then complete. □

**Lemma A.2.** *Let $\bar{\boldsymbol{V}}_k$ be defined in (13). For any $k \geq 0$, it holds that*

$$\bar{\boldsymbol{R}}_k = \bar{\boldsymbol{V}}_k \mathbf{1}_m, \quad \bar{\boldsymbol{C}}_k = \mathbf{1}_n^\top \bar{\boldsymbol{V}}_k, \quad S_{\bar{\boldsymbol{V}}_k} = \mathbf{1}_n^\top \bar{\boldsymbol{R}}_k = \mathbf{1}_n^\top \bar{\boldsymbol{V}}_k \mathbf{1}_m.$$

*As a consequence,*

$$R_{\bar{\boldsymbol{V}}_k}^{(i)} = \beta_{2,k}R_{\bar{\boldsymbol{V}}_{k-1}}^{(i)} + (1 - \beta_{2,k})R_{\bar{\boldsymbol{G}}_{k,\epsilon_1}^2}^{(i)}, \quad C_{\bar{\boldsymbol{V}}_k}^{(j)} = \beta_{2,k}C_{\bar{\boldsymbol{V}}_{k-1}}^{(j)} + (1 - \beta_{2,k})C_{\bar{\boldsymbol{G}}_{k,\epsilon_1}^2}^{(j)}.$$

*Proof.* Note that $\bar{\boldsymbol{R}}_0 = \bar{\boldsymbol{V}}_0 \mathbf{1}_m = \mathbf{0}_n$ and $\bar{\boldsymbol{C}}_0 = \mathbf{1}_n^\top \bar{\boldsymbol{V}}_0 = \mathbf{0}_m^\top$. Suppose that for any $j \leq k-1$, $\bar{\boldsymbol{R}}_j = \bar{\boldsymbol{V}}_j \mathbf{1}_m, \bar{\boldsymbol{C}}_j = \mathbf{1}_n^\top \bar{\boldsymbol{V}}_j$. Then using the updated rule in Algorithm 2 and (13),

$$\begin{aligned}
\bar{\boldsymbol{R}}_k &= \beta_{2,k}\bar{\boldsymbol{R}}_{k-1} + (1 - \beta_{2,k})\bar{\boldsymbol{G}}_{k,\epsilon_1}^2 \mathbf{1}_m = \left( \beta_{2,k}\bar{\boldsymbol{V}}_{k-1} + (1 - \beta_{2,k})\bar{\boldsymbol{G}}_{k,\epsilon_1}^2 \right)\mathbf{1}_m = \bar{\boldsymbol{V}}_k \mathbf{1}_m, \\
\bar{\boldsymbol{C}}_k &= \beta_{2,k}\bar{\boldsymbol{C}}_{k-1} + (1 - \beta_{2,k})\mathbf{1}_n^\top \bar{\boldsymbol{G}}_{k,\epsilon_1}^2 = \mathbf{1}_n^\top \left( \beta_{2,k}\bar{\boldsymbol{V}}_{k-1} + (1 - \beta_{2,k})\bar{\boldsymbol{G}}_{k,\epsilon_1}^2 \right) = \mathbf{1}_n^\top \bar{\boldsymbol{V}}_k.
\end{aligned} \tag{16}$$

Since $S_{\bar{\boldsymbol{V}}_k}$ represents the coordinate sum of $\bar{\boldsymbol{V}}_k$, we could derive that

$$S_{\bar{\boldsymbol{V}}_k} = \sum_{i=1}^n \sum_{j=1}^m \bar{v}_{ij}^{(k)} = \mathbf{1}_n^\top \bar{\boldsymbol{R}}_k = \mathbf{1}_n^\top \bar{\boldsymbol{V}}_k \mathbf{1}_m.$$

Since $R_{\bar{\boldsymbol{V}}_k}^{(i)}$ denotes the $i$-th row sum of $\bar{\boldsymbol{V}}_k$, it's the $i$-th coordinate of $\bar{\boldsymbol{R}}_k$. Hence, for each coordinate of $\bar{\boldsymbol{R}}_k$, using (16),

$$R_{\bar{\boldsymbol{V}}_k}^{(i)} = \beta_{2,k}R_{\bar{\boldsymbol{V}}_{k-1}}^{(i)} + (1 - \beta_{2,k})R_{\bar{\boldsymbol{G}}_{k,\epsilon_1}^2}^{(i)}.$$

Similarly, we could derive the results related to $C_{\bar{\boldsymbol{V}}_k}^{(j)}$. □

**Lemma A.3.** *Following the parameter setting in (3), for any $i \in [n], j \in [m], k \geq 1$, it holds that*

$$R_{\bar{\boldsymbol{V}}_k}^{(i)} \in [m\epsilon_1(1 - \beta_{2,1}), \mathcal{G}_1], \quad C_{\bar{\boldsymbol{V}}_k}^{(j)} \in [n\epsilon_1(1 - \beta_{2,1}), \mathcal{G}_2], \quad S_{\bar{\boldsymbol{V}}_k} \in [mn\epsilon_1(1 - \beta_{2,1}), \mathcal{G}].$$

*Proof.* Recalling the definition of $\bar{V}_k$ in (13) and $\|\nabla f(X_k)\|_F \leq G, \forall k \geq 1$, we derive that

$$
S_{\bar{V}_k} = \sum_{i=1}^{n}\sum_{j=1}^{m} \bar{v}_{ij}^{(k)} = \sum_{i=1}^{n}\sum_{j=1}^{m}\sum_{p=1}^{k}(1-\beta_{2,p})\left(\left(\bar{g}_{ij}^{(p)}\right)^2 + \epsilon_1\right)\left(\prod_{l=p+1}^{k}\beta_{2,l}\right)
$$

$$
\leq \sum_{p=1}^{k}(1-\beta_{2,p})\left(\prod_{l=p+1}^{k}\beta_{2,l}\right)\|\bar{G}_p\|_F^2 + \Gamma_k mn\epsilon_1 \leq G^2\Gamma_k + mn\epsilon_1 \leq \mathcal{G}, \qquad (17)
$$

where the last inequality comes from Lemma A.1. Following (17) and Lemma A.1, we also derive that

$$
S_{\bar{V}_k} \geq mn\epsilon_1\Gamma_k \geq mn\epsilon_1(1-\beta_{2,1}).
$$

We also derive the upper bounds for $R_{\bar{V}_k}^{(i)}$ and $C_{\bar{V}_k}^{(j)}$ as follows,

$$
R_{\bar{V}_k}^{(i)} = \sum_{j=1}^{m}\bar{v}_{ij}^{(k)} \leq \sum_{p=1}^{k}(1-\beta_{2,p})\left(\prod_{l=p+1}^{k}\beta_{2,l}\right)\|\bar{G}_p\|_F^2 + \Gamma_k m\epsilon_1 \leq G^2\Gamma_k + m\epsilon_1 \leq \mathcal{G}_1,
$$

$$
C_{\bar{V}_k}^{(j)} = \sum_{i=1}^{n}\bar{v}_{ij}^{(k)} \leq \sum_{p=1}^{k}(1-\beta_{2,p})\left(\prod_{l=p+1}^{k}\beta_{2,l}\right)\|\bar{G}_p\|_F^2 + \Gamma_k n\epsilon_1 \leq G^2\Gamma_k + n\epsilon_1 \leq \mathcal{G}_2.
$$

Similarly, the lower bound could be derived by

$$
R_{\bar{V}_k}^{(i)} \geq m\epsilon_1\Gamma_k \geq m\epsilon_1(1-\beta_{2,1}), \quad C_{\bar{V}_k}^{(j)} \geq n\epsilon_1\Gamma_k \geq n\epsilon_1(1-\beta_{2,1}).
$$

$\square$

### A.3 PROOF OF THEOREM A.1

Now we move to prove the main result. Using (15) and the updated rule in Algorithm 2,

$$
f(X_{k+1}) \leq f(X_k) + \langle \bar{G}_k, X_{k+1} - X_k \rangle + \frac{L}{2}\|X_{k+1} - X_k\|_F^2
$$

$$
= f(X_k) - \hat{\eta}_k\left\langle \bar{G}_k, \frac{\bar{G}_k}{\sqrt{\bar{W}_k}}\right\rangle + \frac{L\hat{\eta}_k^2}{2}\left\|\frac{\bar{G}_k}{\sqrt{\bar{W}_k}}\right\|_F^2.
$$

We then re-arrange the order, sum up both sides over $k \in [t]$ and apply $f(X_{t+1}) \geq f^*$ from Assumption (A2) to get,

$$
\underbrace{\sum_{k=1}^{t}\hat{\eta}_k\left\|\frac{\bar{G}_k}{\sqrt[4]{\bar{W}_k}}\right\|_F^2}_{\text{(a)}} \leq f(X_1) - f^* + \frac{L}{2}\underbrace{\sum_{k=1}^{t}\hat{\eta}_k^2\left\|\frac{\bar{G}_k}{\sqrt{\bar{W}_k}}\right\|_F^2}_{\text{(b)}}. \qquad (18)
$$

Since $\|X_k\|_\infty \leq \Theta_{\max}$, we have $\text{RMS}(X_k) \leq \Theta_{\max}$ for any $k \geq 1$. Hence, using $\hat{\eta}_k$ defined in Algorithm 2,

$$
\hat{\eta}_k = \frac{\max\{\epsilon_2, \text{RMS}(X_k)\}\rho_k}{\max\left\{1, \|\bar{U}_k\|_F/(d\sqrt{mn})\right\}} \leq (\epsilon_2 + \Theta_{\max})\rho_k \min\left\{1, \frac{d\sqrt{mn}}{\|\bar{U}_k\|_F}\right\}. \qquad (19)
$$

Using (19), $\bar{U}_k = \bar{G}_k/\sqrt{\bar{W}_k}$, $\Delta_0$ in (12) and $\rho_k = \rho_0/\sqrt{k}$, we thus derive that

$$
\text{(b)} \leq \frac{Ld^2mn(\epsilon_2 + \Theta_{\max})^2}{2}\sum_{k=1}^{t}\rho_k^2 \cdot \frac{\|\bar{U}_k\|_F^2}{\|\bar{U}_k\|_F^2} = \Delta_0^2\sum_{k=1}^{t}\frac{1}{k}. \qquad (20)
$$

To lower bound (a), we first discuss the maximum operator inside $\hat{\eta}_k$. Let

$$
E_1 = \left\{k \in [t] \mid \|\bar{U}_k\|_F \geq d\sqrt{mn}\right\}, \quad E_2 = \left\{k \in [t] \mid \|\bar{U}_k\|_F \leq d\sqrt{mn}\right\}.
$$

When $k \in E_1$, it derives that

$$\hat{\eta}_k \geq \frac{d\sqrt{mn}\epsilon_2\rho_k}{\|\bar{U}_k\|_F}. \tag{21}$$

Using Lemma A.2, we first derive that $\bar{w}_{ij}^{(k)} = (R_{\bar{V}_k}^{(i)} C_{\bar{V}_k}^{(j)})/S_{\bar{V}_k}$. Then, applying Lemma A.3 and $\|\nabla f(X_k)\|_F \leq G$, we could upper bound $\|\bar{U}_k\|_F^2$ as follows,

$$\|\bar{U}_k\|_F^2 = \sum_{i=1}^{n}\sum_{j=1}^{m} \frac{\left(\bar{g}_{ij}^{(k)}\right)^2 S_{\bar{V}_k}}{R_{\bar{V}_k}^{(i)} C_{\bar{V}_k}^{(j)}} \leq \frac{\|\bar{G}_k\|_F^2 \mathcal{G}}{mn\epsilon_1^2(1-\beta_{2,1})^2} \leq \frac{G^2\mathcal{G}}{mn\epsilon_1^2(1-\beta_{2,1})^2}. \tag{22}$$

Hence, combining with (21) and (22), we have

$$\sum_{k \in E_1} \hat{\eta}_k \left\|\frac{\bar{G}_k}{\sqrt[4]{\bar{W}_k}}\right\|_F^2 \geq d\sqrt{mn}\epsilon_2 \sum_{k \in E_1} \frac{\rho_k}{\|\bar{U}_k\|_F} \left\|\frac{\bar{G}_k}{\sqrt[4]{\bar{W}_k}}\right\|_F^2$$

$$\geq \frac{d\epsilon_1 mn(1-\beta_{2,1})\epsilon_2}{G\sqrt{\mathcal{G}}} \sum_{k \in E_1} \rho_k \left\|\frac{\bar{G}_k}{\sqrt[4]{\bar{W}_k}}\right\|_F^2. \tag{23}$$

When $k \in E_2$, we obtain that $\hat{\eta}_k = \max\{\epsilon_2, \text{RMS}(X_k)\}\rho_k \geq \epsilon_2\rho_k$ and thus

$$\sum_{k \in E_2} \hat{\eta}_k \left\|\frac{\bar{G}_k}{\sqrt[4]{\bar{W}_k}}\right\|_F^2 \geq \epsilon_2 \sum_{k \in E_2} \rho_k \left\|\frac{\bar{G}_k}{\sqrt[4]{\bar{W}_k}}\right\|_F^2. \tag{24}$$

Combining with (23) and (24), we derive that

$$(\mathbf{a}) \geq \epsilon_2 \min\left\{1, \frac{d\epsilon_1 mn(1-\beta_{2,1})}{G\sqrt{\mathcal{G}}}\right\} \sum_{k=1}^{t} \rho_k \left\|\frac{\bar{G}_k}{\sqrt[4]{\bar{W}_k}}\right\|_F^2. \tag{25}$$

We also derive from Lemma A.2 and Lemma A.3 that for any $i \in [n], j \in [m]$,

$$\bar{w}_{ij}^{(k)} = \frac{R_{\bar{V}_k}^{(i)} C_{\bar{V}_k}^{(j)}}{S_{\bar{V}_k}} \leq \frac{R_{\bar{V}_k}^{(i)} C_{\bar{V}_k}^{(j)}}{\sqrt{R_{\bar{V}_k}^{(i)} C_{\bar{V}_k}^{(j)}}} \leq \sqrt{R_{\bar{V}_k}^{(i)} C_{\bar{V}_k}^{(j)}} \leq \sqrt{\mathcal{G}_1\mathcal{G}_2}. \tag{26}$$

Using (26), we have

$$\left\|\frac{\bar{G}_k}{\sqrt[4]{\bar{W}_k}}\right\|_F^2 = \sum_{i=1}^{n}\sum_{j=1}^{m} \frac{\left(\bar{g}_{ij}^{(k)}\right)^2}{\sqrt{\bar{w}_{ij}^{(k)}}} \geq \frac{\|\bar{G}_k\|_F^2}{\sqrt[4]{\mathcal{G}_1\mathcal{G}_2}} = \frac{\|\bar{G}_k\|_F^2}{A_1}, \tag{27}$$

where $A_1$ has been defined in (12). Plugging (27) into (25), we derive that

$$(\mathbf{a}) \geq \frac{\epsilon_2}{A_1} \min\left\{1, \frac{d\epsilon_1 mn(1-\beta_{2,1})}{G\sqrt{\mathcal{G}}}\right\} \sum_{k=1}^{t} \rho_k \|\bar{G}_k\|_F^2. \tag{28}$$

Plugging (20) and (28) into (18), and using $\rho_k = \rho_0/\sqrt{k}$, we thus derive that

$$\min_{k \in [t]} \|\bar{G}_k\|_F^2 \sum_{k=1}^{t} \frac{1}{\sqrt{k}} \leq \sum_{k=1}^{t} \frac{\rho_k \|\bar{G}_k\|_F^2}{\rho_0} \leq A_0 A_1 \left(f(X_1) - f^* + \Delta_0^2 \sum_{k=1}^{t} \frac{1}{k}\right), \tag{29}$$

where $A_0$ is given in (12). Moreover, we have the following results,

$$\sum_{k=1}^{t} \frac{1}{k} \leq 1 + \int_1^t \frac{1}{x}dx = 1 + \log t, \quad \sum_{k=1}^{t} \frac{1}{\sqrt{k}} \geq \sqrt{t}. \tag{30}$$

We thus derive the first desired result in (10) as follows,

$$\min_{k \in [t]} \|\bar{G}_k\|_F^2 \leq \frac{A_0 A_1}{\sqrt{t}} \left(f(X_1) - f^* + \Delta_0^2 + \Delta_0^2 \log t\right).$$

**A constant step-size $\rho_k = \rho_0$** Setting $\rho_k = \rho_0$, then following the result in (29), we derive that

$$t \cdot \min_{k \in [t]} \|\bar{\boldsymbol{G}}_k\|_F^2 \leq \sum_{k=1}^t \|\bar{\boldsymbol{G}}_k\|_F^2 \leq A_0 A_1 \left( f(\boldsymbol{X}_1) - f^* + \Delta_0^2 \sum_{k=1}^t \frac{1}{k} \right).$$

Using (30) and dividing $t$ on both sides, we obtain that

$$\min_{k \in [t]} \|\bar{\boldsymbol{G}}_k\|_F^2 \leq \frac{A_0 A_1}{t} \left( f(\boldsymbol{X}_1) - f^* + \Delta_0^2 + \Delta_0^2 \log t \right).$$

**Avoiding the curse of dimension** To derive a free-dimension numerator bound, we first derive from (19) and (22) with $\rho_k = \rho_0/\sqrt{k}$ that

$$\textbf{(b)} \leq \frac{L(\epsilon_2 + \Theta_{\max})^2}{2} \sum_{k=1}^t \rho_k^2 \|\bar{\boldsymbol{U}}_k\|_F^2 \leq \frac{LG^2 \mathcal{G}(\epsilon_2 + \Theta_{\max})^2}{2mn\epsilon_1^2 (1 - \beta_{2,1})^2} \sum_{k=1}^t \rho_k^2 = \tilde{\Delta}_0^2 \sum_{k=1}^t \frac{1}{k}, \quad (31)$$

where $\tilde{\Delta}_0$ has been defined in (12). In addition, we derive from Lemma A.2, Lemma A.3 and (14) that

$$\bar{w}_{ij}^{(k)} = \frac{R_{\bar{\boldsymbol{V}}_k}^{(i)} C_{\bar{\boldsymbol{V}}_k}^{(j)}}{S_{\bar{\boldsymbol{V}}_k}} \leq \frac{2\mathcal{G}_1 \mathcal{G}_2}{mn\epsilon_1} \leq 2 \left( \frac{G^4}{mn\epsilon_1} + G^2 + \epsilon_1 \right) = (A_1')^2,$$

where we use $m + n \leq mn$ and $A_1'$ in (12). Thereby, we have

$$\left\| \frac{\bar{\boldsymbol{G}}_k}{\sqrt[4]{\bar{\boldsymbol{W}}_k}} \right\|_F^2 = \sum_{i=1}^n \sum_{j=1}^m \frac{\left( \bar{g}_{ij}^{(k)} \right)^2}{\sqrt{\bar{w}_{ij}^{(k)}}} \geq \frac{\|\bar{\boldsymbol{G}}_k\|_F^2}{A_1'}.$$

Combining with (25), we thus derive that

$$\textbf{(a)} \geq \frac{\epsilon_2}{A_1'} \min \left\{ 1, \frac{d\epsilon_1 mn(1 - \beta_{2,1})}{G\sqrt{\mathcal{G}}} \right\} \sum_{k=1}^t \rho_k \|\bar{\boldsymbol{G}}_k\|_F^2 \quad (32)$$

Plugging (31) and (32) into (18), and using $\rho_k = \rho_0/\sqrt{k}$, we derive that

$$\min_{k \in [t]} \|\bar{\boldsymbol{G}}_k\|_F^2 \sum_{k=1}^t \frac{1}{\sqrt{k}} \leq \sum_{k=1}^t \frac{\rho_k \|\bar{\boldsymbol{G}}_k\|_F^2}{\rho_0} \leq A_0 A_1' \left( f(\boldsymbol{X}_1) - f^* + \tilde{\Delta}_0^2 \sum_{k=1}^t \frac{1}{k} \right),$$

where $A_0$ has been defined in (12). Using (30), we derive the second desired result in (10).

$$\min_{k \in [t]} \|\bar{\boldsymbol{G}}_k\|_F^2 \leq \frac{A_0 A_1'}{\sqrt{t}} \left( f(\boldsymbol{X}_1) - f^* + \tilde{\Delta}_0^2 + \tilde{\Delta}_0^2 \log t \right).$$

# B PROOF DETAIL FOR STOCHASTIC ADAFACTOR WITHOUT UPDATE CLIPPING

We first provide the detailed version of Theorem 6.1.

**Theorem B.1** (*Formal statement of Theorem 6.1*). *Let $\{\boldsymbol{X}_k\}_{k \geq 1}$ be generated by Algorithm 1 without update clipping where $\eta_k$ is given by (4) for each $k \geq 1$. If Assumptions (A1)-(A4) hold, and*

$$\beta_{2,1} = 1/2, \quad \rho_1 = \rho_0,$$
$$\beta_{2,k} = 1 - 1/k^c, \quad \rho_k = \rho_0/\sqrt{k}, \quad \forall k \geq 2,$$

*for some constants $1/2 \leq c \leq 1, \rho_0 > 0$, then for any $T \geq 1, \delta \in (0,1)$, we have the following results.*
*When $c = 1$, with probability at least $1 - \delta$,*

$$\min_{k \in [T]} \|\bar{\boldsymbol{G}}_k\|_F^2 \leq \frac{C_0}{\sqrt{T}} \left( C_1 \log \left( \frac{T}{\delta} \right) + C_2 \log T + C_2 + C_3 \right), \quad (33)$$

$$\min_{k \in [T]} \|\bar{\boldsymbol{G}}_k\|_F^2 \leq \frac{C_0'}{\sqrt{T}} \left( C_1 \log \left( \frac{T}{\delta} \right) + (C_2' + C_3') \log T + C_2' + C_3' \right). \quad (34)$$

When $1/2 \leq c < 1$, with probability at least $1 - \delta$,

$$\min_{k \in [T]} \|\bar{\boldsymbol{G}}_k\|_F^2 \leq \frac{C_0}{\sqrt{T}} \left( C_1 \log \left( \frac{T}{\delta} \right) + \frac{C_2}{1-c} \cdot T^{1-c} + C_2 + C_3 \right), \tag{35}$$

$$\min_{k \in [T]} \|\bar{\boldsymbol{G}}_k\|_F^2 \leq \frac{C_0'}{\sqrt{T}} \left( C_1 \log \left( \frac{T}{\delta} \right) + \frac{2C_2'}{1-c} \cdot T^{1-c} + C_3' \log T + C_2' + C_3' \right). \tag{36}$$

Here, $\Theta_{\max}$ and $\mathcal{G}$ are as in (11), and

$$C_1 = f(\boldsymbol{X}_1) - f^* + \frac{24G^2(\epsilon_2 + \Theta_{\max})\rho_0}{\sqrt{\epsilon_1}}. \tag{37}$$

The $C_0, C_2, C_3$ are constants defined as

$$C_0 = \frac{2\sqrt{2\mathcal{G}}}{\rho_0 \epsilon_2}, \quad C_3 = \frac{C_2}{4} \log \left( 2 + \frac{2G^2}{\epsilon_1} \right),$$

$$C_2 = \frac{32mn\mathcal{G}^{\frac{3}{2}}(\epsilon_2 + \Theta_{\max})\rho_0}{\max\{m,n\}\epsilon_1} + \frac{4Lmn\mathcal{G}(\epsilon_2 + \Theta_{\max})^2\rho_0^2}{\max\{m,n\}\epsilon_1}. \tag{38}$$

The $C_0', C_2', C_3'$ are positive constants (that could be further upper bounded by constants independent from $m, n$), defined by

$$C_0' = \frac{2\sqrt{2\left(\frac{G^2}{mn\epsilon_1} + G + \epsilon_1\right)}}{\rho_0 \epsilon_2}, C_2' = 4G_3(G_1 + G_2)(\epsilon_2 + \Theta_{\max})\rho_0, C_3' = \frac{LG_3(\epsilon_2 + \Theta_{\max})^2\rho_0^2}{2}, \tag{39}$$

and $G_1, G_2, G_3$ are given by

$$G_1 = \sqrt{6\left(\frac{G^4}{mn\epsilon_1} + G^2 + \epsilon_1\right)}, \quad G_3 = \frac{4(G^4 + G^2mn\epsilon_1)}{mn\epsilon_1^2},$$

$$G_2 = 2\left(\frac{G^3}{mn\epsilon_1} + \frac{2G^2}{\sqrt{mn\epsilon_1}} + \frac{G}{\sqrt{mn}} + G + \sqrt{\epsilon_1}\right). \tag{40}$$

**Calculation of hyper-parameter dependency** To derive a free dimension bound, we shall use the convergence bounds in (34) and (36). From (39), it's easy to show that $m, n$ could only exist in the denominator of $C_0', C_2', C_3'$, which could avoid the curse of dimension.

To calculate the dependency of $\epsilon_1$, we first show that its dependency in coefficients $C_0, C_1, C_2, C_3$ as follows, based on the assumption that $0 < \epsilon_1 < 1$,

$$C_0 \sim \mathcal{O}(1), \quad C_1 \sim \mathcal{O}(1/\sqrt{\epsilon_1}), \quad C_2 \sim \mathcal{O}(1/\epsilon_1), \quad C_3 \sim \mathcal{O}(C_2 \log(1/\epsilon_1)). \tag{41}$$

Thereby, with the convergence bounds in (33) and (35), it's easy to show that

$$\min_{k \in [T]} \|\bar{\boldsymbol{G}}_k\|_F^2 \leq \mathcal{O}\left(\epsilon_1^{-1} \log(1/\epsilon_1)\right). \tag{42}$$

**Proposition B.1.** *Following the same assumptions and settings in Theorem 6.1, then with probability at least $1 - \delta$,*

$$\min_{k \in [T]} \|\bar{\boldsymbol{G}}_k\|_F^2 \leq \frac{C_0}{\sqrt{T}} \left( C_1 \log \left( \frac{T}{\delta} \right) + C_2 \sum_{k=1}^{T} \frac{1}{k^c} + C_3 \right),$$

*and with probability at least $1 - \delta$,*

$$\min_{k \in [T]} \|\bar{\boldsymbol{G}}_k\|_F^2 \leq \frac{C_0'}{\sqrt{T}} \left( C_1 \log \left( \frac{T}{\delta} \right) + C_2' \sum_{k=1}^{T} \frac{1}{k^{c/2+1/2}} + C_3' \sum_{k=1}^{T} \frac{1}{k} \right),$$

*where all constants are given as in Theorem B.1.*

## B.1 PRELIMINARY

We first follow the notations of $\bar{\boldsymbol{G}}_k = \left( \bar{g}_{ij}^{(k)} \right)_{ij}$ and $\mathcal{G}, \mathcal{G}_1, \mathcal{G}_2$ in (14). Let $\boldsymbol{G}_k = \left( g_{ij}^{(k)} \right)_{ij}$ and $\boldsymbol{\xi}_k = \boldsymbol{G}_k - \bar{\boldsymbol{G}}_k$. We also define $\boldsymbol{G}_{k,\epsilon_1}^2 = \boldsymbol{G}_k \odot \boldsymbol{G}_k + \epsilon_1 \mathbf{1}_n \mathbf{1}_m^\top$ and $\boldsymbol{V}_k = \left( v_{ij}^{(k)} \right)_{ij}$ as follows,

$$\boldsymbol{V}_0 = \mathbf{0}_{n \times m}, \quad \boldsymbol{V}_k = \beta_{2,k} \boldsymbol{V}_{k-1} + (1 - \beta_{2,k}) \boldsymbol{G}_{k,\epsilon_1}^2, \quad k \geq 1. \tag{43}$$

We also define $R_{\boldsymbol{V}_k}^{(i)}, C_{\boldsymbol{V}_k}^{(j)}$ and $S_{\boldsymbol{V}_k}$ as the $i$-th row sum, $j$-th column sum and coordinate sum of $\boldsymbol{V}_k$ respectively. $R_{\boldsymbol{G}_{k,\epsilon_1}^2}^{(i)}$ and $C_{\boldsymbol{G}_{k,\epsilon_1}^2}^{(j)}$ represent the same definitions with respect to $\boldsymbol{G}_{k,\epsilon_1}^2$. Then, using a similar deduction in Lemma A.2, we also obtain that for all $k \geq 1$,

$$R_{\boldsymbol{V}_k}^{(i)} = \beta_{2,k} R_{\boldsymbol{V}_{k-1}}^{(i)} + (1 - \beta_{2,k}) \boldsymbol{G}_{k,\epsilon_1}^2 \mathbf{1}_m, \quad C_{\boldsymbol{V}_k}^{(j)} = \beta_{2,k} C_{\boldsymbol{V}_{k-1}}^{(j)} + (1 - \beta_{2,k}) \mathbf{1}_n^\top \boldsymbol{G}_{k,\epsilon_1}^2. \tag{44}$$

As a consequence of (44), each coordinate of $\boldsymbol{W}_k$ satisfies that

$$w_{ij}^{(k)} = \frac{R_{\boldsymbol{V}_k}^{(i)} C_{\boldsymbol{V}_k}^{(j)}}{S_{\boldsymbol{V}_k}} = \frac{\left( \beta_{2,k} R_{\boldsymbol{V}_{k-1}}^{(i)} + (1 - \beta_{2,k}) R_{\boldsymbol{G}_{k,\epsilon_1}^2}^{(i)} \right) \left( \beta_{2,k} C_{\boldsymbol{V}_{k-1}}^{(j)} + (1 - \beta_{2,k}) C_{\boldsymbol{G}_{k,\epsilon_1}^2}^{(j)} \right)}{\beta_{2,k} S_{\boldsymbol{V}_{k-1}} + (1 - \beta_{2,k}) S_{\boldsymbol{G}_{k,\epsilon_1}^2}}. \tag{45}$$

Next, we introduce a proxy step-size matrix $\boldsymbol{A}_k = \left( a_{ij}^{(k)} \right)_{ij}$ such that

$$a_{ij}^{(k)} = \frac{\left( \beta_{2,k} R_{\boldsymbol{V}_{k-1}}^{(i)} + (1 - \beta_{2,k}) \mathcal{G}_1 \right) \left( \beta_{2,k} C_{\boldsymbol{V}_{k-1}}^{(j)} + (1 - \beta_{2,k}) \mathcal{G}_2 \right)}{\beta_{2,k} S_{\boldsymbol{V}_{k-1}} + (1 - \beta_{2,k}) \mathcal{G}}. \tag{46}$$

The proxy step-size technique is a standard way in the convergence analysis of adaptive methods, e.g., Ward et al. (2020); Défossez et al. (2022). We provide a new proxy step-size in (46) to handle the matrix factorization in Adafactor. This construction satisfies two properties. First, it's independent from $\boldsymbol{Z}_k$ in order to disrupt the correlation of stochastic gradients and adaptive step-sizes. Second, it needs to remain sufficiently close to the original adaptive step-size $w_{ij}^{(k)}$ to avoid generating divergent terms.

## B.2 TECHNICAL LEMMAS

In the following, we first provide some more necessary technical lemmas. We introduce a concentration inequality for the martingale difference sequence, see (Li & Orabona, 2020) for a proof.

**Lemma B.1.** *Suppose that $\{Z_s\}_{s \in [T]}$ is a martingale difference sequence with respect to $\zeta_1, \cdots, \zeta_T$. Assume that for each $s \in [T]$, $\sigma_s$ is a random variable dependent on $\zeta_1, \cdots, \zeta_{s-1}$ and satisfies that*

$$\mathbb{E} \left[ \exp \left( \frac{Z_s^2}{\sigma_s^2} \right) \mid \zeta_1, \cdots, \zeta_{s-1} \right] \leq \mathrm{e}.$$

*Then for any $\lambda > 0$, and for any $\delta \in (0, 1)$, it holds that*

$$\mathbb{P} \left( \sum_{s=1}^T Z_s > \frac{1}{\lambda} \log \left( \frac{1}{\delta} \right) + \frac{3}{4} \lambda \sum_{s=1}^T \sigma_s^2 \right) \leq \delta.$$

**Lemma B.2.** *Following the parameter setting in (5), for any $i \in [n], j \in [m], k \geq 1$, it holds that*

$$R_{\boldsymbol{G}_{k,\epsilon_1}^2}^{(i)}, R_{\boldsymbol{V}_k}^{(i)} \in [m\epsilon_1/2, \mathcal{G}_1], \quad C_{\boldsymbol{G}_{k,\epsilon_1}^2}^{(j)}, C_{\boldsymbol{V}_k}^{(j)} \in [n\epsilon_1/2, \mathcal{G}_2], \quad S_{\boldsymbol{G}_{k,\epsilon_1}^2}, S_{\boldsymbol{V}_k} \in [mn\epsilon_1/2, \mathcal{G}].$$

*Proof.* First, using Assumption (A4), we derive that

$$mn\epsilon_1/2 \le S_{\boldsymbol{G}_{k,\epsilon_1}^2} = \sum_{i=1}^{n}\sum_{j=1}^{m}\left(\left(g_{ij}^{(k)}\right)^2 + \epsilon_1\right) = \|\boldsymbol{G}_k\|_F^2 + mn\epsilon_1 \le \mathcal{G},$$

$$m\epsilon_1/2 \le R_{\boldsymbol{G}_{k,\epsilon_1}^2}^{(i)} = \sum_{j=1}^{m}\left(\left(g_{ij}^{(k)}\right)^2 + \epsilon_1\right) \le \|\boldsymbol{G}_k\|_F^2 + m\epsilon_1 \le \mathcal{G}_1,$$

$$n\epsilon_1/2 \le C_{\boldsymbol{G}_{k,\epsilon_1}^2}^{(j)} = \sum_{i=1}^{n}\left(\left(g_{ij}^{(k)}\right)^2 + \epsilon_1\right) \le \|\boldsymbol{G}_k\|_F^2 + n\epsilon_1 \le \mathcal{G}_2.$$

Using the similar deduction for Lemma A.3, we could show that $m\epsilon_1(1 - \beta_{2,1}) \le R_{\boldsymbol{V}_k}^{(i)} \le \mathcal{G}_1$. Since $\beta_{2,1} = 1/2$ from (5), we then obtain the desired result. The bounds for $C_{\boldsymbol{V}_k}^{(j)}, S_{\boldsymbol{V}_k}$ could be also derived by using similar arguments. $\qquad\square$

We have the following lemma to upper bound each coordinate of the proxy step-size matrix $\boldsymbol{A}_k$ defined in (46) .

**Lemma B.3.** *For any $k \ge 1$, it holds that*

$$\beta_{2,k}(1 - \beta_{2,k})\epsilon_1 \le a_{ij}^{(k)} \le 2\min\left\{\mathcal{G}, \frac{G^2}{mn\epsilon_1} + G + \epsilon_1\right\}, \quad \forall i \in [n], j \in [m].$$

*Proof.* We first have

$$\frac{\beta_{2,k}R_{\boldsymbol{V}_{k-1}}^{(i)} + (1 - \beta_{2,k})\mathcal{G}_1}{\beta_{2,k}S_{\boldsymbol{V}_{k-1}} + (1 - \beta_{2,k})\mathcal{G}} \le \frac{\beta_{2,k}R_{\boldsymbol{V}_{k-1}}^{(i)}}{\beta_{2,k}S_{\boldsymbol{V}_{k-1}}} + \frac{(1 - \beta_{2,k})\mathcal{G}_1}{(1 - \beta_{2,k})\mathcal{G}} \le 2. \tag{47}$$

Then, recalling the definition of $a_{ij}^{(k)}$ in (46) and Lemma B.2, it derives that $C_{\boldsymbol{V}_{k-1}}^{(j)} \le \mathcal{G}_2$ and thereby $\beta_{2,k}C_{\boldsymbol{V}_{k-1}}^{(j)} + (1 - \beta_{2,k})\mathcal{G}_2 \le \mathcal{G}_2 \le \mathcal{G}$. Then combining with (47), we derive $a_{ij}^{(k)} \le 2\mathcal{G}$. We also derive a free dimension bound from Lemma B.2 for $a_{ij}^{(k)}$ as follows,

$$a_{ij}^{(k)} \le \frac{2\mathcal{G}_1\mathcal{G}_2}{mn\epsilon_1} = \frac{2(G^2 + G(m + n)\epsilon_1 + mn\epsilon_1^2)}{mn\epsilon_1} \le 2\left(\frac{G^2}{mn\epsilon_1} + G + \epsilon_1\right),$$

where we use $m + n \le mn$ when $m, n \ge 2$ and $\beta_{2,k}S_{\boldsymbol{V}_{k-1}} + (1 - \beta_{2,k})\mathcal{G} \ge mn\epsilon_1/2$. To lower bound $a_{ij}^{(k)}$, we derive from Lemma B.2 that $\beta_{2,k}S_{\boldsymbol{V}_{k-1}} + (1 - \beta_{2,k})\mathcal{G} \le \mathcal{G}$. Thereby,

$$a_{ij}^{(k)} \ge \frac{\beta_{2,k}(1 - \beta_{2,k})\left(R_{\boldsymbol{V}_{k-1}}^{(i)}\mathcal{G}_2 + C_{\boldsymbol{V}_{k-1}}^{(j)}\mathcal{G}_1\right)}{\mathcal{G}} \ge \beta_{2,k}(1 - \beta_{2,k}) \cdot \frac{(m\mathcal{G}_2 + n\mathcal{G}_1)\epsilon_1}{2\mathcal{G}}$$

$$= \beta_{2,k}(1 - \beta_{2,k}) \cdot \frac{[(m + n)G^2 + 2mn\epsilon_1]\epsilon_1}{2(G^2 + mn\epsilon_1)} \ge \beta_{2,k}(1 - \beta_{2,k})\epsilon_1.$$

$\qquad\square$

**Lemma B.4.** *Let $\boldsymbol{W}_k$ and $\boldsymbol{V}_k$ be defined in Algorithm 1 without update clipping where $\eta_k$ is given by (4) and (43) respectively. For any $k \ge 1$, it holds that*

$$\left\|\frac{\boldsymbol{G}_k}{\sqrt{\boldsymbol{W}_k}}\right\|_F^2 \le \frac{2\mathcal{G}}{\max\{m, n\}\epsilon_1}\left\|\frac{\boldsymbol{G}_k}{\sqrt{\boldsymbol{V}_k}}\right\|_F^2.$$

*Proof.* Recalling (45), $v_{ij}^{(k)} \le R_{\boldsymbol{V}_k}^{(i)}$ ,$v_{ij}^{(k)} \le C_{\boldsymbol{V}_k}^{(j)}$ and Lemma B.2, one could verify that

$$\frac{\left(g_{ij}^{(k)}\right)^2}{w_{ij}^{(k)}} = \frac{\left(g_{ij}^{(k)}\right)^2 S_{\boldsymbol{V}_k}}{R_{\boldsymbol{V}_k}^{(i)}C_{\boldsymbol{V}_k}^{(j)}} \le \frac{2\left(g_{ij}^{(k)}\right)^2 \mathcal{G}}{n\epsilon_1 v_{ij}^{(k)}}, \quad \frac{\left(g_{ij}^{(k)}\right)^2}{w_{ij}^{(k)}} = \frac{\left(g_{ij}^{(k)}\right)^2 S_{\boldsymbol{V}_k}}{R_{\boldsymbol{V}_k}^{(i)}C_{\boldsymbol{V}_k}^{(j)}} \le \frac{2\left(g_{ij}^{(k)}\right)^2 \mathcal{G}}{m\epsilon_1 v_{ij}^{(k)}},$$

which leads to the desired result that

$$\|\boldsymbol{U}_k\|_F^2 = \left\|\frac{\boldsymbol{G}_k}{\sqrt{\boldsymbol{W}_k}}\right\|_F^2 \leq \frac{2\mathcal{G}}{\max\{m,n\}\epsilon_1} \left\|\frac{\boldsymbol{G}_k}{\sqrt{\boldsymbol{V}_k}}\right\|_F^2.$$

□

The following lemma is inspired by (Défossez et al., 2022, Lemma 5.2) where they considered a constant $\beta_{2,k}$. Here, we generalize the result to the case of time-varying $\beta_{2,k}$ and provide the proof detail.

**Lemma B.5.** *For any $t \geq 1$, if $\beta_{2,k}$ are as in (5), then it holds that*

$$\sum_{k=1}^{t}(1-\beta_{2,k})\left\|\frac{\boldsymbol{G}_k}{\sqrt{\boldsymbol{V}_k}}\right\|_F^2 \leq mn\log\left(\frac{2(G^2+\epsilon_1)}{\epsilon_1}\right) + 4mn\sum_{k=1}^{t}(1-\beta_{2,k}).$$

*Proof.* Recalling the definition of $\boldsymbol{V}_k$ and since $\boldsymbol{V}_0 = \boldsymbol{0}_{n\times m}$, we have that for any $k \geq 1$,

$$v_{ij}^{(k)} = \beta_{2,k}v_{ij}^{(k-1)} + (1-\beta_{2,k})\left[\left(g_{ij}^{(k)}\right)^2 + \epsilon_1\right]$$

$$= \sum_{p=1}^{k}(1-\beta_{2,p})\left[\left(g_{ij}^{(p)}\right)^2 + \epsilon_1\right]\left(\prod_{l=p+1}^{k}\beta_{2,l}\right).$$

Then, we have

$$(1-\beta_{2,k})\cdot\frac{\left(g_{ij}^{(k)}\right)^2}{v_{ij}^{(k)}} = \frac{x_k}{y_k+\theta_k}, \tag{48}$$

where we set $y_0 = 0, \theta_0 = 0$ and

$$x_k = (1-\beta_{2,k})\left(g_{ij}^{(k)}\right)^2, \quad y_k = \sum_{p=1}^{k}(1-\beta_{2,p})\left(g_{ij}^{(p)}\right)^2\left(\prod_{l=p+1}^{k}\beta_{2,l}\right),$$

$$\theta_k = \epsilon_1\sum_{p=1}^{k}(1-\beta_{2,p})\left(\prod_{l=p+1}^{k}\beta_{2,l}\right), \quad \forall k \geq 1.$$

Then we have $y_k - x_k = \beta_{2,k}y_{k-1}, \forall k \geq 1$. Moreover, since $y_k \geq x_k$, we could use $\log x \geq 1 - 1/x, \forall x \geq 1$ to derive that

$$\frac{x_k}{y_k+\theta_k} \leq \log(y_k+\theta_k) - \log(y_k+\theta_k-x_k) = \log(y_k+\theta_k) - \log(\beta_{2,k}y_{k-1}+\theta_k)$$

$$= \log\left(\frac{y_k+\theta_k}{y_{k-1}+\theta_{k-1}}\right) + \log\left(\frac{y_{k-1}+\theta_{k-1}}{\beta_{2,k}y_{k-1}+\theta_k}\right).$$

Noting that $\theta_k = \beta_{2,k}\theta_{k-1} + (1-\beta_{2,k})\epsilon_1$, which leads to $\beta_{2,k}\theta_{k-1} \leq \theta_k$. Hence, we further have

$$\frac{x_k}{y_k+\theta_k} \leq \log\left(\frac{y_k+\theta_k}{y_{k-1}+\theta_{k-1}}\right) + \log\left(\frac{y_{k-1}+\theta_{k-1}}{\beta_{2,k}(y_{k-1}+\theta_{k-1})}\right) = \log\left(\frac{y_k+\theta_k}{y_{k-1}+\theta_{k-1}}\right) - \log\beta_{2,k}. \tag{49}$$

Hence, summing up on both sides of (48) and (49) over $k \in [t]$, and noting that $x_1 = y_1$, we obtain that

$$\sum_{k=1}^{t}(1-\beta_{2,k})\cdot\frac{\left(g_{ij}^{(k)}\right)^2}{v_{ij}^{(k)}} = \frac{x_1}{y_1+\theta_1} + \sum_{k=2}^{t}\frac{x_k}{y_k+\epsilon_k}$$

$$\leq 1 + \log\left(\frac{y_t+\theta_t}{y_1+\theta_1}\right) - \sum_{k=2}^{t}\log\beta_{2,k}. \tag{50}$$

Note that $y_1 + \theta_1 \geq (1 - \beta_{2,1})\epsilon_1 = \epsilon_1/2$. Moreover, using Lemma A.1 and Assumption (A4), we have $\theta_t = \Gamma_t \epsilon_1 \leq \epsilon_1$ and $y_t \leq \Gamma_t G^2 \leq G^2$. We then derive that

$$\frac{y_t + \theta_t}{y_1 + \theta_1} \leq \frac{2(G^2 + \epsilon_1)}{\epsilon_1}. \tag{51}$$

Noting that for $k \geq 2$, $c \in [1/2, 1]$, $\beta_{2,k} \geq \beta_{2,2} = 1 - 1/2^c \geq 1 - 1/\sqrt{2}$, we then derive that

$$-\log \beta_{2,k} \leq \frac{1 - \beta_{2,k}}{\beta_{2,k}} \leq \frac{\sqrt{2}(1 - \beta_{2,k})}{\sqrt{2} - 1} \leq 4(1 - \beta_{2,k}). \tag{52}$$

Finally, plugging (51), (52) into (50), and then summing (50) up over $i \in [n], j \in [m]$, we obtain the desired result. $\qquad\square$

Next, we have the following probabilistic result relying on the property of the martingale difference sequence which is commonly used in the analysis of adaptive methods.

**Lemma B.6.** *Following the parameter setting in* (5), *for any $T \geq 1$ and $\lambda > 0$, with probability at least $1 - \delta$, $\forall t \in [T]$,*

$$-\sum_{k=1}^{t} \eta_k \left\langle \bar{\boldsymbol{G}}_k, \frac{\boldsymbol{\xi}_k}{\sqrt{\boldsymbol{A}_k}} \right\rangle \leq \frac{1}{4} \sum_{k=1}^{t} \eta_k \left\| \frac{\bar{\boldsymbol{G}}_k}{\sqrt[4]{\boldsymbol{A}_k}} \right\|_F^2 + \frac{24 G^2 (\epsilon_2 + \Theta_{\max}) \rho_0}{\sqrt{\epsilon_1}} \log\left(\frac{T}{\delta}\right).$$

*Proof.* Let $\zeta_k = -\eta_k \left\langle \bar{\boldsymbol{G}}_k, \frac{\boldsymbol{\xi}_k}{\sqrt{\boldsymbol{A}_k}} \right\rangle$ and the filtration $\mathcal{F}_k = \sigma(\boldsymbol{Z}_1, \cdots, \boldsymbol{Z}_k)$ where $\sigma(\cdot)$ denotes the $\sigma$-algebra. Note that $\eta_k$, $\bar{\boldsymbol{G}}_k$ and $\boldsymbol{A}_k$ are dependent by $\{\boldsymbol{X}_1, \cdots, \boldsymbol{X}_{k-1}\}$ and thereby $\mathcal{F}_{k-1}$. Since $\boldsymbol{\xi}_k$ is dependent by $\mathcal{F}_k$, we could prove that $\{\zeta_k\}_{k \geq 1}$ is a martingale difference sequence since

$$\mathbb{E}[\zeta_k \mid \mathcal{F}_{k-1}] = -\eta_k \left\langle \bar{\boldsymbol{G}}_k, \frac{\mathbb{E}[\boldsymbol{\xi}_k \mid \mathcal{F}_{k-1}]}{\sqrt{\boldsymbol{A}_k}} \right\rangle = 0,$$

where we apply that $\mathbb{E}[\boldsymbol{\xi}_k \mid \mathcal{F}_{k-1}] = \mathbb{E}_{\boldsymbol{Z}_k}[\boldsymbol{\xi}_k] = 0$ from Assumption (A3). Then, using Assumption (A3) and Assumption (A4), we have

$$\|\bar{\boldsymbol{G}}_k\|_F = \|\mathbb{E}_{\boldsymbol{Z}_k}[\boldsymbol{G}_k]\|_F \leq \mathbb{E}_{\boldsymbol{Z}_k}\|\boldsymbol{G}_k\|_F \leq G, \quad \|\boldsymbol{\xi}_k\|_F = \|\boldsymbol{G}_k - \bar{\boldsymbol{G}}_k\|_F \leq 2G.$$

Let $\omega_k = 2G\eta_k \left\| \frac{\bar{\boldsymbol{G}}_k}{\sqrt{\boldsymbol{A}_k}} \right\|_F$. We thus derive from the Cauchy-Schwarz inequality that

$$\mathbb{E}\left[\exp\left(\frac{\zeta_k^2}{\omega_k^2}\right) \mid \mathcal{F}_{k-1}\right] \leq \mathbb{E}\left[\exp\left(\frac{\left\|\frac{\bar{\boldsymbol{G}}_k}{\sqrt{\boldsymbol{A}_k}}\right\|_F^2 \|\boldsymbol{\xi}_k\|_F^2}{4G^2 \left\|\frac{\bar{\boldsymbol{G}}_k}{\sqrt{\boldsymbol{A}_k}}\right\|_F^2}\right) \mid \mathcal{F}_{k-1}\right] \leq \exp(1). \tag{53}$$

Then, using Lemma B.1, it leads to that for any $\lambda > 0$, with probability at least $1 - \delta$,

$$-\sum_{k=1}^{t} \eta_k \left\langle \bar{\boldsymbol{G}}_k, \frac{\boldsymbol{\xi}_k}{\sqrt{\boldsymbol{A}_k}} \right\rangle \leq 3\lambda G^2 \sum_{k=1}^{t} \eta_k^2 \left\| \frac{\bar{\boldsymbol{G}}_k}{\sqrt{\boldsymbol{A}_k}} \right\|_F^2 + \frac{1}{\lambda} \log\left(\frac{1}{\delta}\right)$$

$$= 3\lambda G^2 \sum_{k=1}^{t} \sum_{i=1}^{n} \sum_{j=1}^{m} \frac{\eta_k}{\sqrt{a_{ij}^{(k)}}} \cdot \eta_k \frac{\left(\bar{g}_{ij}^{(k)}\right)^2}{\sqrt{a_{ij}^{(k)}}} + \frac{1}{\lambda} \log\left(\frac{1}{\delta}\right). \tag{54}$$

Meanwhile, when $\|\boldsymbol{X}_k\|_\infty \leq \Theta_{\max}$, $\rho_k = \rho_0/\sqrt{k}$, we have

$$\text{RMS}(\boldsymbol{X}_k) \leq \Theta_{\max}, \quad \frac{\epsilon_2 \rho_0}{\sqrt{k}} \leq \eta_k \leq \frac{(\epsilon_2 + \Theta_{\max})\rho_0}{\sqrt{k}}. \tag{55}$$

Combining with Lemma B.3, we derive that

$$\frac{\eta_k}{\sqrt{a_{ij}^{(k)}}} \leq \frac{\eta_k}{\sqrt{\beta_{2,k}(1 - \beta_{2,k})\epsilon_1}} \leq \frac{(\epsilon_2 + \Theta_{\max})\rho_0}{\sqrt{\beta_{2,k}\epsilon_1}} \cdot \frac{k^{c/2}}{\sqrt{k}}$$

$$\leq \frac{(\epsilon_2 + \Theta_{\max})\rho_0}{\sqrt{\min\{\beta_{2,1}, \beta_{2,2}\}\epsilon_1}} \leq \frac{2(\epsilon_2 + \Theta_{\max})\rho_0}{\sqrt{\epsilon_1}}, \tag{56}$$

where we use $\beta_{2,1} = 1/2, \beta_{2,2} = 1 - 1/2^c \geq 1 - 1/\sqrt{2}, c \in [1/2, 1]$ from (5) in the last inequality. Hence, plugging (56) into (54) and then re-scaling the $\delta$, we found that with probability at least $1 - \delta$, for all $t \in [T]$,

$$-\sum_{k=1}^{t} \eta_k \left\langle \bar{G}_k, \frac{\xi_k}{\sqrt{A_k}} \right\rangle \leq \frac{6\lambda G^2(\epsilon_2 + \Theta_{\max})\rho_0}{\sqrt{\epsilon_1}} \sum_{k=1}^{t} \eta_k \left\| \frac{\bar{G}_k}{\sqrt[4]{A_k}} \right\|_F^2 + \frac{1}{\lambda} \log\left(\frac{T}{\delta}\right).$$

Setting $\lambda = \sqrt{\epsilon_1}/(24 G^2 (\epsilon_2 + \Theta_{\max})\rho_0)$, we derive the desired result. $\qquad\square$

The following key lemma provides an upper bound for the error brought by the proxy step-size $a_{ij}^{(k)}$, illustrating the error is controllable.

**Lemma B.7.** *For any $k \geq 1, i \in [n], j \in [m]$, it holds that*

$$\frac{\left| w_{ij}^{(k)} - a_{ij}^{(k)} \right|}{\sqrt{a_{ij}^{(k)}}} \leq \sqrt{1 - \beta_{2,k}} \min\{4\sqrt{\mathcal{G}}, G_1 + G_2\}, \tag{57}$$

*where $\mathcal{G}$ is as in (14) and $G_1, G_2$ are as in (40).*

*Proof.* To simplify the notation, we let

$$X = \beta_{2,k} R_{V_{k-1}}^{(i)} + (1 - \beta_{2,k}) R_{G_{k,\epsilon_1}^2}^{(i)}, \quad \Delta X = (1 - \beta_{2,k})(\mathcal{G}_1 - R_{G_{k,\epsilon_1}^2}^{(i)}),$$

$$Y = \beta_{2,k} C_{V_{k-1}}^{(j)} + (1 - \beta_{2,k}) C_{G_{k,\epsilon_1}^2}^{(j)}, \quad \Delta Y = (1 - \beta_{2,k})(\mathcal{G}_2 - C_{G_{k,\epsilon_1}^2}^{(j)}),$$

$$Z = \beta_{2,k} S_{V_{k-1}} + (1 - \beta_{2,k}) S_{G_{k,\epsilon_1}^2}, \quad \Delta Z = (1 - \beta_{2,k})(\mathcal{G} - S_{G_{k,\epsilon_1}^2}). \tag{58}$$

Then we have

$$\left| w_{ij}^{(k)} - a_{ij}^{(k)} \right| = \left| \frac{XY}{Z} - \frac{(X + \Delta X)(Y + \Delta Y)}{Z + \Delta Z} \right| = \left| \frac{XY\Delta Z - XZ\Delta Y - YZ\Delta X - Z(\Delta X \Delta Y)}{Z(Z + \Delta Z)} \right|.$$

Applying Lemma B.2, we could verify that $X, Y, Z \geq 0$ and

$$0 \leq \Delta X \leq (1 - \beta_{2,k})\mathcal{G}_1, \quad 0 \leq \Delta Y \leq (1 - \beta_{2,k})\mathcal{G}_2, \quad 0 \leq \Delta Z \leq (1 - \beta_{2,k})\mathcal{G}. \tag{59}$$

Hence, we derive that

$$\frac{\left| w_{ij}^{(k)} - a_{ij}^{(k)} \right|}{\sqrt{a_{ij}^{(k)}}} = \frac{|XY\Delta Z - XZ\Delta Y - YZ\Delta X - Z(\Delta X \Delta Y)|}{Z\sqrt{(X + \Delta X)(Y + \Delta Y)(Z + \Delta Z)}}$$

$$\leq \underbrace{\frac{|X\Delta Y + Y\Delta X + (\Delta X \Delta Y)|}{\sqrt{(X + \Delta X)(Y + \Delta Y)(Z + \Delta Z)}}}_{(\mathbf{I})} + \underbrace{\frac{XY\Delta Z}{Z\sqrt{(X + \Delta X)(Y + \Delta Y)(Z + \Delta Z)}}}_{(\mathbf{II})}. \tag{60}$$

Since $XY \geq 0$ from (58), Term $(\mathbf{I})$ could be bounded as

$$(\mathbf{I}) \leq \frac{|X\Delta Y + Y\Delta X + (\Delta X \Delta Y)|}{\sqrt{(X\Delta Y + Y\Delta X + (\Delta X \Delta Y))(Z + \Delta Z)}} \leq \sqrt{\frac{X\Delta Y + Y\Delta X + (\Delta X \Delta Y)}{Z + \Delta Z}}. \tag{61}$$

Recalling the definition, we have $R_{V_{k-1}}^{(i)} \leq S_{V_{k-1}}, C_{V_{k-1}}^{(j)} \leq S_{V_{k-1}}$ for any $i \in [n], j \in [m]$. Further, applying Lemma B.2 and (59), we derive that

$$\frac{X\Delta Y}{Z + \Delta Z} \leq \left( \frac{R_{V_{k-1}}^{(i)}}{S_{V_{k-1}}} + \frac{R_{G_{k,\epsilon_1}^2}^{(i)}}{\mathcal{G}} \right) \Delta Y \leq 2(1 - \beta_{2,k})\mathcal{G}_2.$$

$$\frac{Y\Delta X}{Z + \Delta Z} \leq \left( \frac{C_{V_{k-1}}^{(j)}}{S_{V_{k-1}}} + \frac{C_{G_{k,\epsilon_1}^2}^{(j)}}{\mathcal{G}} \right) \Delta X \leq 2(1 - \beta_{2,k})\mathcal{G}_1,$$

$$\frac{\Delta X \Delta Y}{Z + \Delta Z} \leq \frac{\Delta X(1 - \beta_{2,k})\mathcal{G}}{(1 - \beta_{2,k})\mathcal{G}} \leq (1 - \beta_{2,k})\mathcal{G}_1.$$

We then derive from (61), $\mathcal{G}_1 \le \mathcal{G}$ and $\mathcal{G}_2 \le \mathcal{G}$ that

$$(\mathbf{I}) \le \sqrt{5(1 - \beta_{2,k})\mathcal{G}}. \tag{62}$$

To derive a free dimension bound, we could obtain from Lemma B.2, (59) and $\mathcal{G} \ge mn\epsilon_1/2$ that $Z + \Delta Z \ge mn\epsilon_1/2$. Hence,

$$\frac{X\Delta Y}{Z + \Delta Z} \le \frac{2(1 - \beta_{2,k})\mathcal{G}_1\mathcal{G}_2}{mn\epsilon_1}, \quad \frac{Y\Delta X}{Z + \Delta Z} \le \frac{2(1 - \beta_{2,k})\mathcal{G}_1\mathcal{G}_2}{mn\epsilon_1}, \quad \frac{\Delta X\Delta Y}{Z + \Delta Z} \le \frac{2(1 - \beta_{2,k})\mathcal{G}_1\mathcal{G}_2}{mn\epsilon_1}.$$

We then derive that

$$(\mathbf{I}) \le \sqrt{\frac{6(1 - \beta_{2,k})\mathcal{G}_1\mathcal{G}_2}{mn\epsilon_1}} = \sqrt{\frac{6(1 - \beta_{2,k})(G^4 + G^2\epsilon_1(m + n) + mn\epsilon_1^2)}{mn\epsilon_1}} \le G_1\sqrt{1 - \beta_{2,k}}, \tag{63}$$

where we used $m + n \le mn$, and $G_1$ is defined in (40). Then, combining with (62) and (63), we have

$$(\mathbf{I}) \le \sqrt{1 - \beta_{2,k}} \min\{\sqrt{5\mathcal{G}}, G_1\}, \tag{64}$$

where we applied that $m + n \le mn$ when $m, n \ge 2$. Then we move to bound **(II)**. Recalling the definitions in (58), we have $X \le Z, Y \le Z$. Applying (59), we have

$$(\mathbf{II}) \le \frac{XY\Delta Z}{Z\sqrt{XY\Delta Z}} \le \frac{\sqrt{XY\Delta Z}}{Z} \le \sqrt{\Delta Z} \le \sqrt{(1 - \beta_{2,k})\mathcal{G}}.$$

Similarly, we derive from Lemma B.2 that $Z \ge mn\epsilon_1/2$, $X \le \mathcal{G}_1, Y \le \mathcal{G}_2$. Hence,

$$(\mathbf{II}) \le \frac{\sqrt{XY\Delta Z}}{Z} \le \frac{2\sqrt{(1 - \beta_{2,k})\mathcal{G}_1\mathcal{G}_2\mathcal{G}}}{mn\epsilon_1}$$

$$\le 2\sqrt{1 - \beta_{2,k}}\left(\frac{G^3}{mn\epsilon_1} + \frac{2G^2}{\sqrt{mn\epsilon_1}} + G + \frac{G}{\sqrt{mn}} + \sqrt{\epsilon_1}\right) \le G_2\sqrt{1 - \beta_{2,k}},$$

where $G_2$ has been defined in (40). We thus derive that

$$(\mathbf{II}) \le \sqrt{1 - \beta_{2,k}} \min\{\sqrt{\mathcal{G}}, G_2\}. \tag{65}$$

Combining (64) with (65) into (60), we then derive the desired result. $\qquad\square$

### B.3 Proof of Proposition B.1

Using the inequality in (15), we have

$$f(\boldsymbol{X}_{k+1}) \le f(\boldsymbol{X}_k) + \langle \bar{\boldsymbol{G}}_k, \boldsymbol{X}_{k+1} - \boldsymbol{X}_k \rangle + \frac{L}{2}\|\boldsymbol{X}_{k+1} - \boldsymbol{X}_k\|_F^2$$

$$\le f(\boldsymbol{X}_k) - \eta_k\left\langle \bar{\boldsymbol{G}}_k, \frac{\boldsymbol{G}_k}{\sqrt{\boldsymbol{W}_k}}\right\rangle + \frac{L\eta_k^2}{2}\left\|\frac{\boldsymbol{G}_k}{\sqrt{\boldsymbol{W}_k}}\right\|_F^2.$$

Introducing the proxy step-size matrix $\boldsymbol{A}_k$ in (46) and then summing up both sides over $k \in [t]$, we derive that

$$f(\boldsymbol{X}_{t+1}) \le f(\boldsymbol{X}_1) \underbrace{- \sum_{k=1}^{t}\eta_k\left\langle \bar{\boldsymbol{G}}_k, \frac{\boldsymbol{G}_k}{\sqrt{\boldsymbol{A}_k}}\right\rangle}_{\mathbf{A}}$$

$$\underbrace{+ \sum_{k=1}^{t}\eta_k\left\langle \bar{\boldsymbol{G}}_k, \boldsymbol{G}_k \odot \left(\frac{1}{\sqrt{\boldsymbol{A}_k}} - \frac{1}{\sqrt{\boldsymbol{W}_k}}\right)\right\rangle}_{\mathbf{B}} + \underbrace{\sum_{k=1}^{t}\frac{L\eta_k^2}{2}\left\|\frac{\boldsymbol{G}_k}{\sqrt{\boldsymbol{W}_k}}\right\|_F^2}_{\mathbf{C}}. \tag{66}$$

**Estimation for A** We first introduce $\boldsymbol{\xi}_k$ into $\mathbf{A}$,

$$\mathbf{A} = -\sum_{k=1}^{t} \eta_k \left\| \frac{\bar{\boldsymbol{G}}_k}{\sqrt[4]{\boldsymbol{A}_k}} \right\|_F^2 - \sum_{k=1}^{t} \eta_k \left\langle \bar{\boldsymbol{G}}_k, \frac{\boldsymbol{\xi}_k}{\sqrt{\boldsymbol{A}_k}} \right\rangle. \tag{67}$$

Then, using Lemma B.6, with probability at least $1 - \delta$, for all $t \in [T]$,

$$\mathbf{A} = -\frac{3}{4} \sum_{k=1}^{t} \eta_k \left\| \frac{\bar{\boldsymbol{G}}_k}{\sqrt[4]{\boldsymbol{A}_k}} \right\|_F^2 + \frac{24 G^2 (\epsilon_2 + \Theta_{\max}) \rho_0}{\sqrt{\epsilon_1}} \log\left(\frac{T}{\delta}\right). \tag{68}$$

**Estimation for B** Term $\mathbf{B}$ is essentially the error brought by the proxy step-size $\boldsymbol{A}_k$. We will first calculate the gap of $1/\sqrt{w_{ij}^{(k)}}$ and $1/\sqrt{a_{ij}^{(k)}}$ as follows,

$$\left| \frac{1}{\sqrt{w_{ij}^{(k)}}} - \frac{1}{\sqrt{a_{ij}^{(k)}}} \right| = \frac{1}{\sqrt{w_{ij}^{(k)}} \sqrt{a_{ij}^{(k)}}} \left| \sqrt{w_{ij}^{(k)}} - \sqrt{a_{ij}^{(k)}} \right| \leq \frac{1}{\sqrt{w_{ij}^{(k)}} \sqrt{a_{ij}^{(k)}}} \sqrt{\left| w_{ij}^{(k)} - a_{ij}^{(k)} \right|}. \tag{69}$$

We then apply (69) and Young's inequality,

$$\begin{aligned} \mathbf{B} &\leq \sum_{k=1}^{t} \sum_{i=1}^{n} \sum_{j=1}^{m} \eta_k \left| \bar{g}_{ij}^{(k)} g_{ij}^{(k)} \right| \left| \frac{1}{\sqrt{w_{ij}^{(k)}}} - \frac{1}{\sqrt{a_{ij}^{(k)}}} \right| \\ &\leq \sum_{k=1}^{t} \sum_{i=1}^{n} \sum_{j=1}^{m} \eta_k \frac{\left| \bar{g}_{ij}^{(k)} g_{ij}^{(k)} \right|}{\sqrt{w_{ij}^{(k)}} \sqrt{a_{ij}^{(k)}}} \sqrt{\left| w_{ij}^{(k)} - a_{ij}^{(k)} \right|} \\ &\leq \frac{1}{4} \sum_{k=1}^{t} \sum_{i=1}^{n} \sum_{j=1}^{m} \eta_k \cdot \frac{\left( \bar{g}_{ij}^{(k)} \right)^2}{\sqrt{a_{ij}^{(k)}}} + 4 \sum_{k=1}^{t} \sum_{i=1}^{n} \sum_{j=1}^{m} \eta_k \cdot \frac{\left| w_{ij}^{(k)} - a_{ij}^{(k)} \right|}{\sqrt{a_{ij}^{(k)}}} \cdot \left( \frac{g_{ij}^{(k)}}{\sqrt{w_{ij}^{(k)}}} \right)^2. \end{aligned} \tag{70}$$

Thus, plugging (57) in Lemma B.7 into (70), we derive that

$$\begin{aligned} \mathbf{B} &\leq \frac{1}{4} \sum_{k=1}^{t} \eta_k \left\| \frac{\bar{\boldsymbol{G}}_k}{\sqrt[4]{\boldsymbol{A}_k}} \right\|_F^2 + 4\sqrt{\mathcal{G}} \sum_{k=1}^{t} \eta_k \sqrt{1 - \beta_{2,k}} \left\| \frac{\boldsymbol{G}_k}{\sqrt{\boldsymbol{W}_k}} \right\|_F^2 \\ &\leq \frac{1}{4} \sum_{k=1}^{t} \eta_k \left\| \frac{\bar{\boldsymbol{G}}_k}{\sqrt[4]{\boldsymbol{A}_k}} \right\|_F^2 + 4\sqrt{\mathcal{G}} \sum_{k=1}^{t} \frac{(\epsilon_2 + \Theta_{\max}) \rho_0}{\sqrt{k}} \sqrt{1 - \beta_{2,k}} \left\| \frac{\boldsymbol{G}_k}{\sqrt{\boldsymbol{W}_k}} \right\|_F^2 \\ &\leq \frac{1}{4} \sum_{k=1}^{t} \eta_k \left\| \frac{\bar{\boldsymbol{G}}_k}{\sqrt[4]{\boldsymbol{A}_k}} \right\|_F^2 + 4\sqrt{\mathcal{G}} \sum_{k=1}^{t} (\epsilon_2 + \Theta_{\max}) \rho_0 (1 - \beta_{2,k}) \left\| \frac{\boldsymbol{G}_k}{\sqrt{\boldsymbol{W}_k}} \right\|_F^2, \end{aligned} \tag{71}$$

where we used (55) in the second inequality and $1/\sqrt{k} \leq 1/k^{c/2}, c \in [1/2, 1]$. Furthermore, using Lemma B.4 and Lemma B.5, we derive that

$$\mathbf{B} \leq \frac{1}{4} \sum_{k=1}^{t} \eta_k \left\| \frac{\bar{\boldsymbol{G}}_k}{\sqrt[4]{\boldsymbol{A}_k}} \right\|_F^2 + \frac{8 m n \mathcal{G}^{\frac{3}{2}} (\epsilon_2 + \Theta_{\max}) \rho_0}{\max\{m,n\} \epsilon_1} \left[ \log\left(2 + \frac{2G^2}{\epsilon_1}\right) + 4 \sum_{k=1}^{t} (1 - \beta_{2,k}) \right]. \tag{72}$$

**Estimating C** Using the similar deduction in (71) and (72), we derive that

$$\mathbf{C} \leq \frac{L m n \mathcal{G} (\epsilon_2 + \Theta_{\max})^2 \rho_0^2}{\max\{m,n\} \epsilon_1} \left[ \log\left(2 + \frac{2G^2}{\epsilon_1}\right) + 4 \sum_{k=1}^{t} (1 - \beta_{2,k}) \right]. \tag{73}$$

**Putting together** We first re-arrange the order in (66) and use $f(\boldsymbol{X}_{t+1}) \geq f^*$ in Assumption (A2) to derive that

$$0 \leq f(\boldsymbol{X}_1) - f^* + \mathbf{A} + \mathbf{B} + \mathbf{C}. \tag{74}$$

We then plug (68), (72), (73) into (74) and set $t = T$, which leads to that with probability at least $1 - \delta$,

$$\frac{1}{2} \sum_{k=1}^{T} \eta_k \left\| \frac{\bar{\boldsymbol{G}}_k}{\sqrt[4]{\boldsymbol{A}_k}} \right\|_F^2 \leq C_1 \log\left(\frac{T}{\delta}\right) + C_2 \sum_{k=1}^{T} (1 - \beta_{2,k}) + C_3, \tag{75}$$

where $C_1, C_2, C_3$ are as in Theorem B.1. Moreover, using Lemma B.3 and (55), we have

$$\frac{1}{2} \sum_{k=1}^{T} \eta_k \left\| \frac{\bar{\boldsymbol{G}}_k}{\sqrt[4]{\boldsymbol{A}_k}} \right\|_F^2 \geq \sum_{k=1}^{T} \frac{\eta_k \left\| \bar{\boldsymbol{G}}_k \right\|_F^2}{2 \max_{i,j} \sqrt{a_{ij}^{(k)}}} \geq \frac{\rho_0 \epsilon_2}{2\sqrt{2\mathcal{G}}} \sum_{k=1}^{T} \frac{\left\| \bar{\boldsymbol{G}}_k \right\|_F^2}{\sqrt{k}}. \tag{76}$$

Combining with (76) and (75), and using $\sum_{k=1}^{T} 1/\sqrt{k} \geq \sqrt{T}$, we derive that

$$\min_{k \in [T]} \|\bar{\boldsymbol{G}}_k\|^2 \leq \frac{C_0}{\sqrt{T}} \left( C_1 \log\left(\frac{T}{\delta}\right) + C_2 \sum_{k=1}^{T} (1 - \beta_{2,k}) + C_3 \right),$$

where $C_0$ has already been defined in (38). We then derive the first desired result that

$$\min_{k \in [T]} \|\bar{\boldsymbol{G}}_k\|^2 \leq \frac{C_0}{\sqrt{T}} \left( C_1 \log\left(\frac{T}{\delta}\right) + C_2 \sum_{k=1}^{T} \frac{1}{k^c} + C_3 \right).$$

**Free dimension bound**  We follow the similar deduction in (71) and use Lemma B.7 to derive that

$$\mathbf{B} \leq \frac{1}{4} \sum_{k=1}^{t} \eta_k \left\| \frac{\bar{\boldsymbol{G}}_k}{\sqrt[4]{\boldsymbol{A}_k}} \right\|_F^2 + 4(G_1 + G_2)(\epsilon_2 + \Theta_{\max})\rho_0 \sum_{k=1}^{t} \frac{1}{k^{c/2+1/2}} \left\| \frac{\boldsymbol{G}_k}{\sqrt{\boldsymbol{W}_k}} \right\|_F^2. \tag{77}$$

Recalling the definition of $w_{ij}^{(k)}$ in (45) and Lemma B.2, we derive that

$$w_{ij}^{(k)} = \frac{R_{\boldsymbol{V}_k}^{(i)} C_{\boldsymbol{V}_k}^{(j)}}{S_{\boldsymbol{V}_k}} \geq \frac{mn\epsilon_1^2}{4\mathcal{G}}, \quad \left\| \frac{\boldsymbol{G}_k}{\sqrt{\boldsymbol{W}}_k} \right\|_F^2 \leq \frac{\|\boldsymbol{G}_k\|_F^2}{\min_{i,j} w_{ij}^{(k)}} \leq \frac{4G^2\mathcal{G}}{mn\epsilon_1^2} \leq G_3, \tag{78}$$

where $G_3$ is as in (40). We thus derive from (77) and (78) that

$$\mathbf{B} \leq \frac{1}{4} \sum_{k=1}^{t} \eta_k \left\| \frac{\bar{\boldsymbol{G}}_k}{\sqrt[4]{\boldsymbol{A}_k}} \right\|_F^2 + 4G_3(G_1 + G_2)(\epsilon_2 + \Theta_{\max})\rho_0 \sum_{k=1}^{t} \frac{1}{k^{c/2+1/2}}. \tag{79}$$

Using (55) and (78), we derive that

$$\mathbf{C} = \sum_{k=1}^{t} \frac{L\eta_k^2}{2} \left\| \frac{\boldsymbol{G}_k}{\sqrt{\boldsymbol{W}}_k} \right\|_F^2 \leq \frac{LG_3(\epsilon_2 + \Theta_{\max})^2 \rho_0^2}{2} \sum_{k=1}^{t} \frac{1}{k}. \tag{80}$$

Plugging the unchanged estimation for $\mathbf{A}$ in (68), (79) and (80) into (66), we have that with probability at least $1 - \delta$, for all $t \in [T]$,

$$\frac{1}{2} \sum_{k=1}^{t} \eta_k \left\| \frac{\bar{\boldsymbol{G}}_k}{\sqrt[4]{\boldsymbol{A}_k}} \right\|_F^2 \leq C_1 \log\left(\frac{T}{\delta}\right) + C_2' \sum_{k=1}^{t} \frac{1}{k^{c/2+1/2}} + C_3' \sum_{k=1}^{t} \frac{1}{k}, \tag{81}$$

where $C_2', C_3'$ are given as in (39) and $C_1$ is as in (37). Further, using Lemma B.3 and the similar deduction for (76),

$$\frac{1}{2} \sum_{k=1}^{t} \eta_k \left\| \frac{\bar{\boldsymbol{G}}_k}{\sqrt[4]{\boldsymbol{A}_k}} \right\|_F^2 \geq \sum_{k=1}^{t} \frac{\eta_k \left\| \bar{\boldsymbol{G}}_k \right\|_F^2}{2 \max_{i,j} \sqrt{a_{ij}^{(k)}}} \geq \frac{1}{C_0'} \sum_{k=1}^{t} \frac{\left\| \bar{\boldsymbol{G}}_k \right\|_F^2}{\sqrt{k}}, \tag{82}$$

where $C_0'$ is as in (39). Combining with (81) and (82), and setting $t = T$, we derive the second desired result in Proposition B.1 that

$$\min_{k \in [T]} \|\bar{\boldsymbol{G}}_k\|^2 \leq \frac{C_0'}{\sqrt{T}} \left( C_1 \log\left(\frac{T}{\delta}\right) + C_2' \sum_{k=1}^{T} \frac{1}{k^{c/2+1/2}} + C_3' \sum_{k=1}^{T} \frac{1}{k} \right).$$

### B.4 Proof of Theorem B.1

Now based on the result in Proposition B.1, we could further derive the final convergence rate. Noting that when $c = 1$, we could bound that

$$\sum_{k=1}^{T} \frac{1}{k} \leq 1 + \int_1^T \frac{1}{x} dx \leq 1 + \log T. \tag{83}$$

Then, we obtain that

$$\min_{k \in [T]} \|\bar{\boldsymbol{G}}_k\|_F^2 \leq \frac{C_0}{\sqrt{T}} \left( C_1 \log \left( \frac{T}{\delta} \right) + C_2 \log T + C_2 + C_3 \right),$$

$$\min_{k \in [T]} \|\bar{\boldsymbol{G}}_k\|_F^2 \leq \frac{C_0'}{\sqrt{T}} \left( C_1 \log \left( \frac{T}{\delta} \right) + (C_2' + C_3') \log T + C_2' + C_3' \right).$$

When $1/2 \leq c < 1$, we have

$$\sum_{k=1}^{T} \frac{1}{k^c} \leq 1 + \int_1^T \frac{1}{x^c} dx \leq 1 + \frac{T^{1-c}}{1-c},$$

$$\sum_{k=1}^{T} \frac{1}{k^{c/2+1/2}} \leq 1 + \int_1^T \frac{1}{x^{c/2+1/2}} dx \leq 1 + \frac{2T^{(1-c)/2}}{1-c}. \tag{84}$$

Then, we obtain that

$$\min_{k \in [T]} \|\bar{\boldsymbol{G}}_k\|_F^2 \leq \frac{C_0}{\sqrt{T}} \left( C_1 \log \left( \frac{T}{\delta} \right) + \frac{C_2}{1-c} \cdot T^{1-c} + C_2 + C_3 \right),$$

$$\min_{k \in [T]} \|\bar{\boldsymbol{G}}_k\|_F^2 \leq \frac{C_0'}{\sqrt{T}} \left( C_1 \log \left( \frac{T}{\delta} \right) + \frac{2C_2'}{1-c} \cdot T^{1-c} + C_3' \log T + C_2' + C_3' \right).$$

## C  An extension to sub-Gaussian noise with bounded gradients

We first recall the sub-Gaussian noise assumption.

**Assumption 1.** *The gradient oracle $g(\boldsymbol{X}, \boldsymbol{Z})$ satisfies that for some constant $\sigma > 0$,*

$$\mathbb{E}\left[\exp\left(\frac{\|g(\boldsymbol{X}, \boldsymbol{Z}) - \nabla f(\boldsymbol{X})\|^2}{\sigma^2}\right) \Big| \boldsymbol{X}\right] \leq \exp(1), \quad \forall \boldsymbol{X} \in \mathbb{R}^{n \times m}.$$

We state a standard concentration inequality for sub-Gaussian noise as follows.

**Lemma C.1.** *Given $T \geq 1$, let the noise sequence $\{\boldsymbol{\xi}_t\}_{t \in [T]}$ where $\boldsymbol{\xi}_t = g(\boldsymbol{X}_t, \boldsymbol{Z}_t) - \nabla f(\boldsymbol{X}_t)$ satisfies Assumption 1. Then, with probability at least $1 - \delta$,*

$$\max_{t \in [T]} \|\boldsymbol{\xi}_t\|^2 \leq \sigma^2 \log \left( \frac{eT}{\delta} \right).$$

*Proof.* See (Li & Orabona, 2020, Lemma 5) for a proof. $\qquad\square$

We also assume that the gradient is bounded, satisfying that $\|\nabla f(\boldsymbol{X})\| \leq G_0, \forall \boldsymbol{X} \in \mathbb{R}^{n \times m}$. Then, we have the following convergence bound.

**Theorem C.1.** *Let $\{\boldsymbol{X}_k\}_{k \geq 1}$ be generated by Algorithm 1 without update clipping where $\eta_k$ is given by (4) for each $k \geq 1$. If Assumptions (A1)-(A3) hold, $\|\nabla f(\boldsymbol{X})\|_F \leq G_0, \forall \boldsymbol{X} \in \mathbb{R}^{n \times m}$, Assumption 1 holds, and*

$$\beta_{2,1} = 1/2, \quad \rho_1 = \rho_0,$$

$$\beta_{2,k} = 1 - 1/k^c, \quad \rho_k = \rho_0/\sqrt{k}, \quad \forall k \geq 2,$$

*for some constants $1/2 \leq c \leq 1, \rho_0 > 0$, then for any $T \geq 1, \delta \in (0, 1)$, with probability at least $1 - 2\delta$,*

$$\min_{k \in [T]} \|\bar{\boldsymbol{G}}_k\|_F^2 \leq \frac{\tilde{C}_0}{\sqrt{T}} \left( \tilde{C}_1 \log\left(\frac{T}{\delta}\right) + \frac{\tilde{C}_2}{1-c} \cdot T^{1-c} + \tilde{C}_2 + \tilde{C}_3 \right),$$

*where we define*

$$\tilde{C}_1 = f(\boldsymbol{X}_1) - f^* + \frac{6\sigma^2(\epsilon_2 + \Theta_{\max})\rho_0}{\sqrt{\epsilon_1}},$$

$\tilde{C}_0, \tilde{C}_2, \tilde{C}_3$ *follow the definitions of $C_0, C_2, C_3$ in (38) with $G, \mathcal{G}$ replaced by $G', \mathcal{G}'$ and*

$$G' = G_0 + \sigma\sqrt{\log\left(\frac{\mathrm{e}T}{\delta}\right)}, \mathcal{G}' = (G')^2 + mn\epsilon_1.$$

The proof begins with the probabilistic estimations and follows the deterministic estimations. We will show the key steps as follows.

## C.1 PROBABILISTIC BOUNDS

We will rely on the definition of sub-Gaussian to estimate the summation of the martingale difference sequence as shown in (67). Letting $\zeta_k = -\eta_k \left\langle \bar{\boldsymbol{G}}_k, \frac{\boldsymbol{\xi}_k}{\sqrt{\boldsymbol{A}_k}} \right\rangle$ and $\omega'_k = \sigma\eta_k \left\| \frac{\bar{\boldsymbol{G}}_k}{\sqrt{\boldsymbol{A}_k}} \right\|_F$, we could derive from Assumption 1 and Cauchy-Schwarz inequality that for any $k \in [T]$,

$$\mathbb{E}\left[ \exp\left(\frac{\zeta_k^2}{(\omega'_k)^2}\right) \mid \mathcal{F}_{k-1} \right] \leq \exp(1).$$

Thereby, relying on Lemma B.1, we derive a similar result to Lemma B.6: with probability at least $1 - \delta$, for all $t \in [T]$,

$$-\sum_{k=1}^{t} \eta_k \left\langle \bar{\boldsymbol{G}}_k, \frac{\boldsymbol{\xi}_k}{\sqrt{\boldsymbol{A}_k}} \right\rangle \leq \frac{3\lambda\sigma^2}{4} \sum_{k=1}^{t} \eta_k^2 \left\| \frac{\bar{\boldsymbol{G}}_k}{\sqrt{\boldsymbol{A}_k}} \right\|_F^2 + \frac{1}{\lambda} \log\left(\frac{T}{\delta}\right). \tag{85}$$

Using (56) where $\eta_k / \sqrt{a_{ij}^{(k)}} \leq 2(\epsilon_2 + \Theta_{\max})\rho_0/\sqrt{\epsilon_1}$, we have

$$-\sum_{k=1}^{t} \eta_k \left\langle \bar{\boldsymbol{G}}_k, \frac{\boldsymbol{\xi}_k}{\sqrt{\boldsymbol{A}_k}} \right\rangle \leq \frac{3\lambda\sigma^2(\epsilon_2 + \Theta_{\max})\rho_0}{2\sqrt{\epsilon_1}} \sum_{k=1}^{t} \eta_k \left\| \frac{\bar{\boldsymbol{G}}_k}{\sqrt[4]{\boldsymbol{A}_k}} \right\|_F^2 + \frac{1}{\lambda} \log\left(\frac{T}{\delta}\right).$$

Setting $\lambda = \sqrt{\epsilon_1}/(6\sigma^2(\epsilon_2 + \Theta_{\max})\rho_0)$, we then derive that with probability at least $1 - \delta$,

$$-\sum_{k=1}^{t} \eta_k \left\langle \bar{\boldsymbol{G}}_k, \frac{\boldsymbol{\xi}_k}{\sqrt{\boldsymbol{A}_k}} \right\rangle \leq \frac{1}{4} \sum_{k=1}^{t} \eta_k \left\| \frac{\bar{\boldsymbol{G}}_k}{\sqrt[4]{\boldsymbol{A}_k}} \right\|_F^2 + \frac{6\sigma^2(\epsilon_2 + \Theta_{\max})\rho_0}{\sqrt{\epsilon_1}} \log\left(\frac{T}{\delta}\right).$$

Relying on the bounded gradient $\|\nabla f(\boldsymbol{X})\| \leq G_0$ and Lemma C.1, we could derive the second probability event: with probability at least $1 - \delta$,

$$\|g(\boldsymbol{X}_t, \boldsymbol{Z}_t)\| \leq G_0 + \sigma\sqrt{\log\left(\frac{\mathrm{e}T}{\delta}\right)}, \quad \forall t \in [T]. \tag{86}$$

where we let $G' = G_0 + \sigma\sqrt{\log\left(\frac{\mathrm{e}T}{\delta}\right)}, \mathcal{G}' = (G')^2 + mn\epsilon_1$.

## C.2 DETERMINISTIC BOUNDS

Then, we will assume both two events, (85) and (86), always happen. Based on the events, stochastic gradients are now bounded with $G'$. Then, recalling (67) and using (85), we derive that

$$\boldsymbol{A} \leq -\frac{3}{4} \sum_{k=1}^{t} \eta_k \left\| \frac{\bar{\boldsymbol{G}}_k}{\sqrt[4]{\boldsymbol{A}_k}} \right\|_F^2 + \frac{6\sigma^2(\epsilon_2 + \Theta_{\max})\rho_0}{\sqrt{\epsilon_1}} \log\left(\frac{T}{\delta}\right). \tag{87}$$

Using the same deduction in Lemma B.7, the gap in the following is now bounded as

$$\frac{\left| w_{ij}^{(k)} - a_{ij}^{(k)} \right|}{\sqrt{a_{ij}^{(k)}}} \leq 4\sqrt{1 - \beta_{2,k}}\sqrt{\mathcal{G}'}.$$

Then, following the same result in (70), we derive that

$$\mathbf{B} \leq \frac{1}{4} \sum_{k=1}^{t} \eta_k \left\| \frac{\bar{\boldsymbol{G}}_k}{\sqrt[4]{\boldsymbol{A}_k}} \right\|_F^2 + 4\sqrt{\mathcal{G}'} \sum_{k=1}^{t} (\epsilon_2 + \Theta_{\max})\rho_0(1 - \beta_{2,k}) \left\| \frac{\boldsymbol{G}_k}{\sqrt{\boldsymbol{W}_k}} \right\|_F^2.$$

Further, using Lemma B.4 and Lemma B.5 with $G, \mathcal{G}$ replaced by $G', \mathcal{G}'$,

$$\mathbf{B} \leq \frac{1}{4} \sum_{k=1}^{t} \eta_k \left\| \frac{\bar{\boldsymbol{G}}_k}{\sqrt[4]{\boldsymbol{A}_k}} \right\|_F^2 + \frac{8mn\mathcal{G}'^{\frac{3}{2}}(\epsilon_2 + \Theta_{\max})\rho_0}{\max\{m,n\}\epsilon_1} \left[ \log\left(2 + \frac{2G'^2}{\epsilon_1}\right) + 4\sum_{k=1}^{t}(1 - \beta_{2,k}) \right]. \tag{88}$$

Similarly, we replace $G, \mathcal{G}$ with $G', \mathcal{G}'$ in (73) and (76), leading to

$$\mathbf{C} \leq \frac{Lmn\mathcal{G}'(\epsilon_2 + \Theta_{\max})^2\rho_0^2}{\max\{m,n\}\epsilon_1} \left[ \log\left(2 + \frac{2G'^2}{\epsilon_1}\right) + 4\sum_{k=1}^{t}(1 - \beta_{2,k}) \right], \tag{89}$$

and

$$\frac{1}{2} \sum_{k=1}^{T} \eta_k \left\| \frac{\bar{\boldsymbol{G}}_k}{\sqrt[4]{\boldsymbol{A}_k}} \right\|_F^2 \geq \frac{\rho_0\epsilon_2}{2\sqrt{2\mathcal{G}'}} \sum_{k=1}^{T} \frac{\|\bar{\boldsymbol{G}}_k\|_F^2}{\sqrt{k}}. \tag{90}$$

As we assume two probability events happen, we then plug (87), (88), (89) and (90) into (66), and use $\beta_{2,k} = 1 - 1/k^c$, leading to with probability at least $1 - 2\delta$,

$$\min_{k \in [T]} \|\bar{\boldsymbol{G}}_k\|_F^2 \leq \frac{\tilde{C}_0}{\sqrt{T}} \left( \tilde{C}_1 \log\left(\frac{T}{\delta}\right) + \tilde{C}_2 \sum_{k=1}^{T} \frac{1}{k^c} + \tilde{C}_3 \right).$$

Finally, we shall estimate $\sum_{k=1}^{T} 1/k^c$ following the same deduction in Appendix B.4, which leads to the desired convergence bounds as follows:

$$\min_{k \in [T]} \|\bar{\boldsymbol{G}}_k\|_F^2 \leq \frac{\tilde{C}_0}{\sqrt{T}} \left( \tilde{C}_1 \log\left(\frac{T}{\delta}\right) + \frac{\tilde{C}_2}{1 - c} \cdot T^{1-c} + \tilde{C}_2 + \tilde{C}_3 \right).$$

# D    PROOF DETAIL FOR STOCHASTIC ADAFACTOR WITH UPDATE CLIPPING

We first provide the detailed version of Theorem 7.1 as follows.

**Theorem D.1.** *Let $\{\boldsymbol{X}_k\}_{k\geq 1}$ be the sequence generated by Algorithm 1 with (6). If Assumptions (A1)-(A4) hold, and*

$$\rho_k = \rho_0/\sqrt{k}, \quad d_k = k^{\frac{c}{2(\alpha-1)}}, \quad \forall k \geq 1,$$
$$\beta_{2,1} = 1/2, \quad \beta_{2,k} = 1 - 1/k^c, \forall k \geq 2.$$

*When $c = 1$, with probability at least $1 - \delta$,*

$$\min_{k \in [T]} \|\bar{\boldsymbol{G}}_k\|_F^2 \leq \frac{D_0}{\sqrt{T}} \left( C_1 \log\left(\frac{T}{\delta}\right) + (C_2 + D_1(\alpha))\log T + C_2 + D_1(\alpha) + C_3 \right), \tag{91}$$

$$\min_{k \in [T]} \|\bar{\boldsymbol{G}}_k\|_F^2 \leq \frac{D_0}{\sqrt{T}} \left( C_1 \log\left(\frac{T}{\delta}\right) + (C_2' + C_3' + D_1(\alpha))\log T + C_2' + C_3' + D_1(\alpha) \right). \tag{92}$$

*When $1/2 \leq c < 1$, with probability at least $1 - \delta$,*

$$\min_{k \in [T]} \|\bar{\boldsymbol{G}}_k\|_F^2 \leq \frac{D_0}{\sqrt{T}} \left( C_1 \log \left( \frac{T}{\delta} \right) + \frac{C_2 + D_1(\alpha)}{1 - c} \cdot T^{1-c} + C_2 + D_1(\alpha) + C_3 \right), \tag{93}$$

$$\min_{k \in [T]} \|\bar{\boldsymbol{G}}_k\|_F^2 \leq \frac{D_0}{\sqrt{T}} \left( C_1 \log \left( \frac{T}{\delta} \right) + C_3' \log T + \frac{2(C_2' + D_1(\alpha))}{1 - c} \cdot T^{\frac{1-c}{2}} + C_2' + C_3' + D_1(\alpha) \right), \tag{94}$$

*where $C_1, C_2, C_3, C_2', C_3'$ are as in Theorem B.1 and*

$$D_0 = \min\{C_0, C_0'\}, \quad D_1(\alpha) = \frac{G^{1+\alpha} G_4^{1-\alpha} \sqrt{\mathcal{G}}(\epsilon_2 + \Theta_{\max})\rho_0}{\sqrt{mn}\epsilon_1}, \quad G_4 = \frac{mn\epsilon_1}{2\sqrt{\mathcal{G}}}. \tag{95}$$

**Calculation of hyper-parameters' dependency** We first calculate the dependency on $m, n, \epsilon_1, \alpha$ in the additional coefficient $D_1(\alpha)$ as follows,

$$D_1(\alpha) \sim \mathcal{O} \left( \left( \frac{\sqrt{1 + mn\epsilon_1}}{mn\epsilon_1} \right)^{\alpha-1} \sqrt{\frac{1}{mn\epsilon_1^2} + \frac{1}{\epsilon_1}} \right), \tag{96}$$

which is free of the curse of dimension since $mn$ exists in the denominator. Recalling the definitions of $C_0', C_1, C_2', C_3'$ in (37) and (39), it's easy to verify that these coefficients are also free of the curse of dimension factor $m, n$ since $m, n$ exist in the denominator. Thereby, we also derive a free dimension bound selecting (92) and (94).

To calculate the dependency on $\epsilon_1$, we could combine with (41) and (96) to derive that

$$C_0 D_1(\alpha) \sim \mathcal{O}\left( \epsilon_1^{-\alpha} \right), \quad C_0 C_1 \sim \mathcal{O}\left( 1/\epsilon_1^{-1/2} \right), \quad C_0 C_3 \sim \mathcal{O}\left( \epsilon_1^{-1} \log(1/\epsilon_1) \right).$$

Thereby, selecting the bounds in (91) and (93) and noting that $\alpha > 1$, we derive that the order on $\epsilon_1$ is

$$\mathcal{O} \left( \frac{1}{\epsilon_1^\alpha} \log \left( \frac{1}{\epsilon_1} \right) \right).$$

Moreover, it's clear to reveal that there exists $mn$ in the denominator, which could improve the dependency on $\epsilon_1$. If we suppose that $mn$ is comparable to $\epsilon_1$, then we derive that $C_0 D_1(\alpha) \sim \mathcal{O}(\epsilon_1^{-1/2})$ and the order on $\epsilon_1$ is

$$\mathcal{O} \left( \frac{1}{\epsilon_1} \log \left( \frac{1}{\epsilon_1} \right) \right).$$

### D.1 Proof of Theorem D.1

We define

$$\tilde{\boldsymbol{G}}_k = \frac{\boldsymbol{G}_k}{\max\{1, \|\boldsymbol{U}_k\|_F/(d_k\sqrt{mn})\}}, \quad \hat{\rho}_k = \max\{\epsilon_2, \mathrm{RMS}(\boldsymbol{X}_k)\}\rho_k. \tag{97}$$

Since $\mathrm{RMS}(\boldsymbol{U}_k) = \|\boldsymbol{U}_k\|_F/\sqrt{mn}$, $\mathrm{RMS}(\boldsymbol{X}_k) \leq \Theta_{\max}$, we derive that

$$\boldsymbol{X}_{k+1} = \boldsymbol{X}_k - \hat{\rho}_k \frac{\tilde{\boldsymbol{G}}_k}{\sqrt{\boldsymbol{W}_k}},$$

$$\frac{\epsilon_2 \rho_0}{\sqrt{k}} \leq \hat{\rho}_k \leq \frac{(\epsilon_2 + \Theta_{\max})\rho_0}{\sqrt{k}} \leq (\epsilon_2 + \Theta_{\max})\rho_0 \sqrt{1 - \beta_{2,k}}, \tag{98}$$

where we applied that $1/\sqrt{k} \leq 1/k^{c/2}, c \in [1/2, 1]$ and $\beta_{2,k} = 1 - 1/k^c$ in the last inequality. Using the inequalities in (15) and (98), we have

$$f(\boldsymbol{X}_{k+1}) \leq f(\boldsymbol{X}_k) + \langle \bar{\boldsymbol{G}}_k, \boldsymbol{X}_{k+1} - \boldsymbol{X}_k \rangle + \frac{L}{2} \|\boldsymbol{X}_{k+1} - \boldsymbol{X}_k\|_F^2$$

$$\leq f(\boldsymbol{X}_k) - \hat{\rho}_k \left\langle \bar{\boldsymbol{G}}_k, \frac{\tilde{\boldsymbol{G}}_k}{\sqrt{\boldsymbol{W}_k}} \right\rangle + \frac{L\hat{\rho}_k^2}{2} \left\| \frac{\tilde{\boldsymbol{G}}_k}{\sqrt{\boldsymbol{W}_k}} \right\|_F^2.$$

Summing up both sides over $k \in [t]$ and using $f(\boldsymbol{X}_{t+1}) \geq f^*$ from Assumption (A2), we derive that

$$0 \leq f(\boldsymbol{X}_1) - f^* + \underbrace{\sum_{k=1}^{t} -\hat{\rho}_k \left\langle \bar{\boldsymbol{G}}_k, \frac{\tilde{\boldsymbol{G}}_k}{\sqrt{\boldsymbol{W}_k}} \right\rangle}_{\mathbf{D}} + \underbrace{\sum_{k=1}^{t} \frac{L\hat{\rho}_k^2}{2} \left\| \frac{\tilde{\boldsymbol{G}}_k}{\sqrt{\boldsymbol{W}_k}} \right\|_F^2}_{\mathbf{E}}. \tag{99}$$

Introducing $\boldsymbol{A}_k$ in (46), we further have the following decomposition,

$$\mathbf{D} = -\sum_{k=1}^{t} \hat{\rho}_k \left\langle \bar{\boldsymbol{G}}_k, \frac{\tilde{\boldsymbol{G}}_k}{\sqrt{\boldsymbol{A}_k}} \right\rangle + \underbrace{\sum_{k=1}^{t} \hat{\rho}_k \left\langle \bar{\boldsymbol{G}}_k, \left( \frac{1}{\sqrt{\boldsymbol{A}_k}} - \frac{1}{\sqrt{\boldsymbol{W}_k}} \right) \odot \tilde{\boldsymbol{G}}_k \right\rangle}_{\mathbf{D.1}}$$

$$= -\sum_{k=1}^{t} \hat{\rho}_k \left\| \frac{\bar{\boldsymbol{G}}_k}{\sqrt[4]{\boldsymbol{A}_k}} \right\|_F^2 + \mathbf{D.1}$$

$$\underbrace{-\sum_{k=1}^{t} \hat{\rho}_k \left\langle \bar{\boldsymbol{G}}_k, \frac{\tilde{\boldsymbol{G}}_k}{\sqrt{\boldsymbol{A}_k}} - \mathbb{E}_{\boldsymbol{Z}_k}\left[ \frac{\tilde{\boldsymbol{G}}_k}{\sqrt{\boldsymbol{A}_k}} \right] \right\rangle}_{\mathbf{D.2}} + \underbrace{\sum_{k=1}^{t} \hat{\rho}_k \left\langle \bar{\boldsymbol{G}}_k, \frac{\tilde{\boldsymbol{G}}_k}{\sqrt{\boldsymbol{A}_k}} - \mathbb{E}_{\boldsymbol{Z}_k}\left[ \frac{\tilde{\boldsymbol{G}}_k}{\sqrt{\boldsymbol{A}_k}} \right] \right\rangle}_{\mathbf{D.3}}. \tag{100}$$

**Estimating E** Hence, using (97), (98), Lemma B.4 and Lemma B.5, we derive that

$$\mathbf{E} \leq \frac{L}{2} \sum_{k=1}^{t} \hat{\rho}_k^2 \left\| \frac{\boldsymbol{G}_k}{\sqrt{\boldsymbol{W}_k}} \right\|_F^2 \leq \frac{L(\epsilon_2 + \Theta_{\max})^2 \rho_0^2}{2} \sum_{k=1}^{t} (1 - \beta_{2,k}) \left\| \frac{\boldsymbol{G}_k}{\sqrt{\boldsymbol{W}_k}} \right\|_F^2$$

$$\leq \frac{Lmn\mathcal{G}(\epsilon_2 + \Theta_{\max})^2 \rho_0^2}{\max\{m,n\}\epsilon_1} \left[ \log\left(2 + \frac{2G^2}{\epsilon_1}\right) + 4\sum_{k=1}^{t} (1 - \beta_{2,k}) \right]. \tag{101}$$

To avoid the curse of dimension, we drive from (97) and (78) that

$$\left\| \frac{\tilde{\boldsymbol{G}}_k}{\sqrt{\boldsymbol{W}_k}} \right\|_F^2 = \frac{1}{(\max\{1, \|\boldsymbol{U}_k\|_F/(d_k\sqrt{mn})\})^2} \left\| \frac{\boldsymbol{G}_k}{\sqrt{\boldsymbol{W}_k}} \right\|_F^2 \leq \left\| \frac{\boldsymbol{G}_k}{\sqrt{\boldsymbol{W}_k}} \right\|_F^2 \leq G_3. \tag{102}$$

Then, using (98) and (102), we derive that

$$\mathbf{E} \leq \frac{LG_3(\epsilon_2 + \Theta_{\max})^2 \rho_0^2}{2} \sum_{k=1}^{t} \frac{1}{k}. \tag{103}$$

**Estimating D.1** We could follow the similar deduction in (69) and (70) to derive that

$$\mathbf{D.1} \leq \sum_{k=1}^{t} \sum_{i=1}^{n} \sum_{j=1}^{m} \hat{\rho}_k |\bar{g}_{ij}^{(k)} \tilde{g}_{ij}^{(k)}| \left| \frac{1}{\sqrt{w_{ij}^{(k)}}} - \frac{1}{\sqrt{a_{ij}^{(k)}}} \right|$$

$$\leq \sum_{k=1}^{t} \sum_{i=1}^{n} \sum_{j=1}^{m} \hat{\rho}_k \frac{|\bar{g}_{ij}^{(k)} \tilde{g}_{ij}^{(k)}|}{\sqrt{w_{ij}^{(k)}} \sqrt{a_{ij}^{(k)}}} \sqrt{\left| w_{ij}^{(k)} - a_{ij}^{(k)} \right|}$$

$$\leq \frac{1}{4} \sum_{k=1}^{t} \sum_{i=1}^{n} \sum_{j=1}^{m} \hat{\rho}_k \cdot \frac{\left(\bar{g}_{ij}^{(k)}\right)^2}{\sqrt{a_{ij}^{(k)}}} + 4 \sum_{k=1}^{t} \sum_{i=1}^{n} \sum_{j=1}^{m} \hat{\rho}_k \cdot \frac{\left| w_{ij}^{(k)} - a_{ij}^{(k)} \right|}{\sqrt{a_{ij}^{(k)}}} \cdot \left( \frac{\tilde{g}_{ij}^{(k)}}{\sqrt{w_{ij}^{(k)}}} \right)^2. \tag{104}$$

Using Lemma B.7 and (104), we further derive that

$$\mathbf{D.1} \leq \frac{1}{4} \sum_{k=1}^{t} \hat{\rho}_k \left\| \frac{\bar{\boldsymbol{G}}_k}{\sqrt[4]{\boldsymbol{A}_k}} \right\|_F^2 + 4\sqrt{\mathcal{G}} \sum_{k=1}^{t} \hat{\rho}_k \sqrt{1 - \beta_{2,k}} \left\| \frac{\tilde{\boldsymbol{G}}_k}{\sqrt{\boldsymbol{W}_k}} \right\|_F^2$$

$$\leq \frac{1}{4} \sum_{k=1}^{t} \hat{\rho}_k \left\| \frac{\bar{\boldsymbol{G}}_k}{\sqrt[4]{\boldsymbol{A}_k}} \right\|_F^2 + 4\sqrt{\mathcal{G}} \sum_{k=1}^{t} \hat{\rho}_k \sqrt{1 - \beta_{2,k}} \left\| \frac{\boldsymbol{G}_k}{\sqrt{\boldsymbol{W}_k}} \right\|_F^2.$$

Using (98), Lemma B.4 and Lemma B.5, we further have

$$\mathbf{D.1} \leq \frac{1}{4} \sum_{k=1}^{t} \hat{\rho}_k \left\| \frac{\bar{\boldsymbol{G}}_k}{\sqrt[4]{\boldsymbol{A}_k}} \right\|_F^2 + 4\sqrt{\mathcal{G}}(\epsilon_2 + \Theta_{\max})\rho_0 \sum_{k=1}^{t} (1 - \beta_{2,k}) \left\| \frac{\boldsymbol{G}_k}{\sqrt{\boldsymbol{W}_k}} \right\|_F^2$$

$$\leq \frac{1}{4} \sum_{k=1}^{t} \hat{\rho}_k \left\| \frac{\bar{\boldsymbol{G}}_k}{\sqrt[4]{\boldsymbol{A}_k}} \right\|_F^2 + \frac{8mn\mathcal{G}^{\frac{3}{2}}(\epsilon_2 + \Theta_{\max})\rho_0}{\max\{m,n\}\epsilon_1} \left[ \log\left(2 + \frac{2G^2}{\epsilon_1}\right) + 4\sum_{k=1}^{t}(1 - \beta_{2,k}) \right].$$
$$(105)$$

To avoid the curse of dimension, we apply Lemma B.7, (98) and (78) to derive that

$$\mathbf{D.1} \leq \frac{1}{4} \sum_{k=1}^{t} \hat{\rho}_k \left\| \frac{\bar{\boldsymbol{G}}_k}{\sqrt[4]{\boldsymbol{A}_k}} \right\|_F^2 + 4(G_1 + G_2) \sum_{k=1}^{t} \hat{\rho}_k \sqrt{1 - \beta_{2,k}} \left\| \frac{\boldsymbol{G}_k}{\sqrt{\boldsymbol{W}_k}} \right\|_F^2$$

$$\leq \frac{1}{4} \sum_{k=1}^{t} \hat{\rho}_k \left\| \frac{\bar{\boldsymbol{G}}_k}{\sqrt[4]{\boldsymbol{A}_k}} \right\|_F^2 + 4(G_1 + G_2)(\epsilon_2 + \Theta_{\max})\rho_0 \sum_{k=1}^{t} \frac{1}{k^{c/2+1/2}} \left\| \frac{\boldsymbol{G}_k}{\sqrt{\boldsymbol{W}_k}} \right\|_F^2$$

$$\leq \frac{1}{4} \sum_{k=1}^{t} \hat{\rho}_k \left\| \frac{\bar{\boldsymbol{G}}_k}{\sqrt[4]{\boldsymbol{A}_k}} \right\|_F^2 + 4G_3(G_1 + G_2)(\epsilon_2 + \Theta_{\max})\rho_0 \sum_{k=1}^{t} \frac{1}{k^{c/2+1/2}}.$$
$$(106)$$

**Estimating D.2** Since $\boldsymbol{A}_k$ is independent from $\boldsymbol{Z}_k$, it further leads to

$$\mathbf{D.2} = -\sum_{k=1}^{t} \hat{\rho}_k \left\langle \frac{\bar{\boldsymbol{G}}_k}{\sqrt{\boldsymbol{A}_k}}, \tilde{\boldsymbol{G}}_k - \mathbb{E}_{\boldsymbol{Z}_k}\left[\tilde{\boldsymbol{G}}_k\right] \right\rangle.$$

Then, the deduction for estimating **D.2** follows the similar idea as in Lemma B.6, relying on a martingale difference sequence.

Let us set $\varphi_k = -\hat{\rho}_k \left\langle \frac{\bar{\boldsymbol{G}}_k}{\sqrt{\boldsymbol{A}_k}}, \tilde{\boldsymbol{G}}_k - \mathbb{E}_{\boldsymbol{Z}_k}\left[\tilde{\boldsymbol{G}}_k\right] \right\rangle$ and the filtration $\mathcal{F}_k = \sigma\left(\boldsymbol{Z}_1, \cdots, \boldsymbol{Z}_k\right)$. Noting that $\hat{\rho}_k, \bar{\boldsymbol{G}}_k$ and $\boldsymbol{A}_k$ are dependent by $\mathcal{F}_{k-1}$. Since $\boldsymbol{\xi}_k$ is dependent by $\mathcal{F}_k$, we could prove that $\{\varphi_k\}_{k\geq 1}$ is a martingale difference sequence by showing that

$$\mathbb{E}\left[\varphi_k \mid \mathcal{F}_{k-1}\right] = -\hat{\rho}_k \left\langle \frac{\bar{\boldsymbol{G}}_k}{\sqrt{\boldsymbol{A}_k}}, \mathbb{E}_{\boldsymbol{Z}_k}\left[\tilde{\boldsymbol{G}}_k - \mathbb{E}_{\boldsymbol{Z}_k}[\tilde{\boldsymbol{G}}_k]\right] \right\rangle = 0.$$

In addition, using Assumptions (A3), (A4) and Jensen's inequality, we have

$$\|\tilde{\boldsymbol{G}}_k\|_F = \frac{\|\boldsymbol{G}_k\|_F}{\max\{1, \|\boldsymbol{U}_k\|/(d_k\sqrt{mn})\}} \leq \|\boldsymbol{G}_k\|_F \leq G, \quad \|\mathbb{E}_{\boldsymbol{Z}_k}[\tilde{\boldsymbol{G}}_k]\|_F \leq \mathbb{E}_{\boldsymbol{Z}_k}\|\tilde{\boldsymbol{G}}_k\|_F \leq G.$$

Therefore, we derive that

$$\|\tilde{\boldsymbol{G}}_k - \mathbb{E}_{\boldsymbol{Z}_k}[\tilde{\boldsymbol{G}}_k]\|_F \leq \|\tilde{\boldsymbol{G}}_k\|_F + \|\mathbb{E}_{\boldsymbol{Z}_k}[\tilde{\boldsymbol{G}}_k]\|_F \leq 2G. \tag{107}$$

Let $\omega_k' = 2G\hat{\rho}_k \left\| \frac{\bar{\boldsymbol{G}}_k}{\sqrt{\boldsymbol{A}_k}} \right\|_F$. We thus derive from the Cauchy-Schwarz inequality and (107) that

$$\mathbb{E}\left[\exp\left(\frac{\varphi_k^2}{(\omega_k')^2}\right) \mid \mathcal{F}_{k-1}\right] \leq \mathbb{E}\left[\exp\left(\frac{\left\|\frac{\bar{\boldsymbol{G}}_k}{\sqrt{\boldsymbol{A}_k}}\right\|_F^2 \|\tilde{\boldsymbol{G}}_k - \mathbb{E}_{\boldsymbol{Z}_k}[\tilde{\boldsymbol{G}}_k]\|_F^2}{4G^2 \left\|\frac{\bar{\boldsymbol{G}}_k}{\sqrt{\boldsymbol{A}_k}}\right\|_F^2}\right) \mid \mathcal{F}_{k-1}\right] \leq \exp(1).$$

Then, using Lemma B.1, it leads to that for any $\lambda > 0$, with probability at least $1 - \delta$,

$$\mathbf{D.2} = \sum_{k=1}^{t} \varphi_k \leq 3\lambda G^2 \sum_{k=1}^{t} \hat{\rho}_k^2 \left\| \frac{\bar{\boldsymbol{G}}_k}{\sqrt{\boldsymbol{A}_k}} \right\|_F^2 + \frac{1}{\lambda} \log\left(\frac{1}{\delta}\right)$$

$$= 3\lambda G^2 \sum_{k=1}^{t} \sum_{i=1}^{n} \sum_{j=1}^{m} \frac{\hat{\rho}_k}{\sqrt{a_{ij}^{(k)}}} \cdot \hat{\rho}_k \frac{\left(\bar{g}_{ij}^{(k)}\right)^2}{\sqrt{a_{ij}^{(k)}}} + \frac{1}{\lambda} \log\left(\frac{1}{\delta}\right).$$

Since $\{\beta_{2,k}\}_{k\geq 2}$ is non-decreasing, we could apply Lemma B.3 to derive that

$$\frac{1}{\sqrt{a_{ij}^{(k)}}} \leq \sqrt{\frac{1}{\beta_{2,k}(1-\beta_{2,k})\epsilon_1}} \leq \sqrt{\frac{1}{\min\{\beta_{2,1},\beta_{2,2}\}(1-\beta_{2,k})\epsilon_1}} \leq \frac{2}{\sqrt{(1-\beta_{2,k})\epsilon_1}}.$$

Then, we apply (98), and re-scale $\delta$ to obtain that for any $\lambda > 0$, with probability at least $1 - \delta$, for all $t \in [T]$,

$$\mathbf{D.2} \leq \frac{6\lambda G^2 \rho_0(\epsilon_2 + \Theta_{\max})}{\sqrt{\epsilon_1}} \sum_{k=1}^t \hat{\rho}_k \left\|\frac{\bar{G}_k}{\sqrt[4]{A_k}}\right\|_F^2 + \frac{1}{\lambda}\log\left(\frac{T}{\delta}\right).$$

Setting $\lambda = \sqrt{\epsilon_1}/(24G^2\rho_0(\epsilon_2 + \Theta_{\max}))$, we derive that

$$\mathbf{D.2} \leq \frac{1}{4}\sum_{k=1}^t \hat{\rho}_k \left\|\frac{\bar{G}_k}{\sqrt[4]{A_k}}\right\|_F^2 + \frac{24G^2\rho_0(\epsilon_2 + \Theta_{\max})}{\sqrt{\epsilon_1}}\log\left(\frac{T}{\delta}\right). \tag{108}$$

**Estimating D.3** First, since $A_k$ is independent from $Z_k$ and $\mathbb{E}_{Z_k}[G_k] = \bar{G}_k$, we have

$$\mathbf{D.3} = \sum_{k=1}^t \hat{\rho}_k \left\langle \bar{G}_k, \frac{\mathbb{E}_{Z_k}[G_k]}{\sqrt{A_k}} - \frac{\mathbb{E}_{Z_k}[\tilde{G}_k]}{\sqrt{A_k}} \right\rangle$$

$$\leq \sum_{k=1}^t \hat{\rho}_k \left\|\frac{\bar{G}_k}{\sqrt{A_k}}\right\|_F \cdot \left\|\mathbb{E}_{Z_k}\underbrace{\left[G_k - \frac{G_k}{\max\{1, \|U_k\|_F/(d_k\sqrt{mn})\}}\right]}_{\Omega_k}\right\|_F. \tag{109}$$

We define the random variable $S_k^{(1)}$, $S_k^{(2)}$ and $\tilde{S}_k^{(1)}$ using the indicator function $\chi$ and $G_4$ in (95) as follows,

$$S_k^{(1)} = \chi_{\{\|U_k\|_F > d_k\sqrt{mn}\}}, \quad S_k^{(2)} = \chi_{\{\|U_k\|_F \leq d_k\sqrt{mn}\}}, \quad \tilde{S}_k^{(1)} = \chi_{\{\|G_k\|_F \geq d_k G_4\}}.$$

From (78), we derive that

$$\|U_k\|_F \leq \|G_k\|_F \cdot \frac{2\sqrt{\mathcal{G}}}{\sqrt{mn\epsilon_1}}.$$

Hence, $S_k^{(1)} \leq \tilde{S}_k^{(1)}, \forall k \geq 1$. Note that when $S_k^{(2)} = 1$, it's equivalent to $\Omega_k = 0$. Then, we derive that

$$\|\mathbb{E}_{Z_k}[\Omega_k]\|_F = \left\|\mathbb{E}_{Z_k}[\Omega_k S_k^{(1)}] + \mathbb{E}_{Z_k}[\Omega_k S_k^{(2)}]\right\|_F = \left\|\mathbb{E}_{Z_k}[\Omega_k S_k^{(1)}]\right\|_F$$

$$\leq \mathbb{E}_{Z_k}\left\|\Omega_k S_k^{(1)}\right\|_F \leq \mathbb{E}_{Z_k}\left\|\Omega_k \tilde{S}_k^{(1)}\right\|_F \leq \mathbb{E}_{Z_k}\left\|G_k \tilde{S}_k^{(1)}\right\|_F \leq G^\alpha(d_k G_4)^{1-\alpha}, \tag{110}$$

Furthermore, we use Assumption (A4) and Lemma B.2 to derive a lower bound for $a_{ij}^{(k)}$ where

$$a_{ij}^{(k)} \geq \frac{mn\epsilon_1^2}{4\mathcal{G}}, \quad \left\|\frac{\bar{G}_k}{\sqrt{A_k}}\right\|_F \leq \frac{\|\bar{G}_k\|_F}{\min_{i,j}\sqrt{a_{ij}^{(k)}}} \leq \frac{2G\sqrt{\mathcal{G}}}{\sqrt{mn}\epsilon_1}. \tag{111}$$

Combining with (98), (109), (110) and (111), we thus derive that

$$\mathbf{D.3} \leq \frac{2G^{1+\alpha}G_4^{1-\alpha}\sqrt{\mathcal{G}}(\epsilon_2 + \Theta_{\max})\rho_0}{\sqrt{mn}\epsilon_1}\sum_{k=1}^t \frac{1}{d_k^{\alpha-1}\sqrt{k}}. \tag{112}$$

**Putting together** Both **E** and **D.1** are bounded with two estimations, one of which owns a better dependency to $1/\epsilon_1$ and the other avoids the curse of the dimension. We thereby derive two results.

Plugging (105), (108) and (112) into (100) and then combining with (101) and (99), we then derive that with probability at least $1 - \delta$, for all $t \in [T]$,

$$\frac{1}{2} \sum_{k=1}^{t} \hat{\rho}_k \left\| \frac{\bar{G}_k}{\sqrt[4]{A_k}} \right\|_F^2 \leq C_1 \log\left(\frac{T}{\delta}\right) + C_2 \sum_{k=1}^{t} (1 - \beta_{2,k}) + C_3 + D_1(\alpha) \sum_{k=1}^{t} \frac{1}{d_k^{\alpha-1}\sqrt{k}}, \quad (113)$$

where $C_1, C_2, C_3$ are as in Theorem B.1 and $D_1(\alpha)$ is as in (95). Plugging (106), (108) and (112) into (100), then combining with (103) and (99), we then derive that with probability at least $1 - \delta$, for all $t \in [T]$,

$$\frac{1}{2} \sum_{k=1}^{t} \hat{\rho}_k \left\| \frac{\bar{G}_k}{\sqrt[4]{A_k}} \right\|_F^2 \leq C_1 \log\left(\frac{T}{\delta}\right) + C_2' \sum_{k=1}^{t} \frac{1}{k^{c/2+1/2}} + C_3' \sum_{k=1}^{t} \frac{1}{k} + D_1(\alpha) \sum_{k=1}^{t} \frac{1}{d_k^{\alpha-1}\sqrt{k}}. \tag{114}$$

where $C_2', C_3'$ are as in Theorem B.1. Moreover, using (98), we reveal that the lower bound for $\hat{\rho}_k$ is the same the one for $\eta_k$ in (55). Thereby, following the same deduction in (76) and (81), we derive that

$$\frac{1}{2} \sum_{k=1}^{T} \hat{\rho}_k \left\| \frac{\bar{G}_k}{\sqrt[4]{A_k}} \right\|_F^2 \geq \sum_{k=1}^{T} \frac{\hat{\rho}_k}{2} \frac{\|\bar{G}_k\|_F^2}{\max_{i,j} \sqrt{a_{ij}^{(k)}}} \geq \frac{1}{D_0} \sum_{k=1}^{T} \frac{1}{\sqrt{k}} \|\bar{G}_k\|_F^2, \quad (115)$$

where $D_0 = \min\{C_0, C_0'\}$ that has been defined in (95). Setting $t = T$ on (113) and (114), and then using (115), we then derive that

$$\min_{t \in [T]} \|\bar{G}_k\|_F^2 \leq \frac{D_0}{\sum_{t=1}^{T} 1/\sqrt{k}} \left( C_1 \log\left(\frac{T}{\delta}\right) + C_2 \sum_{k=1}^{t} (1 - \beta_{2,k}) + C_3 + D_1(\alpha) \sum_{k=1}^{t} \frac{1}{d_k^{\alpha-1}\sqrt{k}} \right),$$

$$\min_{t \in [T]} \|\bar{G}_k\|_F^2 \leq \frac{D_0}{\sum_{t=1}^{T} 1/\sqrt{k}} \left( C_1 \log\left(\frac{T}{\delta}\right) + C_2' \sum_{k=1}^{t} \frac{1}{k^{(c+1)/2}} + C_3' \sum_{k=1}^{t} \frac{1}{k} + D_1(\alpha) \sum_{k=1}^{t} \frac{1}{d_k^{\alpha-1}\sqrt{k}} \right).$$

Then, using the results in (83) and (84), we could derive the desired result in Theorem D.1.

# E  SOME COMPLEMENTARY EXPERIMENTS

## E.1  TEST ACCURACY OF TRAINING BERT-BASE MODEL

First, we report the test accuracy for the experiment in Section 9, using Adafactor with different $c$ and Adam to train BERT-Base.

Table 1: The test accuracy after 5 epochs. We use Adafactor and Adam to train BERT-Base on GLUE/MNLI dataset. All the setup is aligned with the one in Figure 1.

|  | $c = 0.5$ | $c = 0.6$ | $c = 0.8$ | $c = 0.9$ | $c = 1.0$ | Adam |
|---|---|---|---|---|---|---|
| accuracy | 0.7785 | 0.7803 | 0.7795 | 0.7827 | 0.7802 | 0.8014 |

Table 1 implies that the performance of Adafactor and Adam is comparable. It's also reasonable that Adafactor sacrifices some accuracy as the memory is saved in comparison to Adam.

## E.2  EXPERIMENTS ON RESNET MODEL

In the following experiments, the initialization is $R_0 = 0_m$ and $C_0 = 0_n^\top$. We use a learning rate with the warm-up technique as described in (Shazeer & Stern, 2018), specifically $\rho_k = \min\{10^{-6} \cdot k, 1/\sqrt{k}\}$ for all experiments unless otherwise specified. The batch size is set to 256, and the total number of epochs is 400 by default. Our models are ResNet-20 and ResNet-110 (He et al., 2016), and we use the CIFAR-10 and CIFAR-100 datasets (Krizhevsky et al., 2009) without any data augmentation. The experiments are conducted using the PyTorch implementation of Adafactor on a single NVIDIA GeForce RTX 4090 GPU.

### E.3 REPORT ON EXPERIMENT 1

We train ResNet-20 and ResNet-110 using Adafactor (no update clipping) with decay rate parameter $c$ ranging from 0.5 to 1.0 in increments of 0.05, while keeping other hyper-parameters at their default values. Each experiment is run 10 times with 100 epochs, and we plot the average training curve and the average test accuracy with standard deviation (shallow blue region) in Figure 2 and Figure 3, respectively. The training curves under different decay rates $c$ are not obviously different. Hence, we turn to use the test accuracy as the measurement. Figure 3 indicates that $c = 1.0$ yields better test performance and stability compared to $c < 1.0$ on different models and datasets, corresponding to the highest test accuracy and thinner shallow blue band. These performances align roughly with the results in Theorem 6.1.

### E.4 REPORT ON EXPERIMENT 2

Table 2: The test accuracy after 400 epochs. We use Adafactor without update clipping under different $\epsilon_1$ and other hyper-parameters are set by default.

| $\epsilon_1$ | ResNet 20 / CIFAR 10 | ResNet 20 / CIFAR 100 | ResNet 110 / CIFAR 100 |
|---|---|---|---|
| $10^{-30}$ | 0.7526 | 0.4072 | 0.4159 |
| $10^{-15}$ | 0.7439 | 0.3936 | 0.4288 |
| $10^{-8}$ | 0.7425 | 0.4157 | 0.4266 |
| $10^{-5}$ | 0.7480 | 0.4141 | 0.3951 |
| $10^{-3}$ | 0.6864 | 0.3247 | 0.3377 |

In the second experiment, we test Adafactor (no update clipping) under different $\epsilon_1$ values. We plot the training loss curve against the step $t$ on different models and datasets in Figure 4. We also report the test accuracy after training 400 epochs in Table 2. The performance for $\epsilon_1 = 10^{-8}$ and $\epsilon_1 = 10^{-5}$ is nearly identical to that for $\epsilon_1 = 10^{-30}$. Moreover, even a larger value of $10^{-3}$ achieves comparable training performance, though with a slower decrease in loss and a worse test accuracy compared to other values of $\epsilon_1$. Notably, $\epsilon_1 = 10^{-3}$ requires approximately the same number of steps ($t \approx 20000$) as $\epsilon_1 = 10^{-30}$ to achieve near-zero training loss. We conclude that Adafactor is not sensitive to the choice of $\epsilon_1$, and a relatively large $\epsilon_1$ can still lead to convergence, making the polynomial dependency $\mathcal{O}(1/\epsilon_1)$ in our convergence bounds acceptable.

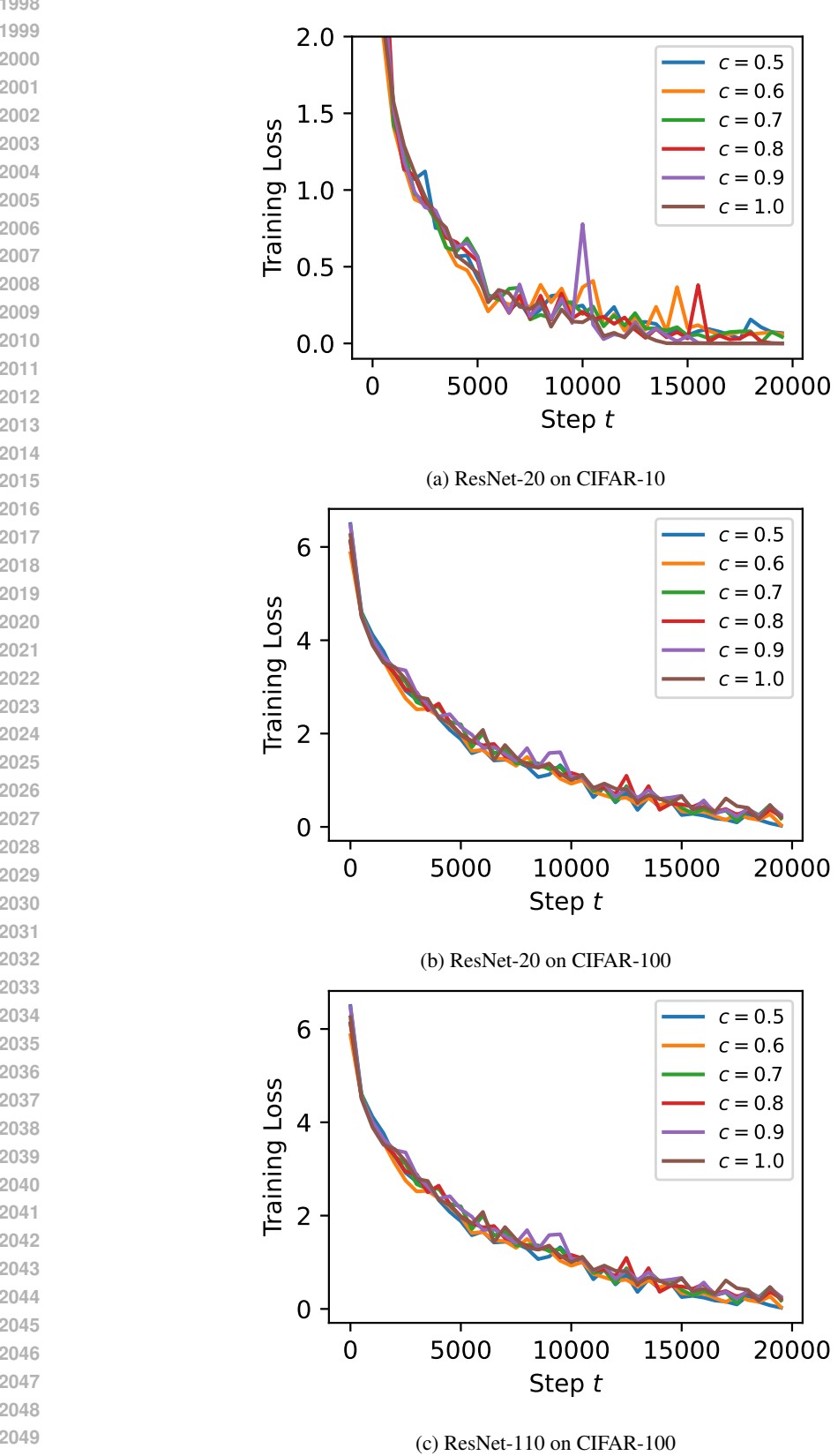

(a) ResNet-20 on CIFAR-10

(b) ResNet-20 on CIFAR-100

(c) ResNet-110 on CIFAR-100

Figure 2: Average training loss curve under different decay rate parameters $c$.

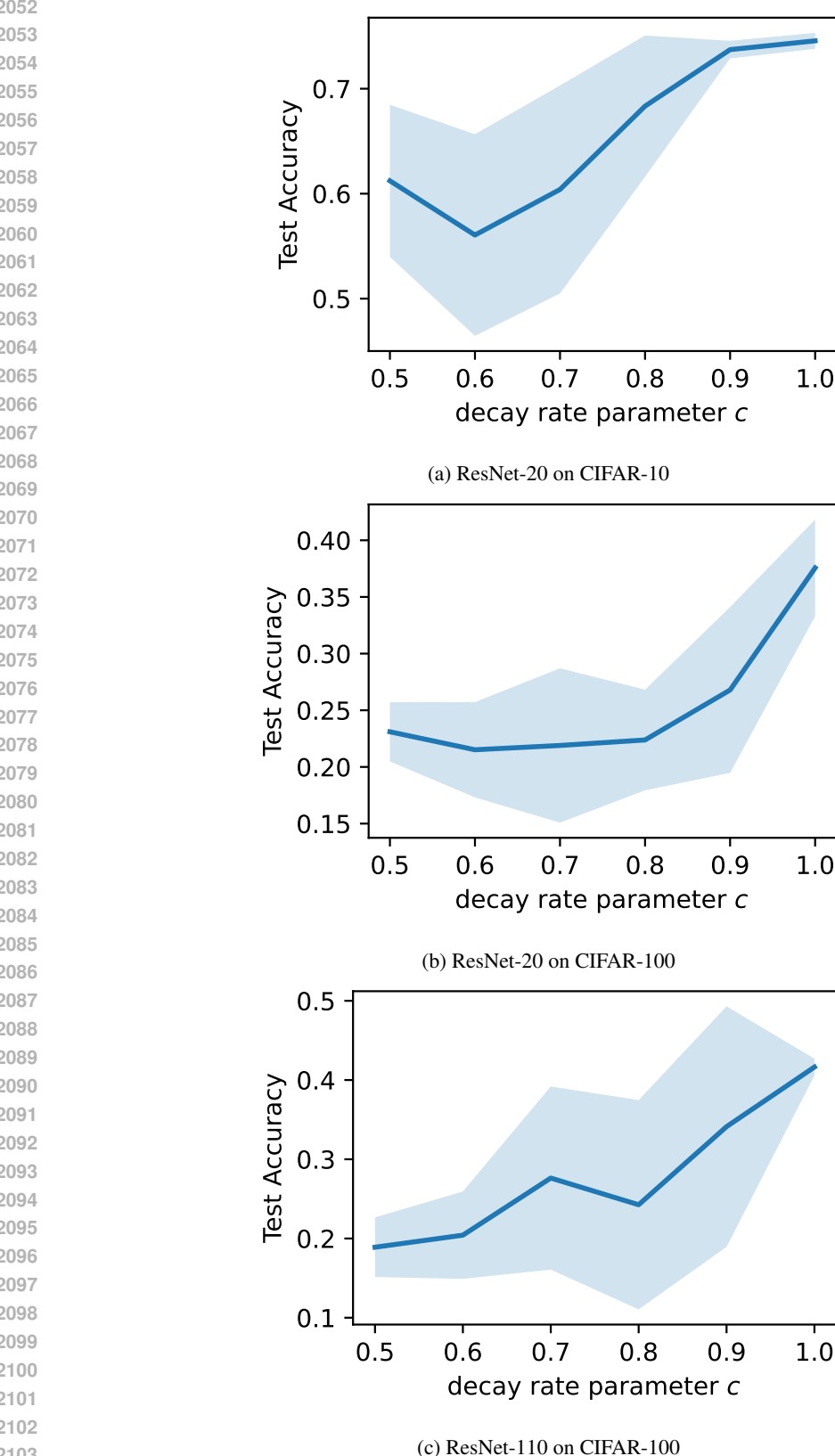

(a) ResNet-20 on CIFAR-10

(b) ResNet-20 on CIFAR-100

(c) ResNet-110 on CIFAR-100

Figure 3: Average test accuracy and standard deviation (shallow blue region) under different decay rate parameters $c$.

## E.5 REPORT ON EXPERIMENT 3

Table 3: The test accuracy after 400 epochs. We use Adafactor with different time-varying clipping thresholds and other hyper-parameters are set by default. We do not apply the warm-up technique.

| $\alpha$ (no warm up) | ResNet 20 / CIFAR 10 | ResNet 20 / CIFAR 100 | ResNet 110 / CIFAR 100 |
|---|---|---|---|
| $\alpha = 4.0$ | 0.6947 | 0.3096 | 0.3508 |
| $\alpha = 6.0$ | 0.7420 | 0.3600 | 0.4359 |
| $\alpha = 7.0$ | 0.7558 | 0.3564 | 0.4483 |
| $\alpha = 8.0$ | 0.7556 | 0.3729 | 0.4586 |
| $\alpha = 9.0$ | 0.7751 | 0.3771 | 0.4401 |
| $\alpha = 1.0$ (default) | 0.8031 | 0.4535 | 0.4906 |

Table 4: The test accuracy after 400 epochs. We use Adafactor with different time-varying clipping thresholds and other hyper-parameters are set by default. We apply the warm-up technique.

| $\alpha$(warm up) | ResNet 20 / CIFAR 10 | ResNet 20 / CIFAR 100 | ResNet 110 / CIFAR 100 |
|---|---|---|---|
| $\alpha = 4.0$ | 0.6331 | 0.2753 | 0.2958 |
| $\alpha = 6.0$ | 0.6812 | 0.2988 | 0.3433 |
| $\alpha = 7.0$ | 0.6811 | 0.3111 | 0.3547 |
| $\alpha = 8.0$ | 0.6930 | 0.3195 | 0.3658 |
| $\alpha = 9.0$ | 0.6969 | 0.2969 | 0.3855 |
| $\alpha = 1.0$ (default) | 0.7371 | 0.3812 | 0.4085 |

In this experiment, we explore the appropriate values of $\alpha$ in Theorem 7.1 and compare the training performance to the default setting of $d = 1$. As indicated by Theorem 7.1, a relatively small $\alpha$ is desirable for better dependency on $\epsilon_1$. We train models with $\alpha$ set to 4, 6, 7, 8, and 9, keeping other hyper-parameters at their default values. We also train models with the default $d = 1$ setting as the baseline. We report the test accuracy after training 400 epochs. We also plot the training loss against the steps in Figure 5 without step-size warm-up and Figure 6 with step-size warm-up.

The results indicate that, for the values of $\alpha = 6, 7, 8, 9$, Adafactor achieves comparable convergence speed compared to the default threshold (represented by "Baseline"), which helps to complement the theoretical results in Theorem 7.1.

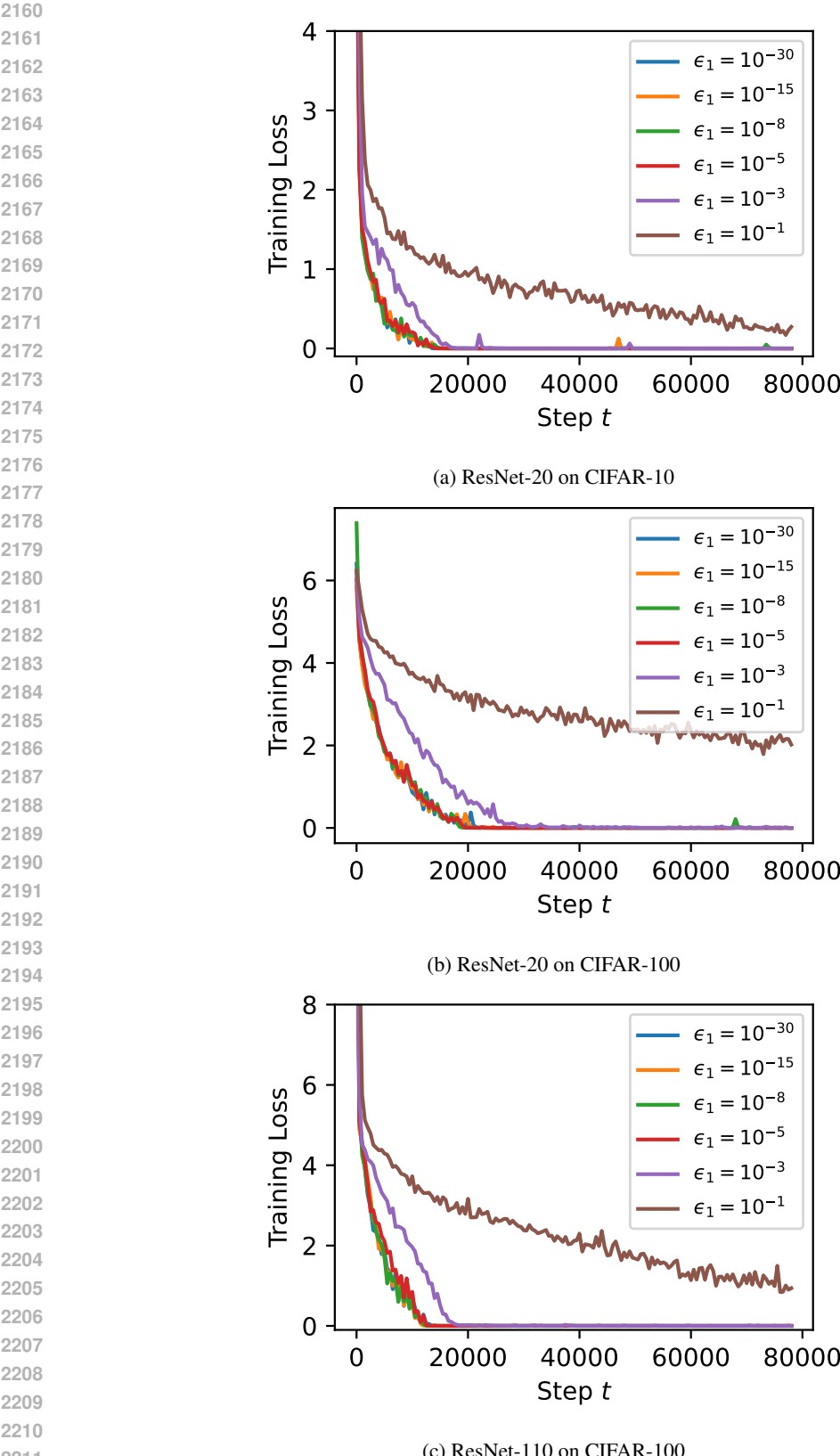

(a) ResNet-20 on CIFAR-10

(b) ResNet-20 on CIFAR-100

(c) ResNet-110 on CIFAR-100

Figure 4: Training loss vs. steps using Adafactor without update clipping under different $\epsilon_1$. The step-size $\eta_t$, decay rate $\beta_{2,k}$, and learning rate warm-up are set by default.

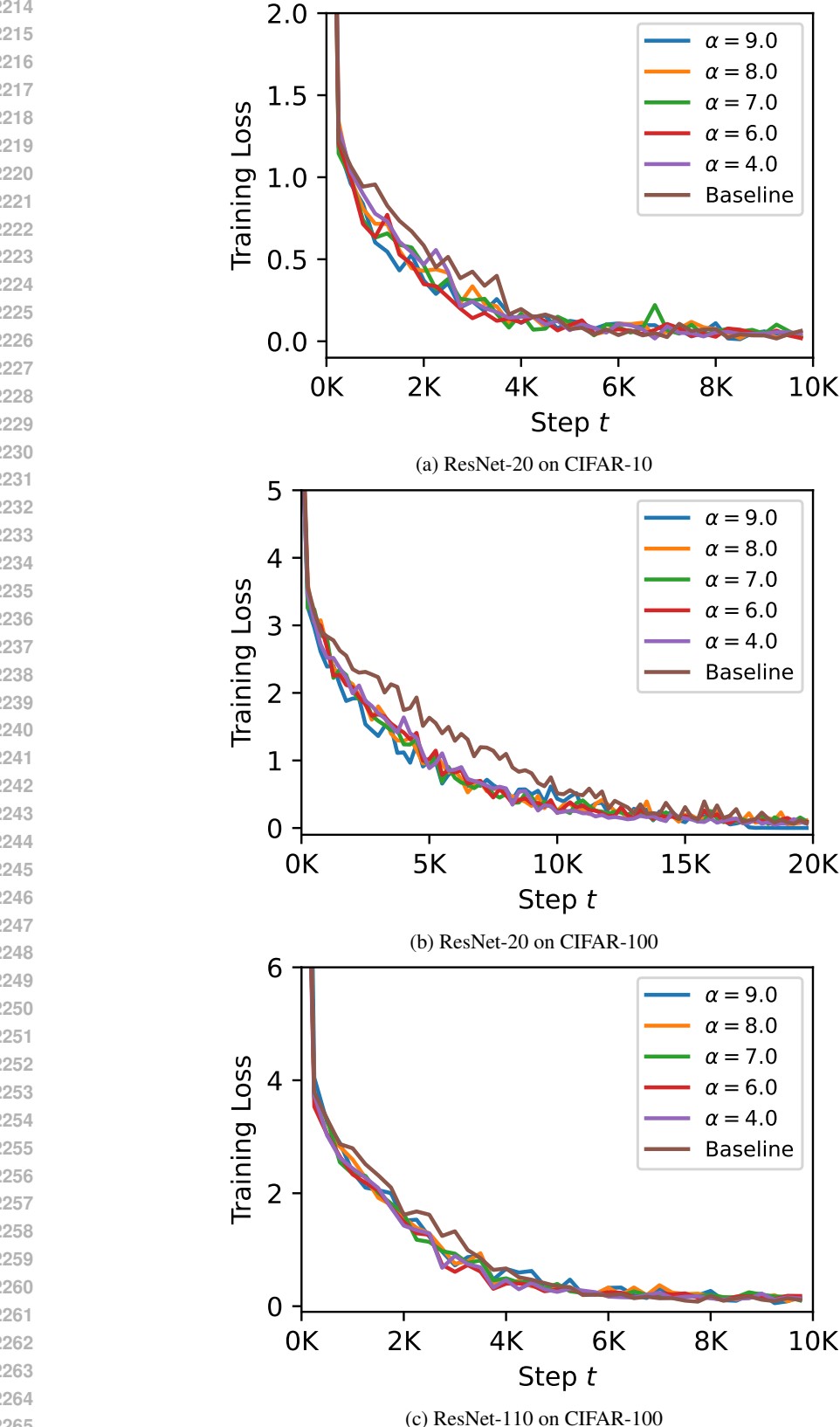

(a) ResNet-20 on CIFAR-10

(b) ResNet-20 on CIFAR-100

(c) ResNet-110 on CIFAR-100

Figure 5: Training loss vs. steps on different models and datasets. We use step-size without warm-up technique and test under different $\alpha$.

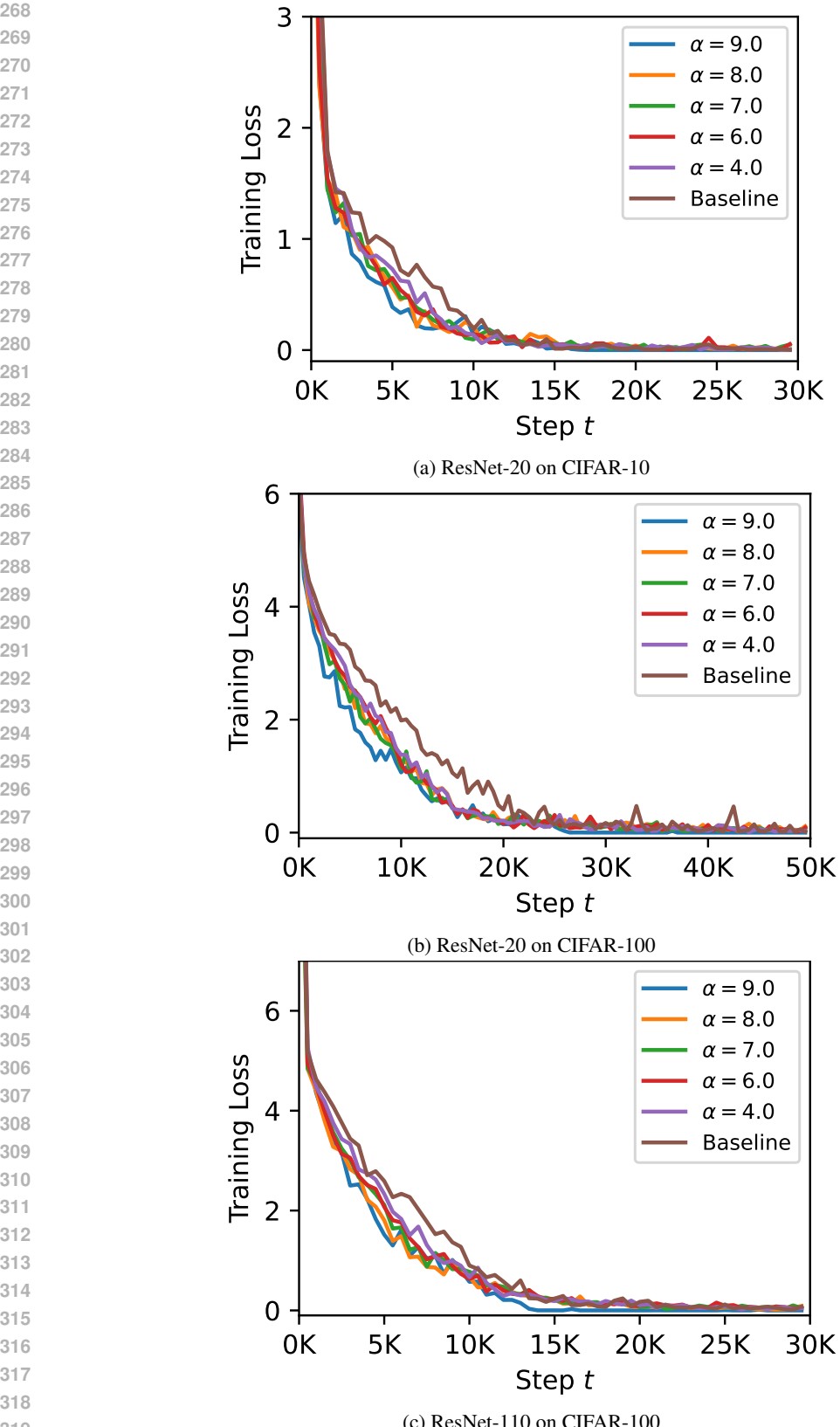

(a) ResNet-20 on CIFAR-10

(b) ResNet-20 on CIFAR-100

(c) ResNet-110 on CIFAR-100

Figure 6: Training loss vs. steps on different models and datasets. We use step-size with warm-up technique by default and test under different $\alpha$.

