# OpenReview forum: "Convergence of Adafactor under Non-Convex Smooth Stochastic Optimization"
_ICLR.cc/2025/Conference — Submitted to ICLR 2025_

### Official Review · Reviewer_uk85 · 2024-10-22

**Soundness:** 3
**Presentation:** 2
**Contribution:** 2
**Rating:** 5
**Confidence:** 2

**Summary:**

This paper investigated the first convergence behavior of Adafactor on non-convex smooth case. The results show that Adafactor could achieve comparable convergence performance to Adam. The authors also designed a new proxy step-size to decouple the stochastic gradients from the unique adaptive step-size and update clipping.

**Strengths:**

1.The authors list all major differences of Adafactor from Adam, which benefits readers’ understanding.

2.Some discussions are given to emphasize the advantages of the results in this paper.

3.The challenges and techniques of proof are presented in detail.

**Weaknesses:**

1.Although the quality of paper doesn’t depend on the word count of Abstract, readers may have a bad impression towards this paper. I think this abstract doesn’t fully present all contributions of this paper.

2.Abstract (line 16) shows the convergence rate of full-batch Adafactor is $\tilde{\mathcal{O}}(1/T)$. However, the first contribution (lines 67-70) is that the convergence rate of full-batch Adafactor is $\tilde{\mathcal{O}}(1/\sqrt{T})$.

3.In Equation (13), the second result should not have $\triangle_0^2$ rather than $\tilde{\triangle}_0^2$.

4.In line 1430, the first $C_0$ should be $C_0^\prime$.

5.The word “estimation” is just mentioned once in the main text. Some explanations of this word are necessary for readers’ understanding.

**Questions:**

1.The third contribution (lines 75-77) states that Adafactor a time-varying clipping threshold could also achieve the best convergence rate of  $\tilde{\mathcal{O}}(1/\sqrt{T})$. Why do the authors think the rate $\tilde{\mathcal{O}}(1/\sqrt{T})$ is best? Isn't the rate $\tilde{\mathcal{O}}(1/\sqrt{T})$ better than $\tilde{\mathcal{O}}(1/\sqrt{T})$?

2.How do the equations (26) and (38) hold since we don’t know whether $\frac{d\sqrt{mn}}{\|\bar{\mathbf{U}}_k\|_F} > 1$ holds, which is the key point to derive a free-dimension numerator bound (the second result in Equation (13))?

3.In Theorem B.1, I don’t find any dependency on the dimension $d$ like Theorem A.1. So, why do the authors regard the results (Equations (43), (45)) of Theorem B.1 as free dimension bounds?

---

> ### Author Response · Authors · 2024-11-17
> **Rebuttal**
>
> We thank a lot for the reviewer's effort and valuable suggestions on our manuscript.
>
> > **W1.** Missing full presentation on contributions in the Abstract.
>
> **Re.** We have added some more sentences in the Abstract to clarify our contributions. Please refer to the revised version.
>
> ---
>
> > **W2.** Abstract (line 16) shows the convergence rate of full-batch Adafactor is $\tilde{\mathcal{O}}(1/T)$. However, the first contribution (lines 67-70) is that the convergence rate of full-batch Adafactor is $\tilde{\mathcal{O}}(1/\sqrt{T})$.
>
> **Re.** In Theorem 5.1, we derive $\tilde{\mathcal{O}}(1/T)$ rate for constant $\rho$ and  $\tilde{\mathcal{O}}(1/\sqrt{T})$ rate for time-varying $\rho_k$. To clarify this, we have updated the "Contribution" part (Lines 74-75) in the new version.
>
> ---
>
> > **W3 and W4.** Some typos.
>
> **Re.** We have corrected these typos in the revised version.
>
> ---
>
> > **W5.** The word “estimation” is just mentioned once in the main text.
>
> **Re.** The term "estimations" appears only in Section 10 in the main text. In the revised version, we provide some further explanation:
> "Additionally, we rely on the unique structure of proxy step-sizes and an appropriate choice of $\beta_2$ to control the additional errors."
>
> ---
>
> > **Q1.** Why do the authors think the rate $\tilde{\mathcal{O}}( 1/\sqrt{T})$ is best?
>
> **Re. The "best rate" refers to the convergence rate that matches the lower bound.** We note that the lower bound is $\mathcal{O}( 1/\sqrt{T})$ for first-order stochastic gradient methods on non-convex smooth optimization (as shown in [1]). If we ignore the logarithm order, the convergence rate in our paper shares the same order as the lower bound. Hence, it is in general optimal and unimprovable.
>
> ---
>
> > **Q2.** How do the equations (26) and (38) hold since we don’t know whether $d\sqrt{mn} > \\|\bar{U}_k\\|_F$.
>
> **Re. We don't necessarily need to know whether $d\sqrt{mn} > \\|\bar{U}_k\\|_F$ due to the maximum operator inside the update clipping.** Specifically, it's clear to see that $1/\max\\{1,\\|\bar{U}\_k\|\_F/(d\sqrt{mn})\\} \le d\sqrt{mn}/\\|\bar{U}\_k\\|\_F$. Therefore, we derive Eq. (26) as
> $$
> {\bf (b)} = \frac{L}{2}\sum\_{k=1}^t\left(\frac{\max\\{\epsilon\_2,RMS(X\_k)\\}\rho\_k}{\max\\{1,\\|\bar{U}\_k\\|\_F/(d\sqrt{mn})\\}} \right)^2 \\|\bar{U}\_k\\|\_F^2 \le \frac{Ld^2mn(\epsilon\_2+\Theta\_{\max})^2}{2}\sum\_{k=1}^t \rho\_k^2 \cdot \frac{\\|\bar{U}\_k\\|\_F^2}{\\|\bar{U}\_k\\|\_F^2}.
> $$
> Eq. (38) could be similarly derived since $1/\max\\{1,\|\bar{U}\_k\\|\_F/(d\sqrt{mn})\\} \le 1$.
>
> ---
>
> > **Q3.** Why do the authors regard the results (Equations (43), (45)) of Theorem B.1 as free dimension bounds?
>
> **Re.** **The factors involving $m$ and $n$ appear in the denominator, so the convergence bounds are dimension-free.** Specifically, the dimension dependencies $m,n$ in Theorem B.1 are indeed present. These dependencies are determined by $G_1,G_2,G_3$ in Eq. (40), through coefficients $C_0',C_2',C_3'$ in Eq. (39). All the above reference numbers are the new revised version.
>
> ---
>
> **References.**
>
> [1] Arjevani et al., Lower Bounds for Non-Convex Stochastic Optimization, Mathematical Programming, 2023.

---

> > ### Comment · Reviewer_uk85 · 2024-11-25
> >
> > Thanks for the authors' responses which answer my questions. I have checked other reviewers' questions and the authors' corresponding responses. I will hold my current score.

---

> > > ### Author Response · Authors · 2024-11-25
> > >
> > > Thanks a lot for the reviewer's reply. We have now replied to other reviewers' new questions and updated a new version of our paper. If there is any further question, we are willing to discuss it.
> > >
> > > Best,
> > >
> > > Authors.

---

### Official Review · Reviewer_pL2X · 2024-10-30

**Soundness:** 3
**Presentation:** 2
**Contribution:** 2
**Rating:** 5
**Confidence:** 4

**Summary:**

This paper provides a theoretical analysis of the convergence of AdaFactor in the non-convex smooth setting. The analysis is given both for the deterministic case (full-batch) and for the stochastic case. Moreover, the authors obtain theoretical guarantees of convergence when clipping is added to AdaFactor.

**Strengths:**

1) These results look novel, as authors demonstrate the analysis of AdaFactor for the non-convex smooth case with various considerations of oracle (with and without stochasticity).
2) The proofs look correct.
3) The paper is easy to follow except some shortcomings (see section Weaknesses).

**Weaknesses:**

1)  The paper is positioned as theoretical, so it is expected to see some more advanced theoretical results, especially they already exist in the literature.

а) Assumption A.4 (almost surely bounded stochastic gradient) is quite strong and noticeably reduces the class of possible stochastic oracles (as a consequence, possible distributions of stochasticity). Indeed, there exist some analyses for adaptive methods (AdaGrad) which provide theoretical guarantees of convergence under weaker assumption on stochastic oracle (e.g. see [1]), more specifically, with sub-Gaussian noise. Is it possible, in the authors' opinion, to obtain similar convergence rates as in the provided analysis by relaxing Assumption A.4? For example bounded not stochastic gradient but variance or other moment.

b) It would also be interesting to know what the authors were guided by when they integrated the clipping technique into AdaFactor. Based on convergence bounds (see Theorems 6.1 and 7.1), the addition of clipping is not really justified since the theoretical guarantees for AdaFactor with and without clipping are the same. What is the authors' main idea to add clipping? Typically clipping is needed when the stochastic gradient has heavy tails [2]. But again the authors have a simple assumption of boundedness.

[1] Liu, Zijian, et al. "High probability convergence of stochastic gradient methods." International Conference on Machine Learning. PMLR, 2023.

[2] Gorbunov, Eduard, Marina Danilova, and Alexander Gasnikov. "Stochastic optimization with heavy-tailed noise via accelerated gradient clipping." Advances in Neural Information Processing Systems 33 (2020): 15042-15053.

2) The experimental section is pretty weak. As I said the paper is positioned as theoretical, so weak experiments are not a big disadvantage. But it is nice to add at least one experiment comparing with other methods (Adam, RMSProp, Lion and other SOTA) except AdaFactor. Just to understand which method we talk about in terms of its practical feasibility.

3) Some shortcomings in terms of the writing of the paper:

    a) In the part with contribution, there must be $\mathcal{O}\left(\frac{1}{T}\right)$ instead of $\mathcal{O}\left(\frac{1}{\sqrt{T}}\right)$ (for the full-batch setting), see Theorem 5.1. Right?

    b) Superfluous numbering of formulas: did not found any references on $(4), (5), (11), (19), (21), (23), (37), (39), (41), (64), (69), (76), (86), (101), (102)$. It is worth noting that, possibly, they should be removed. It can increase the readability of the paper.

========================

The article is not bad, but the theoretical analysis is rather rough for me, there are more delicate results in the literature. And theory is the main contribution.

**Questions:**

See Weaknesses

---

> ### Author Response · Authors · 2024-11-17
> **Rebuttal**
>
> We thank a lot for the reviewer's effort and valuable suggestions on our manuscript.
>
> > **W1(a).** Assumption A.4 (almost surely bounded stochastic gradient) is quite strong.
>
> **Re.** Please refer to the **global rebuttal** related to the discussion of the bounded stochastic gradient assumption.
>
> ---
>
> > **W1(b).** What is the authors' main idea to add clipping?
>
> **Re.** **Main idea of adding clipping.**
>
> We follow the update clipping introduced by [1] to recover the vanilla Adafactor as closely as possible. As we mentioned in Section 4 "Update clipping" paragraph, the clipping is used to "calibrate the second moment estimator $W_k$ when it exceeds a threshold $d$." We highlight that it's different from the standard clipping using $1/\max\left\\{1,\frac{\lambda}{\\|{{G}_k}\\|} \right\\}$. It remains unknown whether the clipping in Adafactor plays the same role in heavy-tail cases as the standard one, which is beyond the scope of this paper.
>
> **Same rate for two cases.**
>
> We believe that bounded noise assumption could be the main reason for deriving the same rates for Adafactor both with and without clipping. We suspect that there may be some essential differences when considering the heavy-tail noise.
>
> ---
>
> > **W2.** The experimental section is weak.
>
> **Re.** We have added an experiment using Adam to train BERT-Base under the same setup. The corresponding training loss curve is added to Figure 1. It's shown that Adafactor could achieve a comparable convergence rate as Adam.
>
> ---
>
> > **W3.** Some shortcomings in terms of the writing of the paper
>
> **W3(a).** **Re.** To clarify this, we have updated the corresponding expressions in the "Contribution" part. See Lines 74-75.
>
> **W3(b).** **Re.** We have removed these redundant references in the revised version.
>
> ---
>
> **References.**
>
> [1] Shazeer et al., Adafactor: Adaptive Learning Rates with Sublinear Memory Cost, arXiv:1804.04235.

---

> > ### Comment · Reviewer_pL2X · 2024-11-25
> >
> > Thanks to the authors for the response!
> >
> > 1) The fact that such an assumption (bounded gradients) has been used before is not a pre-satisfying argument. I found another paper [1] (or [1.5] - new version, it can be ignored since it is submitted to the same confrence, but [1] is sufficient for the example). Theoretical results on analyzing adaptive methods (AdaGrad and others) are now at a much more advanced level than the authors give. Since theory is the main contribution - this is important!
> >
> > 2) Moreover, it seems getting a theory that a method converges like Adam is not too hard [2]. Especially considering that the theory for Adam is far from the most advanced and worse than for the usual SGD [3].
> >
> > 3) Sketch of the proof is not enough for the theoretical paper.
> >
> > [1] Gradient Clipping Improves AdaGrad when the Noise Is Heavy-Tailed
> >
> > [1.5] Clipping Improves Adam and AdaGrad when the Noise Is Heavy-Tailed
> >
> > [2] Stochastic Gradient Methods with Preconditioned Updates
> >
> > [3] A Simple Convergence Proof of Adam and Adagrad
> >
> > I think the current version of the paper still needs improvement! I'm leaving my assessment.

---

> ### Author Response · Authors · 2024-11-25
>
> Thanks a lot for the reviewer's valuable reply!
>
> As far as we can understand, the reviewer raises two major concerns (if not please correct me):
>
> - the bounded gradient assumption may be restrictive as many recent works for Adam and AdaGrad have dropped;
> - the proof techniques may be similar to some other precondition updated algorithms such as [1] and may be easily derived.
>
> We thank a lot for the reviewer's reminder of the three works. **We have cited them with proper comments in Section 2, from Line 107 to 111.** In response to the above concerns:
>
> - We agree that there are some advanced results for AdaGrad or Adam without relying on bounded gradients.  However, due to the page limitation, we choose standard assumptions to help readers better understand the key ideas of the proof technique.
>
> - **The analysis for Adafactor can be very different from existing adaptive methods even considering simple bounded gradients.** We summarize two major challenges, leading to a preconditioner different from [1], Adam and AdaGrad:
>   - the non-linear adaptive step-size $W_k$ with a rather complicated structure to estimate;
>   - an update-clipping which is essentially different from the standard clipping, incorporating more correlation terms when estimating ${\bf (I)}$ and ${\bf (II)}$ in Eq.(8).
> - We think that the motivations of [1] and Adafactor are also different. [1] focuses on approximating the diagonal of the Hessian when the problem is ill-conditioned. It's a memory-unconstrained algorithm similar to AdaGrad and Adam. In comparison, Adafactor focuses on memory saving.
>
>
>
> For the third point, as far as we understand, (if not please correct me), the lack of the proof sketch refers to the newly added sub-Gaussian case in Appendix C. **We have revised it in detail in the new version.**
>
> References.
>
> [1] Sadiev et al. Stochastic Gradient Methods with Preconditioned Updates, arXiv:2206.00285.
>
> ---
>
> We thank you so much for the constructive reply. If there is any further concern, we are willing to discuss it.
>
> Best,
>
> Authors.

---

### Official Review · Reviewer_nLPm · 2024-11-02

**Soundness:** 2
**Presentation:** 4
**Contribution:** 2
**Rating:** 5
**Confidence:** 3

**Summary:**

The paper studies the theoretical convergence of Adafactor without momentum for smooth non-convex optimization. That is, under the assumption of Lipschitzness of the gradient, lower boundedness of the objective, and uniform boundedness of the (stochastic) gradients, the authors derive new convergence guarantees for Adafactor with and without update clipping.

In particular, for the deterministic version of Adafactor, the authors derive $O(\log(T) / T)$ rate for the constant relative stepsize parameter $\rho_k$ and $O(\log(T) / \sqrt{T})$ rate for $\rho_k \sim k^{-1/2}$. Both results hold for $\beta_{2,1} = 1/2$ and arbitrary $\beta_{2,k} \in (0,1)$ for $k \geq 2$

Next, for the stochastic version of Adafactor without update clipping, the authors prove $O(\log(T/\delta) T^{-c + 1/2})$ high-probability convergence rate, where $\delta \in (0,1)$ is the failure probability and parameter $c \in [1/2,1]$ defines the rate of the increase of $\beta_{2,k}$ as $\beta_{2,k} = 1 - k^{-c}$ for $k\geq 2$. In particular, when $c = 1$, the derived rate matches the one known for Adam.

Finally, for the stochastic version of Adafactor with update clipping, the authors show similar results when the clipping parameter $d_k$ is chosen as $d_k = k^{\frac{c}{2(\alpha - 1)}}$ for some $\alpha > 1$.

The authors also provide several experimental results where they test the performance of Adagrad for different increase rates for $\beta_{2,k}$ parameter and the sensitivity of the method to the choices of $\epsilon_1$ and $\alpha$ parameters.

**Strengths:**

S1. To the best of my knowledge, this paper provides the first theoretical convergence analysis of Adafactor. Given the popularity of the method and the relevance of memory-efficient optimizers to modern problems such as the training of LLMs, obtaining rigorous theoretical convergence guarantees for Adafactor is important for the community.

S2. The authors provide a sketch of the proof that helps to understand the key steps of the analysis.

**Weaknesses:**

W1. The analysis relies on the assumption that the $\ell_\infty$-norms of **the iterates are bounded from below and from above by $\Theta_{\min}$ and $\Theta_{\max}$ respectively**. This is a restrictive assumption, and it should be included in the list of assumptions provided in Section 3. In lines 302-304, the authors briefly mention that they assume $\|\| X_k\|\| \leq \Theta_{{\max}}$ and that the rate depends on $\Theta_{\max}$, but they do not mention in the main text that all the results of the paper rely on the additional assumption that $\|\|X_k \|\| \geq \Theta_{\min}$ for some $\Theta_{\min} > 0$. Both conditions are restrictive and it is not obvious why the norms of the iterates are bounded with probability $1$. The best-known convergence results for Adam do not require such assumptions.

W2. The analysis relies on the boundedness of the gradients and the boundedness of the gradient noise. Although the boundedness of the gradients is a common assumption in the literature, assuming boundedness of the noise is quite restrictive: this assumption seems to be unrealistic (e.g., even for Gaussian noise, it doesn't hold), and, in addition, even if the noise is bounded for a particular run of the method, the bound can be huge, especially for training LLMs [1].

W3. The derived bounds are proportional at least to $\epsilon_1^{-1/2}$ (e.g., the first bound from (13)), some bounds are proportional to $\epsilon_1^{-5/2}$ (e.g., the second bound from (13)). Since $\epsilon_1$ is typically small, this extra factor is noticeable. Moreover, the best-known results for Adam do not have such a dependency on an analogous parameter, e.g., in [2], the dependency is only logarithmical.

W4. The rate in the deterministic case is $O(mn\log(T)/T)$ (for constant $\rho_k$) or to $O(mn\log(T)/\sqrt{T})$ (if we assume that $A_0$ and $A_1$ are $O(1)$ in the first bound in (13)) or to $O(\epsilon_1^{-2}\log(T)/\sqrt{T})$ (if we assume that $A_0$ and $A_1'$ are $O(1)$ in the second bound in (13)). That is, the rate is either proportional to $mn$, i.e., the dimension dependence is similar to Adam, or to $\epsilon_1^{-2}$, which is of the order $10^{60}$ - much larger than the dimensionality of the largest existing model - for the default value $\epsilon_1$.

W5. Experiments do not clearly illustrate the dependence on parameters $\epsilon_1, c, \alpha$. In particular, in the experiment for BERT-base model, the authors report only training loss. Next, in the experiments with ResNet, the authors show that the accuracy, in fact, largely depends on $c$, but from the training loss, it is not that clear. Finally, in the experiments from sections D.2 and D.3 the authors do not report the accuracy and the test loss, thus, the real influence of $\epsilon_1$ and $\alpha$ remains unclear. Moreover, in Figures 4, 5, and 6, the difference between the lines is not visible. I recommend to zoom in the plots and show the area where the loss is small.

W6. The paper requires a thorough proofreading. I noticed several inaccuracies in English and flaws in the structure. Here are some of them:
- title: under --> for
- line 48: on non-convex smooth setup --> for non-convex smooth setup
- line 54: under --> for
- lines 92-94: this part should be rephrased because it could be misunderstood by the readers, i.e., one can interpret that the authors mention only a part of closely related works
- line 145: dependent by the random sample --> dependent on the random sample
- page 4: missing periods after "Matrix factorization", "Increasing decay rate", and so on
- line 182: maintain --> maintains
- section 6.1: the first and the second paragraphs in the discussion of $c$ and optimal rate are redundant
- section 6.1: the whole paragraph about $\epsilon_1$ and $\epsilon_2$ gives a technical discussion of the results that are given in the appendix. This discussion is not self-contained: to understand what the authors mean, one needs to read the result in the appendix. I suggest adding the details to the main text.


---
References

[1] Zhang et al. Why are Adaptive Methods Good for Attention Models? NeurIPS 2020

[2] Défossez et al. A Simple Convergence Proof of Adam and Adagrad. TMLR 2022

**Questions:**

Q1. Can the authors remove the assumption from (14), i.e., boundedness of the iterates, and derive the results without it?

Q2. Is it possible to generalize the results to the case of unbounded noise?

Q3. Is it possible to remove the assumption of boundedness of the gradients at least for the deterministic case?

Q4. Is it possible to show similar bounds to the ones derived in the paper but with logarithmic dependence on $\epsilon_1$?

Q5. Is it possible to prove $O((m+n)\log(T)/T)$ convergence rate for Adafactor?

Q6. Why is a time-varying threshold considered in Section 7? Is it crucial? Is it possible to prove a similar result for constant $d_k$?

**Other comments.**

C1. Reference to PaLM in line 41 is a bit misleading since, in that work, the authors use Adafactor without factorization, which is essentially Adam with re-scaling.

C2. In the discussion of the choice of $\beta_2$, it is worth to add that with constant $\beta_2$ (stochastic) Adam is not guaranteed to converge [1].

C3. It is worth mentioning that update clipping in Adafactor is not a standard gradient clipping.

C4. In formula (10), $\bar{G}_k$ is not defined (it is defined in the appendix).

C5. Line 378: (18) --> (10)

C6. Line 399: $V_k \to R_k$

---
References

[1] Reddi et al. On the convergence of Adam and beyond. ICLR 2018

---

> ### Author Response · Authors · 2024-11-17
> **Rebuttal (I)**
>
> We thank a lot for the reviewer's effort and valuable suggestions on our manuscript.
>
> > **W1.** The assumption of $\Theta_{\min},\Theta_{\max}$ is restrictive.
>
> **Re. The requirement for $\Theta_{\min}$ is not necessary.** In all our convergence results, $\Theta_{\min}$ only appears in the factor $1/\max\\{\epsilon_2, \Theta_{\min}\\}$ (Line 689 and Line 931 in the original version), which can be further bounded by $1/\epsilon_2$. We have removed $\Theta_{\min}$ in the revised version.
>
> **The requirement for $\Theta_{\max}$ is not essential.** The dependency of $\Theta_{\max}$ comes from the relative step-size $\max\\{\epsilon_2,{\rm RMS}(X_k)\\}$ in vanilla Adafactor. In contrast, Adam does not involve such a relative step-size, and therefore, its convergence results do not require a bounded iteration norm. If we use the same constant step-size as in Adam, $\Theta_{\max}$ could be dropped. We have clarified this point in the revised version.
>
> ---
>
> > **W2.** The bounded stochastic gradient assumption is restrictive.
>
> **Re.** Please refer to the global rebuttal.
>
> ---
>
> > **W3 and W4.** The dependencies of $\epsilon_1,m,n$ may not be so tight.
>
> **Re.** **The same parameter dependencies regarding $\epsilon_1$ and $m,n$  are consistent with several existing convergence bounds for memory-unconstraint optimizers.** Below is a brief discussion.
>
> **In terms of $\epsilon_1$.**
>
> The polynomial dependency appears in several convergence results, e.g., [1, 2, 3] for Adam and [4, 5] for AdaGrad. We note that some of these works are even derived in recent years. We believe that the relative rough dependency of $\epsilon_1$ in Eq. (13) mainly comes from the framework we used to handle the unique adaptive step-size and update clipping in Adafactor.
>
>  **In terms of $m,n$.**
>
> As you mentioned, the dimension dependency is similar to Adam with the right parameter. We also highlight that most existing convergence results for coordinate-wise algorithms suffer from the "curse of the dimension", e.g., [6, 7] for Adam-type and [8] for AdaGrad.
>
> As we list above, these rough dependencies appear similarly in several convergence results for Adam and AdaGrad. As the first theoretical result for Adafactor, our focus is on showing that it achieves a convergence rate comparable to Adam with the right parameter, with some of the parameter dependencies following the previous literature.  We agree that improving the orders for $\epsilon_1$ and $mn$ is vital, particularly for LLMs and we also try to derive dimension-free bounds in our paper. We will leave the improvement for these parameter dependencies for future work.
>
> ---
>
> > **W5.** Experiments do not clearly illustrate the theory.
>
> **Re.** **There exists a gap between the theoretical convergence upper bound and practical results.** We mainly provide the training loss curve as our theoretical analysis focuses on convergence rates rather than test accuracy. In response to the reviewer's concern, we also report the test accuracy for each experiment in Appendix E.
>
> We thank a lot for the reviewer's valuable suggestion on the presentation of Figures 4, 5, and 6. We have revised it accordingly in the new version.
>
> ---
>
> > **W6.** Some typos and presentation issues.
>
> **Re.** **We have corrected the typos as suggested.** For the last second point, we have removed the paragraph after Theorem 6.1 and kept the first and second paragraphs in the discussion of $c$ and optimal rate in Section 6.1. For the last point, we add hyperlinks to the specific formulas and sections in the Appendix instead of the detailed expressions due to the page limitation.
>
> ---
>
> **References.**
>
> [1] Zaheer et al., Adaptive Methods for Nonconvex Optimization, NeurIPS 2018.
>
> [2] Li et al., Convergence of Adam Under Relaxed Assumptions, NeurIPS 2023.
>
> [3] Wang et al., Closing the Gap Between the Upper Bound and the Lower Bound of Adam's Iteration Complexity, NeurIPS 2023.
>
> [4] Kavis et al., High Probability Bounds for a Class of Nonconvex Algorithms with AdaGrad Stepsize, ICLR 2022.
>
> [5] Wang et al., Convergence of AdaGrad for Non-convex Objectives: Simple Proofs and Relaxed Assumptions, COLT 2023.
>
> [6] Reddi et al., On the Convergence of Adam and Beyond, ICLR 2018.
>
> [7] Défossez et al., A Simple Convergence Proof of Adam and Adagrad, TMLR 2022.
>
> [8] Ward et al., AdaGrad stepsizes: Sharp convergence over nonconvex landscapes, JMLR 2020.

---

> ### Author Response · Authors · 2024-11-17
> **Rebuttal (II)**
>
> > **Q1.** Could the boundedness of the iterates be removed?
>
> **Re.** Please refer to our response to W1.
>
> ---
>
> > **Q2.** Could the results be extended to the case of unbounded noise?
>
> **Re.** Please refer to our response to W2.
>
> ---
>
> > **Q3.** Could the bounded gradient assumption be removed?
>
> **Re.** **Removing the assumption of bounded gradient would bring essential challenges for the analysis.** We note that some results for AdaGrad and Adam do not rely on bounded gradients, and we believe that incorporating their techniques might yield a convergence rate for Adafactor without this assumption. However, this remains an open question. It would be interesting for the future work.
>
> ---
>
> > **Q4 and Q5.** Could the dependencies on $\epsilon_1,m,n$ be improved?
>
> **Re.** The dependencies of $\epsilon_1,m,n$ are aligned with several existing convergence bounds of memory-unconstraint optimizers. We agree that it's significant to improve these dependencies. However, it remains unknown for us right now and we will leave it for the future works.
>
> ---
>
> > **Q6.** Is it possible to prove a similar result for constant $d_k$?
>
> **Re.** In our analysis, using a finite horizon constant such as $d_k = d \sim \mathcal{O}\left(T^{\frac{1}{2(\alpha-1)}}\right),\alpha > 1,\forall k \ge 1$ still leads to convergence. We refer to the formulas in Lines 1785-1790, where the convergence is derived by directly using this finite horizon constant setup.
>
> ---
>
> > **C1.** Incorrect comment on the reference of PaLm.
>
> **Re.** We have revised the incorrect expression by adding a footnote on Page 1.
>
> ---
>
> > **C2.** Adding the discussion on negative results using constant $\beta_2$.
>
> **Re.** We have added the sentence in Line 285-287 regarding the negative results in [1] when using constant $\beta_2$.
>
> ---
>
> > **C3.** It is worth mentioning that update clipping in Adafactor is not a standard gradient clipping.
>
> **Re.** We have added the sentence to clarify the difference between the two clipping mechanisms in Lines 315-316.
>
> ---
>
> > **C4/C5/C6.** Some typos.
>
> **Re.** We have replaced $\bar{G}_k$ with $\nabla f(X_k)$. For other typos, we have revised them accordingly.
>
> ---
> **References.**
>
> [1] Reddi et al., On the Convergence of Adam and Beyond, ICLR 2018.

---

> > ### Comment · Reviewer_nLPm · 2024-11-24
> >
> > I thank the authors for their detailed responses and for their work on the revision. In particular, the authors addressed W2 and W6 from the list of weaknesses I provided. Therefore, I decided to increase my score from 3 to 5.
> >
> > However, W1, W3, W4, and W5 remain, namely:
> >
> > - dependence on $\Theta_{\max}$
> > - noticeable dependence on $\epsilon_1$ and $mn$
> > - in Figures 4-6, the difference between the lines is not clear in the end (it is better to show the last few epochs for the lines that overlap)
> >
> > In particular, I believe that the first two items from the list above are very important (though I realize that it is the first analysis of Adafactor): it is unclear how to bound $\Theta_{\max}$ and, as I explained in W4, the convergence bounds either proportional to $mn$ or to $\epsilon_1^{-2}$ (e.g., the bound from [1] from the list of papers mentioned by the authors is proportional to $\epsilon_1$ only). These aspects prevent me from increasing the score further.

---

> ### Author Response · Authors · 2024-11-25
> **Thank you so much for raising the score!**
>
> Thanks a lot for the positive feedback from the reviewer. Below is the response to the concerns from the reviewer.
>
> > Dependency on $\Theta_{\max}$.
>
> 1. **The assumption of bounded iteration norm is well-established in the literature.** Bounded iteration norms have been a common assumption in earlier works on adaptive gradient methods for online convex optimization. Notable examples include:
>    - [Duchi et al., 2011] for AdaGrad;
>    - [Kingma and Ba., 2015] for Adam;
>    - [Reddi et al., 2018] for AMSGrad.
>
> 2. **The dependency on $\Theta_{\max}$ can be improved.** We note that $\Theta_{\max}$ only comes from the relative step-size $\max\\{\epsilon_2, \text{RMS}(X_k) \\}$. If we suppose that $\max_{k \in [T]} \\|X_k\\|_F \le \Theta_0$, which is also a common assumption in the literature, we could obtain that
> $$
> RMS(X\_k) = \sqrt{\frac{1}{mn}\sum\_{i=1}^n\sum\_{j=1}^m \left(x\_{ij}^{(k)}\right)^2} \le \frac{\Theta\_0}{\sqrt{mn}}.
> $$
> ​The dependency is then improved as it's controlled by the inverse of the dimension.
>
> 3. **Dropping relative step-size may be fairer when comparing Adafactor and Adam.** If we consider a simplified form of Adafactor without the relative step-size, the need for $\Theta_{\max}$ vanishes. **This modification will provide a fairer comparison between Adafactor and Adam.** Specifically, without the relative step-size, Adafactor adopts a similar step-size form as Adam, both of which are $\rho_t = c_0/\sqrt{t}$ . Then, the only difference comes from the adaptive step-sizes as:
>
>    - Adafactor: matrix factorization for memory-saving;
>
>    - Adam: full matrix without memory-saving.
> ​The result in our paper can still show that two algorithms achieve a comparable convergence rate, 	meaning that the memory-saving mechanism in Adafactor does not fundamentally harm the convergence speed.
>
>
>
> > Dependency on $\epsilon_1$ and $mn$.
>
> We first summarize the dependency for $\epsilon_1$ and $mn$ in our convergence bounds as follows:
>
> -  convergence bounds with better $\epsilon_1$ order:
>
> Theorem A.1: $
>   \mathcal{O}\left(\frac{1}{\epsilon\_1} + mn \right)$.
>
> Theorem B.1: $ \mathcal{O}\left(\frac{(mn)^2\epsilon\_1^{3/2}+\sqrt{mn}}{\epsilon\_1}\log\frac{1}{\epsilon\_1} \right)$.
>
> - convergence bounds with better $mn$ order (we only keep the highest order):
>
> Theorem A.1: $\mathcal{O}\left(\frac{1}{(mn)^{3/2}\epsilon_1^{5/2}}+ \frac{1}{\epsilon_1} \right)$,
>
> Theorem B.1: $ \mathcal{O}\left(\frac{1}{(mn)^{5/2}\epsilon^{7/2}} + \frac{1}{\epsilon_1}\right)$.
>
>
> **For convergence bounds with better $\epsilon_1$ order.**
>
> - **The dependency on $\epsilon_1$ is at worse $\mathcal{O}\left(\epsilon\_1^{-1}\log(\epsilon\_1^{-1})  \right)$.**
> - **We clarify that Adafactor may not be sensitive to $\epsilon_1$.** As we show in Appendix E.4, using $\epsilon_1= 10^{-5}$ does not essentially harm the test accuracy (Table 2, Page 35) and training speed (Figure 4, Page 39) on Resnet experiments. Using $\epsilon_1=10^{-5}$, the $\mathcal{O}\left(\epsilon^{-1}\right)$ order could be somehow omitted given sufficiently large $T$. We also note that the same dependency on $\epsilon_1$ and similar sensitive experiment results appear in [1] for Adam.
>
> **For convergence bounds with better $mn$ order.**
>
> - **The effect of high order $\epsilon_1$ can be reduced by $mn$**. Although the dependency on $\epsilon_1$ is worse to e.g., $\mathcal{O}\left(\epsilon^{-7/2}\right)$ in dimension-free bounds, the effect can be reduced by $(mn)^{5/2}$ factor. Therefore, the dependency can be somehow regarded as $\mathcal{O}(\epsilon^{-1})$ in dimension-free bounds, as shown in the above formula.
>
> In conclusion, given the sufficiently large $T$, some improvement efforts on the dependency of $\epsilon_1,m,n$, and experimental results, we believe that the effect for the relatively relaxed order of $\epsilon_1,m,n$ is reduced. However, we really agree with the reviewer that the order for $\epsilon_1$ and $mn$ are not optimal due to the proof limitation. We will leave it for the future work.
>
> > In Figures 4 and 6, the difference between the lines is not clear in the end.
>
> We will revise the figure, zooming in on the final stage to make it more clear.
>
> **Reference.**
>
> [1] Li et al., 2023. Convergence of Adam Under Relaxed Assumptions. NeurIPS 2023.
>
> ---
>
> We thank you so much for the positive reply. If there is any further concern, we are willing to discuss it.
>
> Best,
>
> Authors.

---

### Author Response · Authors · 2024-11-17
**Global Rebuttal**

We thank all reviewers for their efforts and comments on our manuscript!

**The discussion on bounded stochastic gradient assumption.**

A major concern from reviewers regards the potentially restrictive bounded stochastic gradient assumption. We clarify as follows.

- **The bounded stochastic gradient (or bounded noise) is a commonly used assumption in the literature.**

We note that many early convergence results for AdaGrad, Adam, and AMSGrad, such as [1, 2, 3, 4, 5], all rely on bounded stochastic gradient assumptions.  **As the first theoretical convergence result for Adafactor**, we follow their lines to adopt this relatively simple assumption and focus on showing that Adafactor could obtain a comparable convergence rate as Adam with the right parameter.

We agree that it's important to consider more relaxed assumptions and we will add some remarks in the conclusion part. However, due to the complexity of analyzing a new algorithm, incorporating relaxed assumptions in a single paper is a challenge, which would also make the paper lengthy and hard to understand the key ideas of the proof technique.

- **We could relax to Gaussian/sub-Gaussian noise with bounded gradients.**

We also highlight that our analysis could be applied to the Gaussian/sub-Gaussian noise case. We have added a proof sketch regarding sub-Gaussian noise in the revised version, please refer to **Appendix C** in Line 1491. Here, we briefly discuss the extension.  Recalling the sub-Gaussian noise assumption,
$$
\text{sub-Gaussian:} \quad \mathbb{E}\left[\exp\left(\frac{\\|g(X,Z)-\nabla f(X)\\|^2}{\sigma^2}\right) \Big| X \right] \le \exp(1). \quad \quad \quad (1)
$$
Let the noise sequence $\\{\xi\_t\\}\_{t \in [T]}$ where $\xi\_t = g(X\_t,Z\_t)-\nabla f(X\_t)$ satisfies sub-Gaussian. Using a standard result, see e.g., [Lemma 5, 6], it holds that with probability at least $1-\delta$,
$$
\max\_{t \in [T]}\\|\xi\_t\\|^2 \le \sigma^2\log\left(\frac{{\rm e}T}{\delta} \right).
$$
Letting $\\|\nabla f(X\_t)\\|\le G\_0$ and the noise is sub-Gaussian (covering Gaussian), we could derive that with high probability $\\|g(X\_t,Z\_t)\\| \le  G\_0 +\sigma\log\left(\frac{{\rm e}T}{\delta} \right),\forall t \in [T]$. In addition, the high probability inequality in Lemma B.1 for bounding the martingale difference sequence could be applied to sub-Gaussian noise based on the definition in (1). Based on the two probability events, we could obtain similar convergence bounds as the results in our paper.

---

**The revised parts for presentation issues in the new version.**

All the revised parts are highlighted with **blue** color in the new version.

**Main text.**

- A revision on the Abstract, clarifying our contributions regarding full-batch Adafactor and experiments;
- Adding AMSGrad and its citation. See Line 27;
- Adding "PaLM applies Adafactor without matrix factorization." in the footnote of Page 1;
- Revision on  "Contribution" part regarding full-batch Adafactor. See Line 74-75;
- Revision on "Related Work" part. Adding some existing works on AMSGrad. See Line 97, Line 103-105;
- Adding a remark on the extension of our results to the sub-Gaussian noise case. See Line 163-165;
- Adding a remark that a constant $\beta_2$ could not ensure convergence for Adam. See Line 286-287;
- Clarifying that $\Theta_{\min}$ is not required and $\Theta_{\max}$ is not fundamentally necessary. See Line 305-306;
- Clarifying the essential difference between clipping in Adafactor and the standard one, citing [7]. See Line 211-212, 315-318;
- Adding the training loss curve using Adam to train BERT-Base in Figure 1;
- Adding an explanation on the word "estimation" in "Conclusion" part. See Line 534;
- Adding a sentence regarding the bounded stochastic gradient in "Limitation" part, citing [8]. See Line 538.

**Appendix.**

- All the convergence bounds eliminate the unnecessary dependencies to $\Theta_{\min}$;
- Adding a proof sketch for the extension to sub-Gaussian noise. See Appendix C;
- Zooming in all the figures in Appendix E;
- Adding test accuracy results for BERT-Base and ResNet experiments in Appendix E.

All the other typos pointed out by reviewers have been revised. We appreciate these comments from reviewers.

---

**References.**

[1] Kingma and Ba, Adam: A Method for Stochastic Optimization, ICLR 2015.

[2] Reddi et al., On the Convergence of Adam and Beyond, ICLR 2018.

[3] Zaheer et al., Adaptive Methods for Nonconvex Optimization, NeurIPS 2018.

[4] Ward et al., AdaGrad stepsizes: Sharp convergence over nonconvex landscapes, JMLR 2020.

[5] Kavis et al., High Probability Bounds for a Class of Nonconvex Algorithms with AdaGrad Stepsize, ICLR 2022.

[6] Li and Orabona, A High Probability Analysis of Adaptive SGD with Momentum, ICML 2020 Workshop.

[7] Gorbunov et al., Stochastic optimization with heavy-tailed noise via accelerated gradient clipping, NeurIPS 2020.

[8] Zhang et al., Why are adaptive methods good for attention models? NeurIPS 2020.

---

### Author Response · Authors · 2024-11-22
**Kindly Reminder for Reviewers' Responses**

Dear Reviewers ``nLPm``, ``pL2X``, and ``uk85``,

Thank you once again for your valuable feedback on our work! We understand all of you may have a busy schedule, but we kindly wish to remind you that **the discussion deadline is approaching in several days.** We would greatly appreciate it if you could let us know whether our rebuttal has sufficiently addressed your concerns or if there are any additional points you would like us to clarify.

We look forward to hearing from you.

Best regards,

The Authors

---

### Author Response · Authors · 2024-11-25
**Kinderly reminder of a new version**

Dear Reviewers ``nLPm``, ``pL2X``, and ``uk85``,

Thank you very much for all of your valuable feedback on our work! We have updated a new version according to the reviewers' concerns:
- We have added some citations in "Related work" part;
- We have provided a more detailed proof for the sub-Gaussian noise case, please refer to Appendix C.

Best,

Authors.

---

### Meta-Review · Area_Chair_gXec · 2024-12-21

**Metareview:**

This paper studies the convergence of Adafactor, a memory-saving optimization method, for non-convex smooth problems. The authors give theoretical results for both deterministic (full-batch) and stochastic versions of Adafactor. They also look at how adding update clipping affects the method. The paper uses special proof techniques for Adafactor and includes some experiments to support the theory. Reviewers appreciated the analysis of Adafactor, and they found the proofs clear and easy to follow. The topic is interesting because Adafactor is widely used in machine learning.
However, the paper has some problems. The authors use strong assumptions, like bounded gradients and bounded iteration norms, which make the results less general. Some recent works on optimization do not need these assumptions, so the paper feels behind in terms of theoretical advances. The experiments are not strong enough and do not compare Adafactor with other popular methods, like Adam or RMSProp. Also, the theoretical bounds in the paper are not as precise as in other recent studies, and adding update clipping does not give much improvement in performance. Reviewers like the main idea and suggest that the authors improve their theory, address the assumptions, and improve the experiments.

**Additional Comments On Reviewer Discussion:**

During the rebuttal, the authors responded to concerns about strong assumptions in their analysis, like bounded gradients and iteration norms, and explained that these are common in earlier works. They added that removing these assumptions is possible but would require significant changes to the method or new experiments. The authors also clarified details in their theoretical bounds and addressed minor presentation issues, such as typos and unclear figures. However, the updates did not fully resolve the main concerns, especially about the need for stronger theoretical results and better experiments comparing Adafactor with other methods. While the changes improved the paper slightly, the main weaknesses remained.

---

### Decision · Program_Chairs · 2025-01-22

Reject